# Why people follow rules

**Simon Gächter** [1,2,3] ✉, **Lucas Molleman** [1,4] **& Daniele Nosenzo** [1,5]

Why people follow rules, especially laws and social norms, is debated across the human sciences. The importance of intrinsic respect for rules is particularly controversial. To reveal the behavioural principles of rule-following, we develop CRISP, an interdisciplinary framework that explains rule-conformity $C$ as a function of intrinsic respect for rules $R$, extrinsic incentives $I$, social expectations $S$ and social preferences $P$. We deploy CRISP in four series of online experiments with 14,034 English-speaking participants. In our baseline experiments, 55–70% of participants conform to an arbitrary costly rule, even though they act anonymously and alone, and violations hurt no one. We show that people expect rule-conformity and view it as socially appropriate. Rule-breaking is contagious but remains moderate. Pro-social motives and extrinsic incentives increase rule-conformity, but unconditional rule-following and social expectations explain most of it. Our results demonstrate that respect for rules and social expectations are basic elements of rule-conformity that can explain why people follow laws and social norms even without extrinsic incentives and social preferences.

Rules regulate social life[1–3] and are fundamental for maintaining stable social order and fostering large-scale cooperation in society[4–13]. Rules are principles or maxims that prescribe or proscribe a particular standard of behaviour ('do $x$!', 'don't do $y$!') and come in various forms, as orders, regulations and guidelines issued by authorities, as laws and legal statutes, and as informal social and moral norms[14–18]. Rules classify actions as permissible or impermissible and can be formal or informal, written or unwritten, flexible or rigid, clear or ambiguous, explicitly stated or implicitly assumed, and enforced or not enforced. Rules shape everyday practices in personal, professional and political affairs in numerous ways, such as in rules that coordinate behaviour and rules that govern social relations; rules that demand or prohibit certain speech and conduct in private, in public and in the political sphere; and rules that define etiquettes of behaviour and dress for different social situations. In short, '[w]e are, all of us, everywhere, always, enmeshed in a web of rules that supports and constrains'[3] our actions and interactions.

Despite the omnipresence of rules in everyone's lives, the fundamental reasons why people follow them are not well understood. This lack of understanding is possibly due to a confluence of motives, which are debated controversially across the human sciences[19,20]. Do people follow rules out of obedience to authority or respect for tradition[6,7]? Or do people have an intrinsic respect for a rule because the rule says 'do $x$!' and therefore people perceive it as a 'deontic constraint'[21], that is, an unconditional, non-instrumentalist requirement that places an interior demand ('a duty') on how they should behave? Or do people follow rules out of self-interest, fearing that the potential costs of sanctions outweigh the benefits of breaking them[22,23]? Do people conform to rules to meet social expectations (what they think others will do or demand of them)[1] or do they follow rules due to pro-social motivations[24–28], considering the impact of their behaviour on others?

In this Article we present a conceptual experimental framework, called CRISP, that is motivated by the questions raised above and that identifies the proximate behavioural channels that generate conformity ($C$) with rules: intrinsic respect for rules ($R$), extrinsic incentives ($I$), social expectations ($S$) and social preferences ($P$). CRISP incorporates arguments and evidence from across the behavioural sciences[1,15,16,19,20,22,29–43] and is inspired by related efforts to integrate the behavioural sciences[44,45].

¹Centre for Decision Research and Experimental Economics, University of Nottingham, Nottingham, UK. ²IZA Institute of Labour Economics, Bonn, Germany. ³CESifo, Munich, Germany. ⁴Amsterdam Brain and Cognition, University of Amsterdam, Amsterdam, The Netherlands. ⁵Department of Economics and Business Economics, Aarhus University, Aarhus, Denmark. ✉e-mail: simon.gaechter@nottingham.ac.uk

In CRISP, the most basic reason for rule-conformity is that it may occur for unconditional (intrinsic) respect for rules: people might conform out of a sense of duty to obey a rule[12,19], that is, because they unconditionally respect the prescriptive or proscriptive demand a rule places on them[21]. According to social theorists, intrinsic respect for rules is the most fundamental channel to maintain social order[19,21]. The reason for this is that there are many situations where sanctions are weak or absent (that is, extrinsic incentives suggest breaking the rule), and consequences for others of breaking a rule are not salient[8–10,12,15,19].

By contrast, many economists neglect intrinsic motives and instead argue that conformity with rules results from extrinsic incentives that are high enough[11,22]. Extrinsic incentives can be based on the economic costs and benefits of following or breaking a rule (for example, the expected cost of rule-following; expected formal or informal sanctions for rule-breaking)[22,23], but can also consist of reputational incentives[46,47]. Many social scientists argue that people follow rules to conform with social expectations, regardless of extrinsic incentives—people want to conform with what they expect others will do and think ought to be done[1,31,34,48,49]. Finally, people with social preferences (pro-social motives)[24–28] might conform with rules because rules often exist to prevent harm or to provide benefits to others[29,50].

Based on our CRISP framework, we conducted four sets of experiments with a total of 14,034 participants. Our approach in experiment 1 is to eliminate all conventional reasons to follow a rule by creating a strong material incentive to violate it. Participants act anonymously and alone, which removes reputational concerns. We also eliminate any consequences of rule-following decisions for others, to mute pro-social motives.

Keeping the settings of experiment 1, experiment 2 measures social expectations in a variety of ways. Measuring social expectations is important, because they can exist in any situation, and people may have a desire to conform to these expectations[1,31,37,48,49,51,52]. Humans can have social expectations even when anonymous and alone, because people routinely rely on the capacities of mind-reading and perspective-taking[20,53,54]. The capacity of mind-reading allows people to form beliefs regarding what others expect of them (their normative beliefs) and what others are doing (their descriptive beliefs).

In experiment 3 we embed rule-following into a social context where participants observe the rule-following of other people, which is a realistic characteristic of many naturally occurring situations. Experiment 3 is also a test of the predictions from experiment 2 (and CRISP) that rule-following is on average conditional on social expectations.

In a final set of experiments (experiment 4), we add social consequences of rule-following and sanctions for rule-breaking to our baseline set-up. These experiments allow for other-regarding concerns and provide extrinsic incentives for rule-following. Although these added motives increase rule-following, we find that the baseline rate of rule-following already explains most of it. These results suggest that intrinsic respect for rules and social expectations are the key motives underlying rule-following; other-regarding concerns and extrinsic incentives, while enhancing rule-following, are not necessary to achieve high rates of rule-following in the first place.

## The CRISP framework

CRISP formalizes the four channels introduced above and explains rule-conformity $C$ as a function of four motivations:

CRISP : Rule Conformity = $f$(Intrinsic Respect for Rules;

Extrinsic Incentives; Social Expectations; Social Preferences)

CRISP assumes that, when deciding whether to conform with a rule or not, people weigh up all four elements of CRISP.

CRISP guides the interpretation of our experiments. We manipulate the extent of extrinsic incentives, which can encourage or discourage rule-following. Negative extrinsic incentives imply that rule-following is costly, and it creates an incentive to break the rule. Conversely, with sufficiently strong positive extrinsic incentives, rule-following is in the decision-maker's self-interest. Our experiments also control whether following or violating a rule generates a positive or negative externality (a harm or benefit) for someone else. If an externality is present, there is room for social preferences (pro-social motives) to play a role; otherwise, they are muted.

Central to CRISP are the social expectations (beliefs) that humans can entertain in any situation. Social expectations exist in two empirically measurable forms that can be separate sources of rule-conformity: 'normative beliefs' and 'descriptive beliefs'[1,48,49]. Normative beliefs ($b^n$) represent the extent to which people believe conforming with a rule is socially appropriate and violating the rule is deemed socially inappropriate by most other people. Descriptive beliefs ($b^d$) describe people's perceived degree of rule-conformity of other people. Crucially, for social expectations to influence rule-following, it must be the case that people make their rule-conformity dependent on their social beliefs: we therefore also elicit the functions $n(b^n)$ and $d(b^d)$, that is, we elicit people's rule-conformity for various increasing levels of the percentage of people $b^n$ who disapprove of a rule violation (resulting in the function $n(b^n)$), and, similarly, for increasing levels of the percentage of people $b^d$ who conform with the rule (resulting in the function $d(b^d)$).

CRISP identifies intrinsic respect for rules as any rule-following that is independent of social expectations, social preferences and incentives: following a rule regardless of what others do or think about it, that is, for all normative or descriptive beliefs, and regardless of incentives and social preferences. Unconditional disrespect for a rule is also possible, which manifests itself in breaking the rule regardless of social expectations (and possibly regardless of incentives and social preferences).

## Following an arbitrary rule in a highly stylized set-up

We conducted our four experiments (Methods and Supplementary Information) with 14,034 participants (average age 34.6 years, 49% female; demographic details are provided in Supplementary Table 1). We recruited most participants on the online platform Amazon Mechanical Turk. Our decision settings use a minimalist 'traffic light task' (Fig. 1a) suitable for online participants. This task is a simplified version of the rule-following task originally introduced by Kimbrough and Vostroknutov[55]. We also developed an even more abstract version with the same logic (Fig. 1b). This abstract task removes any naturalistic content that might guide people's behaviour in the traffic light task[55,56]. Participants must move a circle to a red traffic light (or a grey waiting area in our abstract task), and then to a finishing line. Participants have a starting capital of 20 money units (MU), which visibly counts down by 1 MU per second (Fig. 1a,b). The instructions (Supplementary Information) explain that participants earn the most by moving as fast as possible to the finishing line; that is, participants can jump the traffic light at any point in time. Adopting a previous approach, we implemented the rule by explicitly stating that 'The rule is to wait at the stop light until it turns green'[55] ('… until the cross disappears').

In experiments 1 to 3 (but not in some experiments in experiment 4), participants lose about 50–60% of their MUs if they follow the rule (that is, extrinsic incentives are negative). Hence, the tasks introduce a tension between rule-following and material self-interest, which is a feature of many real-world rules. The tasks are also non-strategic and asocial; unlike in previous tasks that have been used to study social norms[36], there is a single anonymous decision-maker, and a violation affects nobody, which mutes pro-social motives. Thus, in terms of CRISP, other than an intrinsic respect for rules and a desire to conform to social expectations, there are no reasons to follow the rule.

We ran our experiments in two conditions: without and with control questions about the monetary consequences of rule-following

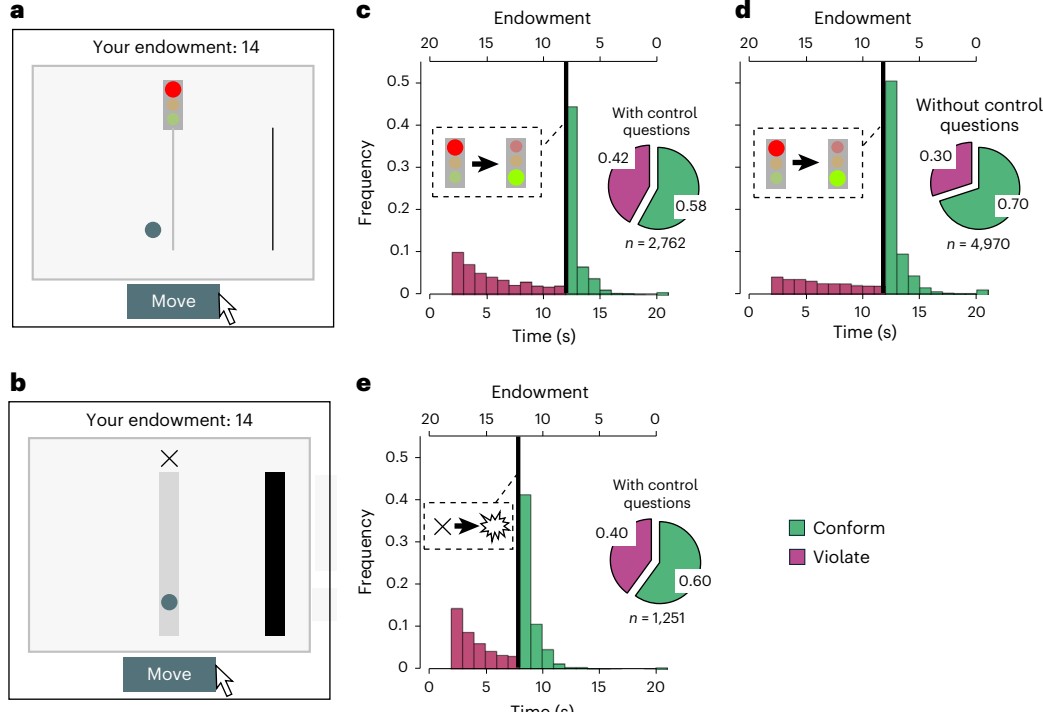

**Fig. 1 | People tend to conform with an arbitrary rule against their self-interest, even a stylized, asocial and unenforced rule stated by the experimenter.** **a**, Experimental task in which people are instructed to follow a rule in a minimalist traffic light setting where the rule read 'The rule is to wait at the stop light until it turns green'[55]. **b**, Abstract task setting without a traffic light context, where the rule read 'The rule is to wait in the grey area until the cross disappears'. In **a** and **b**, violating the rule maximizes own earnings without affecting anyone else. Participants must move a circle across the screen by clicking a 'Move' button.

The endowment indicated on top of the decision screen starts at 20 MU and ticks down by 1 MU per second. **c**–**e**, Frequencies of moving patterns and percentages of rule-conformity (green) and rule violations (purple), in settings with control questions (**c**,**e**) and without control questions (**d**) that test the understanding of payoff consequences. The Methods and Supplementary Sections 1.1 and 1.2 report further background details on procedures and the traffic light task and abstract task, respectively. The video TrafficLightTask_nopeers.mov in the Supplementary Information shows how the task looked like to participants.

(Supplementary Section 4). The condition with control questions arguably increases attentiveness and the salience of monetary incentives and thereby reduces the likelihood that rule-following is merely due to inattention or mistakes.

We ran experiment 1 in two versions: with the traffic light task (experiment 1a) and with the abstract task (experiment 1b). In experiment 1a we observed a high degree of rule-conformity (Fig. 1c,d). The average rate of rule-following across the two control questions' conditions was 65.6% [95% CI: 64.5–66.7%]. In the condition without control questions ($n$ = 4,970), 70.0% [68.7–71.2%] of participants conformed with the rule; with control questions ($n$ = 2,762) rule-following was 57.8% [55.9–59.6%].

Experiment 1b (Fig. 1b) with the abstract task was only run with control questions. The rule-following rate of 59.7% [57.0, 62.4%] of experiment 1b (Fig. 1e) confirmed the results from experiment 1a with control questions (Fig. 1c). Thus, while the salience of incentives reduces rule-following, there still exists substantial rule-following in conditions with control questions in both versions of experiment 1.

We replicated experiment 1a in a university laboratory. The physical laboratory offers more control than the online laboratory, but participants are less anonymous than online because the experimenter knows the identities of participants. We ran the experiments with control questions at the University of Nottingham ($n$ = 103 students) and found a rule-following rate of 60.2% [50.7, 69.6%] (Extended Data Fig. 1). A replication conducted in a laboratory in Berlin, Germany, using the traffic light task with 10- to 20-year-olds as participants, reports 52.7% rule-following[57]. Kimbrough and Vostroknutov's seminal study, which was also conducted in a university laboratory, reports a rule-following rate of 62.5% (ref. 55). These findings suggest that our online results are

not due to diminished control in the online environment (nor fear of not receiving approval after the experiment in the online experiment)[58].

Further replications of our baseline task (as part of separate research questions) with participants recruited from Prolific (traffic light task; $n$ = 363 Americans)[59] and the Qualtrics panel (abstract task; $n$ = 192 Swedish participants)[60] yielded rule-following rates of 58% and 55%, respectively. These results suggest the robustness of rule-following rates across platforms. High rates of rule-following (with little demographic variability[61]) were also observed in another abstract task, the 'ball-division task', where participants were told that the rule is to put balls into a blue bucket although a yellow bucket earned more money[56]. Their experiments revealed similar rule-following rates across participants from five countries ($n$ = 1,138 participants; average rule-following rate of 56%)[56]. In an experiment with the ball-division task conducted in two waves one month apart with the same individuals, individual propensities to follow the rule turned out to be stable across the two waves (rule-following rates of 60.8% and 62.2%)[62].

To put the rule-following rates further into perspective, participants in our experiments gave up 48.0% [46.9, 49.1%] of their possible earnings by following the rule (implied by moving times; Fig. 1). This rate is higher than the earnings people give up in dictator games[26] (where people on average forgo 28% of their possible gains), but lower than the fraction of possible gains (75%) people forfeit in honesty tasks where cheating is a form of rule violation[63].

Rule-following varied little with sociodemographics and the Big Five personality traits[64] (the coefficients are very small and unsystematic, similar to findings in a repeated traffic light task[61]), but was positively related to being 'patient'[65] and being 'shame-prone'[66] (Supplementary Tables 2 and 3). The result for patience lends

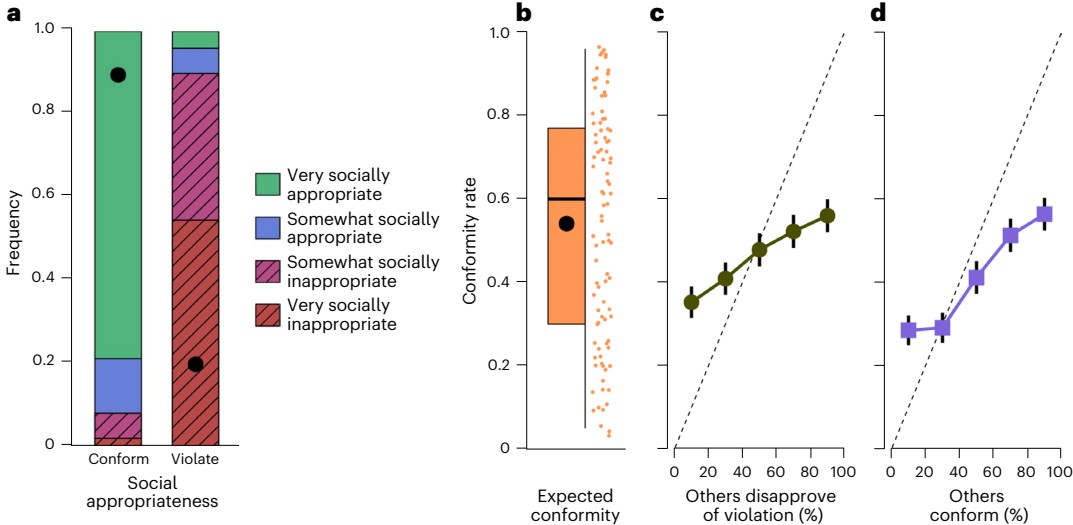

**Fig. 2 | Rules generate social expectations and conditional conformity with them, even in a minimalist set-up. a**, Perceptions of social appropriateness by spectators of rule-following and rule-breaking ($n = 99$) using an incentive-compatible mechanism of eliciting normative beliefs[51]. **b**, Descriptive beliefs of rule-following, as estimated by spectators ($n = 95$; paid for accuracy of estimate). Boxplot shows the distribution of spectators' expected portions of previous participants conforming with the rule (that is, waiting for the light to turn green; box, quartiles; median, thick solid line; mean, black dot; vertical line, range); orange dots are individuals' estimated conformity rates (jittered). **c,d**, Average conditional rule-conformity, elicited with the strategy method. The dashed line

is the diagonal. We elicited people's conformity or violation as a function of the percentage (presented as quintiles) of others who disapprove of a rule violation (normative conditional rule-conformity $n(b^n)$, $n = 158$) (**c**) and conform with the rule (descriptive conditional rule-conformity $d(b^d)$, $n = 159$) (**d**). Dots show ±1 s.e.m. Rule-conformity is significantly higher than zero when either normative or descriptive beliefs are 0 (linear mixed models with 'participant' as random intercept and 'portion others disapprove of violation' (cf. **c**) or 'portion others conform' (cf. **d**) as fixed effect have intercepts significantly higher than 0 (0.33 and 0.22, respectively; $P < 0.001$ for both; Supplementary Table 4 provides models controlling for age and gender).

psychological credibility to our task, because impulse control contributes to the psychological costs of rule-following in our setting. The result on shame-proneness corroborates previous evidence that rule-following is positively correlated with shame-proneness[67]. Rule-followers are also more likely to be conditional cooperators in a social dilemma task[37]. These correlations suggest that our rule-following tasks capture a general tendency to follow rules, which activates psychological mechanisms that also may affect behaviour in other settings[55,56,61,68].

The results shown in Fig. 1 only inform us about a high prevalence of rule-following when it is costly and when it cannot affect anyone (no externalities). In terms of CRISP, rule conformity $C$ can therefore only be due to intrinsic respect for rules $R$ and social expectations $S$. As argued above, people may form social expectations and conform to them even when anonymous and alone, and in the absence of extrinsic incentives or considerations for others. The next set of experiments, in experiment 2, quantify the importance of social expectations for rule-following in our socially minimal set-up.

## Social expectations and conditional rule-conformity

A core element of CRISP, and the research question of experiment 2, is social expectations and how they influence rule-conformity. To measure social expectations and to see how they translate into rule-conformity, we designed four between-subjects tasks (Methods), where, in all tasks, rule-following is costly and there are no externalities (that is, social preferences are muted). Two tasks elicit, respectively, normative beliefs $b^n$ and descriptive beliefs $b^d$ of rule-following from participants who had no experience with the task before ('spectators'). Two further tasks elicit people's conditional rule-conformity as a function of their normative and descriptive beliefs, $n(b^n)$ and $d(b^d)$, respectively.

Eliciting the conditional rule-conformity functions $n(b^n)$ and $d(b^d)$ is crucial for the following reason: some people may only conform with the rule if sufficiently many others disapprove of rule violations, or if

sufficiently many others do follow it too, and thresholds may differ between people. Eliciting $n(b^n)$ and $d(b^d)$ also allows us to determine unconditional rule-followers as a proxy for those who have an intrinsic respect for rules, regardless of their social expectations, and unconditional rule-violators as a proxy for those who disrespect rules even if they believe most other people do follow them and approve of the rule.

We measured normative beliefs of rule-following $b^n$ by using an established incentivized technique[51]. Rather than asking people's personal opinions about how appropriate or inappropriate they thought rule-following was, we asked participants, as 'impartial spectators', to report how socially appropriate or inappropriate they thought most people thought an act of rule-following or an act of rule-breaking would be. We measured descriptive beliefs $b^d$ by asking spectators to guess the percentage of previous participants who conformed with the rule. We incentivized guesses by rewarding accuracy (Methods and Supplementary Sections 1.4 and 1.5).

We found that our rule generated strong normative and descriptive beliefs of rule-following. Despite the non-strategic and asocial nature of our task, nearly 80% of impartial spectators perceived rule-following as very socially appropriate, and 90% perceived rule-breaking as very socially inappropriate or somewhat socially inappropriate (Fig. 2a). Regarding descriptive beliefs, the spectators' median expectation was that 60% would conform with the rule (Fig. 2b). This expectation is in line with the actual rule-following rate observed in the experiment (Fig. 1c).

To measure how social expectations influence rule-conformity, we elicited the conditional conformity functions $n(b^n)$ and $d(b^d)$. Participants had to decide whether to follow or break the rule in five different scenarios. The scenarios provide an exogenous variation in beliefs, presented as quintiles ((0–20%), …, (81–100%)), about what others said was socially appropriate to do (in the normative beliefs condition) or what others did in the task (in the descriptive beliefs condition). We used the strategy method[37,69–73] to elicit within-subject responses of $n(b^n)$ and $d(b^d)$ for each of the five scenarios of either normative

or descriptive beliefs. Thus, the elicited conformity functions provide individual thresholds of others' rule-conformity at which people will conform with the rule as well[74] (Methods and Supplementary Section 1.6).

People's elicited rule-conformity is, on average, conditional on their normative and descriptive beliefs (Fig. 2c,d). Regarding normative conditional rule-following $n(b^n)$, when only a minority of people (between 0 and 20%) disapprove of rule-breaking, the average rule-conformity rate is 35.2% [27.8, 42.6%] (estimated intercept at 0% disapproval rate is 33.2%, $P < 0.001$; Supplementary Table 4). As the share of people who disapprove of rule-breaking increases from the (0–20%) quintile to the (80–100%) quintile, average normative rule-conformity increases from 35.2% to 56.0% (Fig. 2c).

Individual response profiles of $n(b^n)$ reveal the proportions of unconditional rule-followers and rule-breakers, and those who have a threshold level of the share of others disapproving of a rule violation for moving from rule-breaking to rule-following. Nineteen percent of people conformed with the rule regardless of how many others disapproved of a violation (they are unconditional rule-followers); 29% always violated; 30% followed after a threshold; and for 22%, rule-conformity did not monotonically increase with the share of others disapproving of violations (Extended Data Fig. 2).

Elicited rule-following is on average also conditional on whether other people are believed to conform with the rule. Average descriptive conditional rule-conformity $d(b^d)$ ranges from 28.5% [21.4, 35.5%] when hardly anyone else is believed to conform with the rule (in the (0–20%) quintile), to 56% [48.6, 64.1%] when most other people (in the (80–100%) quintile) are expected to conform with the rule (Fig. 2d). The individual response profiles of $d(b^d)$ reveal that 20% conform in all five quintiles (that is, their rule-following is unconditional on descriptive beliefs); 34% never conform (they are unconditional rule-violators); 31% conform conditionally, that is, their threshold is a quintile of descriptive beliefs larger than the lowest one; and 15% had non-monotonic responses, that is, their rule-conformity did not monotonically increase with the share of others following the rule (Extended Data Fig. 2).

The results of Fig. 2c,d suggest that, on average, conformity is conditional in our asocial and anonymous setting. They also suggest that some rule-following is independent of social expectations—even if very few people disapprove of breaking the rule, or if almost everyone else breaks the rule, many participants still followed the rule in our non-social decision setting where nobody was affected by rule-breaking.

Taken together, the results from Fig. 2 suggest that the rule-following rates observed in Fig. 1c,d reflect a mixture of unconditional and conditional rule-following based on social expectations (normative and descriptive beliefs). Our results show that even arbitrary rules in an anonymous and asocial setting with only one decision-maker generate social expectations that the rule should be followed and is followed, demonstrating a fundamental relationship between rules and social expectations—social expectations about following the rules can arise even without a justification for a rule that does not serve an obvious purpose. The mere existence of a rule seems to be enough for triggering social expectations of rule-following.

The following experiments (experiment 3) test a key implication of our results on social expectations. Because rule-following is conditional on descriptive beliefs (Fig. 2d) for many people, changed descriptive beliefs should change rule-following. However, eliciting conditional rule-conformity with the strategy method might be considered psychologically 'cold', and observing others actually following or breaking the rule might lead to more visceral rule-following or rule-breaking. Moreover, in many naturally occurring situations, people often do observe whether other people follow the rules or not. Observing peer behaviour provides information that feeds into descriptive beliefs, which, according to the result from Fig. 2d, should be positively related to own rule-following. Figure 2d suggests we should see rule-following rates of at least 28% even if all observed peers violate the rule.

## Testing the causal role of observing peer rule-following

The results reported in Fig. 2d suggest that, on average, people's rule-following will depend positively on how many other people follow the rule. In experiment 3 we test this prediction using a modified version of our traffic light task in which participants observe how other participants had previously behaved in the task. In terms of CRISP, like in experiments 1 and 2, extrinsic incentives are negative (that is, rule-following is costly) and there are no externalities (implying that pro-social motives are muted). Participants completed three sequential iterations of the task. After completing the task alone (iteration 1, Fig. 1a), in iteration 2 participants were randomly assigned to one of 28 treatments. In a baseline treatment, participants completed a second iteration of the same traffic light task alone, without any information about how other participants had behaved. In the other 27 treatments (Fig. 3), participants in the second iteration could see on their screen, in addition to their own circle, other circles programmed to display the dynamic movements of 'peers', sampled from real movements of previous participants. Figure 3a shows an example with six peers who violate the rule. In the third iteration, the participant completed the task alone again.

The 27 treatments in iteration 2 differed in the number of peers a participant observed (from 1 to 6), and in whether these peers followed or broke the rule. Observing peer behaviour exogenously manipulates descriptive beliefs of rule-conformity. Depending on the treatment, we manipulated beliefs towards more rule-following (when peers were shown to conform with the rule) or more violations (when peers broke the rule). Our design reveals how peer rule-conformity shapes rule-following relative to the no-peer baseline. We also varied the number of peers displayed on the screen to see whether the share or the absolute number of rule-followers or rule-violators in a group influences own rule-following.

In the no-peer baseline in iteration 2 of the task, the average rule-conformity rate was 77.2% [73.1, 81.2%]. Compared to this benchmark, participants who observed fully conforming peers showed a small and statistically insignificant increase in rule-following (Fig. 3b, blue squares; top row of Supplementary Table 5). This increase was similar across all group sizes (Supplementary Tables 5 and 6).

In contrast, the observation of just one rule-breaking peer led to a substantial decrease in rule-following. The decrease in rule-conformity for one rule-breaker is similar (around eight percentage points) for all groups with at least two peers. This observation is consistent with the 'bad apple effect', according to which just one defector can diminish cooperation in social dilemma problems[75] (it should be noted, however, that, depending on the environment, rule-following might have positive spillover effects on other behaviours such as honesty[76]). Observing all peers violating the rule led to a significant drop in rule-following, similar in magnitude for all group sizes (Fig. 3b, leftmost red circles in each subpanel; Supplementary Tables 5 and 6). When participants observed a mixture of rule-following and rule-breaking peers, the effects of the latter prevailed; the average effect was always a decrease in rule-conformity relative to baseline. We did not observe a group size effect in the contagiousness of rule violations—patterns of rule compliance appear to be driven by the proportion rather than the absolute number of violators.

Despite contagious rule violations, overall rule-conformity rates are still remarkably high in absolute terms. Even when six peers violated the rule, 55% [49.4, 61.2%] of participants still followed the rule. Focusing only on conditions with control questions, rule-following never dropped below 40%, even when six out of six peers violated the rule (Supplementary Fig. 1).

Participants in all treatments completed a third and final iteration of the rule-following task without peers. The participants of the baseline treatment (with no peers in either iteration) therefore had to choose three times alone whether or not to follow the rule.

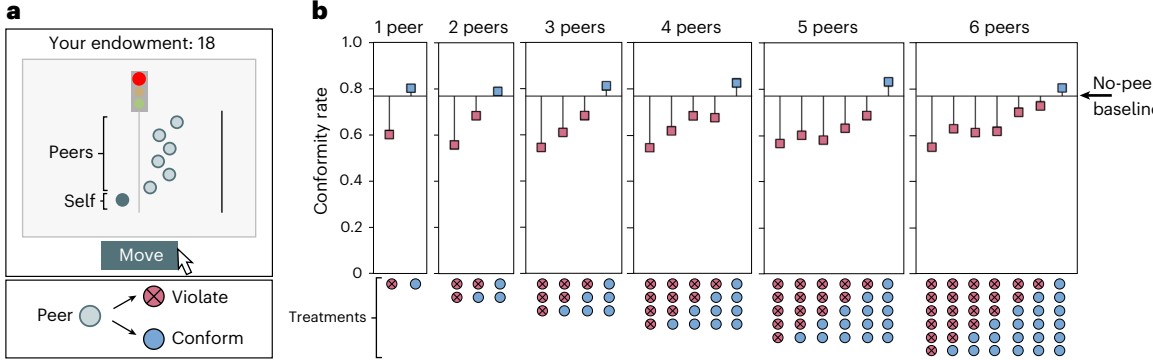

**Fig. 3 | Rule violations are contagious, but rule-following remains high.**
**a**, Screenshot of task with peers. The video TrafficLightTask_peers.mov in the Supplementary Information shows how it looked to participants. Shown is an example task with six peers who all violated the rule. **b**, Rates of rule-conformity in 27 between-subject treatments (total $n = 7,312$; $n > 257$ for each treatment) that exogenously manipulate group size (from one peer to six peers whose dynamic moving behaviour was sampled from participants of a pilot experiment), with

the number of peers either following the rule or violating it (blue and red circles (with a cross), respectively), at the bottom of the panels. Dots in the panels show rule-following rates (with changes from the no-peers weighted average baseline rule-conformity rate of 77%; horizontal line) in iteration 2 in treatments with and without control questions. Further details are provided in the Methods, Supplementary Section 1.2 and Supplementary Fig. 1.

Their rule-following rates across the three iterations were 74.0% [67.7, 80.4%], 84.5% [79.3, 89.8%] and 83.4% [78.0, 88.8%] without control questions ($n = 181$) and 49.8% [43.5, 56.1%], 63.2% [57.1, 69.3%] and 59.0% [52.8, 65.2%] with control questions ($n = 239$). Against this benchmark, those participants who had previously observed peers breaking the rule (in iteration 2) were more likely to break the rule later (in iteration 3, without peers). A 1% increase in observed peer violations in the second iteration increased participants' own rule-breaking in the third iteration by 0.12% (Supplementary Fig. 2 and Supplementary Table 7). Across all treatments, the lowest rule-conformity rate in iteration 3 was 58%. These results indicate that observing rule-following behaviour by peers can have lasting effects, which is relevant for the dynamics of rule-following.

In a complementary analysis ($n = 490$ new participants; Extended Data Fig. 3), we show that descriptive beliefs also influence normative beliefs. Observing rule-breaking does not change the social appropriateness of following the rule, but it makes rule violations less socially inappropriate.

## Rule-following with externalities and extrinsic incentives

In terms of CRISP, experiments 1 to 3 have established the properties of rule-conformity when extrinsic incentives encourage rule-breaking and social preferences do not matter because rule-breaking has no consequences for others. Our final set of experiments (experiment 4) adds consequences for others ('an externality') and weak and strong extrinsic incentives as motives for following the rule. Experiment 4 also measures social expectations ($b^n$, $b^d$) and elicit conditional conformity with them ($n(b^n)$, $d(b^d)$). We use between-subjects designs and the same techniques as in experiment 2 to see, in the CRISP framework, how consequences for others and extrinsic incentives affect social expectations, and conditional conformity with them.

The experiments in experiment 4 are relevant from a CRISP perspective. First, rules typically exist in situations where rule-breaking has consequences for others[29] and breaking a rule often triggers sanctions. Sanctions can be formal, as in the case of breaking the law, or they are informal (expressed as social disapproval), for example, when violating social norms[77]. Second, in terms of CRISP, this set of experiments provides a further opportunity to assess how much of rule-following can be explained by respect for rules and social expectations in settings where social preferences and incentives can also influence rule-following.

There were 4,045 participants in these experiments (Supplementary Table 1). All experiments were conducted between-subjects and

with control questions before the start of the traffic light task. We had four nested treatments. The first was a replication of our baseline traffic light task discussed above (treatment Baseline, BL), which provides a further assessment of the importance of intrinsic respect for rules and social expectations. Second, in treatment EX (treatment Externality) we added an externality to treatment BL. To create an externality—that is, a consequence for others—we provisionally donated US $1 to a well-known charity (the Red Cross). Participants were informed that violating the rule in the (otherwise unmodified) traffic light task cancels the donation to the charity. Thus, in terms of CRISP, a rule violation renders externalities negative (because the provisional donation is lost) and allows social preferences[24–28] to influence rule-following, while keeping the extrinsic incentive to violate the rule.

The third and fourth treatment added sanctions to the setting of EX. Sanctions were implemented by the experimental software and resulted in a loss of all MU for that decision. We varied the probability of being punished for a rule violation, from 10% (treatment Weak Punishment; WP) to 90% (treatment Strong Punishment; SP). Participants knew the consequences of punishment and its probability. Thus, for a risk-neutral person, extrinsic incentives are negative (rule-following is costly) if the likelihood of punishment is 10%; and extrinsic incentives are positive (rule-following is monetarily beneficial) if a rule violation led to a loss of earnings with 90% probability. In short, in SP, all four motivational elements of CRISP can influence rule-following behaviour.

In terms of CRISP, our experiments provide a causal analysis of how externalities, which enable pro-social motives, and extrinsic incentives influence social expectations and rule-conformity. Here we focus on rule-conformity and how much of it can be explained by unconditional respect for rules and descriptive beliefs. Details of how externalities and weak and strong punishment influence social expectations ($b^n$, $b^d$) and how they translate into rule-conformity ($n(b^n)$, $d(b^d)$) are provided in Extended Data Fig. 4.

Rule-conformity in our replication baseline treatment BL was 55%, which is close to the conformity rate of 58% in the initial baseline experiment with control questions (Fig. 1) and also the level of descriptive beliefs $b^d$ in BL (Fig. 4, black dots). Because BL is nested in all other treatments, the rule-conformity rate of 54.6% [50.3, 58.9%] in BL provides a benchmark of the combined extent of unconditional rule-following and social expectations in our setting that exists without externalities and extrinsic incentives. Additional motives for rule-conformity in the CRISP framework, as provided by treatments EX, WP and SP, are expected to increase rule-following, which is what we observe (Fig. 4). For instance, compared to BL, the presence of an externality increased

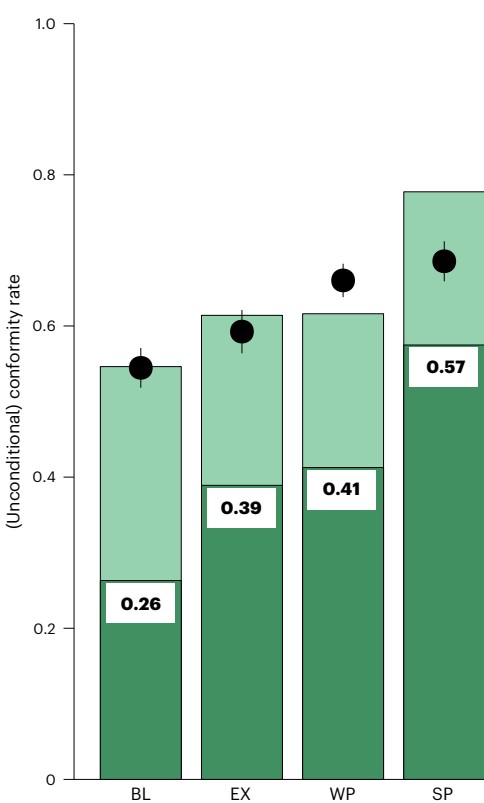

**Fig. 4 | Unconditional rule-following and social expectations explain a large part of rule-following in all conditions.** The four nested treatments consist of a replication of the baseline traffic light task without peers (BL) where only intrinsic respect for rules and social expectations can explain rule-following, and three treatments that cumulatively add motives to BL to follow the rule. Treatment EX adds an externality (a foregone donation of US $1 to a charity) triggered by a rule violation. Treatment WP adds a sanction of rule violations to treatment EX with a detection probability of 10% and treatment SP with a detection probability of 90%. Punishment implied the loss of all earnings (except the flat participation fee). Shown are mean rule-following rates in the behavioural experiment ($n_{BL} = 501$, $n_{EX} = 507$, $n_{WP} = 515$, $n_{SP} = 484$). Dots and whiskers show mean descriptive beliefs (±1 s.e.m.; $n_{BL} = 102$, $n_{EX} = 98$, $n_{WP} = 112$, $n_{SP} = 96$). The dark green shaded parts of the bars show the average proportions of participants who conform unconditionally regardless of their normative and descriptive beliefs (averages calculated from the strategy method data eliciting $n(.)$ and $d(.)$, those who always conformed; $n_{BL} = 152$, $n_{EX} = 156$, $n_{WP} = 154$, $n_{SP} = 157$). The full set of results are shown in Extended Data Figs. 4 and 5 and Supplementary Tables 9 and 10.

**Table 1 | Rule-conformity rates in the CRISP framework across experiments 1 to 4**

| | Rule-conformity according to CRISP | |
| --- | --- | --- |
| | **Rule-following rates (%)** | **Change or range of percentage points** |
| **Respect for rules** | | |
| Baselines (experiments 1 and 4) | 65 | |
| Control questions (no, yes) | 70, 57 | −13 |
| Unconditional $n(b^n)$ (experiments 2 and 4, BL) | 22 | |
| Unconditional $d(b^d)$ (experiments 2 and 4, BL) | 23 | |
| **Social expectations** | | |
| Conditional $n(b^n)$ (experiments 2 and 4, BL) $b^n$: lowest, highest quintile | 40, 60 | 20 |
| Conditional $d(b^d)$ (experiments 2 and 4, BL) $b^d$: lowest, highest quintile | 36, 60 | 23 |
| Observing peer behaviour (experiment 3) (lowest, highest observed rate) | 57, 81 | 24 |
| **Pro-social motives and extrinsic incentives (experiment 4)** | | |
| Social preferences possible (no, yes): comparing treatments BL and EX | 55, 61 | +7 |
| Weak punishment (no, yes): comparing treatments EX and WP | 61, 62 | +0 |
| Strong punishment (no, yes): comparing treatments WP and SP | 62, 78 | +16 |

**Respect for rules**: shown are the percentages of participants who conform with the rule in the baseline settings of experiments 1 and 4 and the fraction of people who follow whether control questions are present or not. Also shown are the rates of unconditional rule-following, that is, rates of rule-conformity for all quintiles of $b^n$ and $b^d$ regardless of how many disapprove of a violation (normative) or how many actually follow the rule (descriptive). Data are taken from strategy-method experiments 2 and 4; BL averages. **Social expectations**: shown are average rule-following rates for lowest (0–20%) and highest (81–100%) quintiles of $b^n$ and $b^d$, respectively. Also shown is the change or range in terms of percentage points. Data are taken from strategy-method experiments 2 and 4; BL averages. Observing peer behaviour in experiment 3 (a form of observing $b^d$) reports the lowest and highest rule-following rates and their range in percentage points. **Pro-social motives and extrinsic incentives**: shown are summary results of experiment 4. Social preferences possible (no, yes) measures the rule-following rates and their change when externalities are absent and when they are present (comparing treatments BL and EX). Weak punishment (no, yes) compares treatments EX when weak punishment is absent, with WP when it is present. Strong punishment (no, yes) compares treatments WP when strong punishment is absent, with SP when it is present. All percentages are rounded to integers.

rule-conformity by 6.8 percentage points, and the presence of strong incentives in SP increased rule-conformity by 23.1 percentage points to 77.8% [74.0, 81.5%] (Fig. 4, Tukey's honestly significant difference test (THSD): $P < 0.001$; Supplementary Table 11).

Interestingly, although SP strongly increased rule-conformity, it is less than the predicted 100%. This might be due to the probabilistic nature of the incentives, which, despite a punishment likelihood of 90%, might have led some people to gamble on getting away with breaking the rule. It can also be due to the fact that, in all treatments, including SP, many people's rule-following is conditional on others' rule-conformity (Extended Data Fig. 4d). If some people believe that others do not obey the rule, conditional conformity with social expectations might have induced rule-breaking. There is also a fraction of 6% of people who, irrationally (in terms of foregone payoffs), and maybe defiantly, violate the rule unconditionally (Extended Data Fig. 5).

The dark-shaded parts of Fig. 4 show the average fraction of people who conformed unconditionally. We estimated the fractions of unconditional rule-followers from the shares of those participants who followed the rule regardless of either their normative or their descriptive beliefs about others' conformity as elicited by $n(b^n)$ or $d(b^d)$. In BL, unconditional rule-following cannot be influenced by social preferences or fear of sanctions, and therefore the conformity rate of 26.9% [19.9, 34.0%] is a proxy for the extent of intrinsic respect for the rule. The presence of externalities (EX) and weak and strong incentives (WP and SP) not only increased overall conformity, but also the rate of participants who did not condition their rule-following on social expectations (Fig. 4; dark green bars).

## Putting the elements of CRISP into perspective

In our analysis so far, we have provided separate evidence for all elements of CRISP. Table 1 brings the evidence from experiments 1 to 4 together in a way that allows a comparison of rule-following rates and their change or range according to the motivational elements of CRISP: respect for rules (top), social expectations (middle) and pro-social motives and extrinsic incentives (bottom).

Our results suggest that respect for rules is a central reason for rule-following (Table 1, top). We can proxy this in two ways. First, the baseline average rule-following rate across the two baseline experiments (in experiment 1 and experiment 4, treatment BL) is 64.9% [63.9, 66.0]. Because extrinsic incentives are negative and no-one is affected by a rule violation, this means that, in terms of CRISP, rule-following is due to intrinsic respect for rules and/or conformity with social expectations that people might entertain even if anonymous and alone. This rate might also be affected by how salient the payoffs are—adding control questions reduces rule-following by 12.7 percentage points [10.6, 14.7], from 70.0% [68.7, 71.2%] to 57.3% [55.6, 59.0%]. The fact that rule-following is so high without control questions suggests that for many people the default reaction is to follow the rule (consistent with Hayek's dictum that 'man is … a rule-following animal')[9], and this is modified once control questions make the incentives to violate the rule salient. Control questions can reduce misperceptions[78], but also induce more reflective thinking[79]. The 22% and 23% of people who follow the rule unconditionally, that is, regardless of their normative or descriptive beliefs (Extended Data Figs. 2 and 5), provide a lower bound for the respect for rules in our data.

In CRISP, social expectations (Table 1, middle) can only influence rule-following to the extent that people condition their rule-following decisions on their normative or descriptive beliefs, as elicited by the functions $n(b^n)$ and $d(b^d)$. Here we restrict attention to experiment 2 (Fig. 2) and the BL treatment of experiment 4 (Extended Data Fig. 4). Moving from the lowest quintile of beliefs (0–20%) to the highest quintile of beliefs (81–100%) increased average rule-following rates by 19.9 [12.2, 27.6] and 23.2 [15.6, 30.9] percentage points, for normative and descriptive conditional rule-conformity, respectively. This range of percentage point changes for conditional rule-conformity is only slightly lower than the 24.8 [21.7, 27.8] percentage point change for observed peer behaviour as measured in experiment 3. Regression estimates (Supplementary Table 6) show that a 1% increase in violation by peers increased violations by 0.19%, which is slightly higher than the 0.13% estimate of peer effects observed in other laboratory and field experiments[80].

Table 1, bottom, draws on the nested treatments of experiment 4 (reported in Fig. 4) and shows how rule-following is affected by extrinsic incentives and an externality, which opens the possibility for social preferences to matter. Rule-conformity increases by seven percentage points if social preferences can play a role. Weak punishment, which still maintains an incentive to violate the rule, barely increases rule-conformity compared to when no punishment is possible at all. By contrast, strong punishment increases rule-conformity by 16 percentage points. Notice that the behavioural effect of extrinsic incentives is lower than unconditional rule-conformity or the change induced by social expectations.

## Discussion

Countless rules govern human social life every day and in all societies. To make progress in understanding ubiquitous rule-following behaviour and its importance for social order, we have introduced the experimental framework CRISP, which explains conformity with rules as a function of respect for the rule, incentives, social expectations and social preferences. Across four sets of comparable experiments (Figs. 1–4), we provided evidence for all four motivational elements of CRISP.

Perhaps our most important finding from the viewpoint of CRISP is that ~55–70% (Figs. 1 and 4) of rule-following cannot be explained by standard motives, because it happens in anonymous single-decision-maker settings where social preferences do not matter, and self-interest suggests breaking the rule. Under these circumstances, rule-conformity according to CRISP is due to a mixture of unconditional and conditional conformity with perceived social expectations (Fig. 2). We estimate that in our experiments ~23% of people

conform with the rule unconditionally, that is, independent of social expectations, and ~30% of people are conditional rule-followers. The 23% rate of unconditional rule-following is substantial when compared to the extent to which conformity with varying social expectations and extrinsic incentives shift rule-following (by 20–25 percentage points; Figs. 3 and 4 and Table 1).

We believe our evidence of substantial unconditional rule-following is an important result. The reason is that intrinsic respect for rules is arguably required for social order in many situations in which extrinsic incentives—which provide self-interest-based reasons for rule-following—are weak or absent, and consequences for others are not salient. This argument has been made long before us[5,9], but its importance has remained controversial, not least for lack of evidence for an intrinsic respect for rules[19]. With the help of CRISP, our study provides evidence for the existence of an intrinsic respect for rules. Our results also suggest that sanctions for rule violations and social preferences increase rule-conformity (Fig. 4) but may not be necessary to induce substantial rule-following. Our findings may therefore help explain why important types of rule—social norms, laws and orders—guide so much of human behaviour, because conformity with them is high without any extrinsic incentives.

Some caveats and tasks for future research are in order. First, our results stem from two minimalist stylized settings, the traffic light task and an abstract rule-following task, where the rule asks people to wait, thereby demanding patience. People in the traffic light task might also wait because the task is reminiscent of a real-world situation where the rule is inferred from the traffic light itself, without stating the rule explicitly[55]. Although we kept our setting constant across our four experiments, the quantitative effects we found are valid within that setting and may be different in tasks that do not require patience, such as the ball-division task[56]. More generally, and more importantly, rules in reality come in many different forms and vary, for instance, in ambiguity and observability of their violation[59]. Our CRISP framework (and its experimental methodology) can be deployed to study how the motivational elements of CRISP interact to produce rule-conformity of any rule (laboratory rules or naturally occurring rules). For instance, intrinsic rule-following is probably more important in many naturally occurring environments where rules have meaning[3,10,12,15,19,81,82], evoke personal norms and internalized values[38,41,45,83,84], trigger moral judgements[18], feelings of moral obligation[85] and guilt aversion[86], or appeal to self-image concerns[87]. It is also important to notice that rule-following in real-world settings can be socially harmful, such as conforming to bad social norms[88–91] or obeying rules that demand nefarious acts or committing atrocities[92–95]. Again, CRISP provides the toolbox to study the behavioural properties of such rules.

Second, our evidence is limited to one online population, Amazon Mechanical Turk workers in the United States, and one student pool at the University of Nottingham. A task for future research is to study methodological limitations of online participant pools for rule-following paradigms, and, more importantly, the generalizability of our results to other populations. Cross-cultural research using CRISP might be particularly promising[56,96]. Recent research suggests that distal ecological and historical processes[2], the rule of law[97,98] and the historical importance of kinship in the organization of social life[99,100] can explain differences in societal outcomes across contemporary societies—factors that might also impact rule-following and its behavioural determinants according to CRISP.

Third, our experiments are 'between-subjects', which does not allow us to study how the behavioural motives of CRISP are linked at the individual level. Because potentially all four motives behind CRISP can determine rule-following in many realistic settings, developing a within-subject framework that allows comparing the relative importance of the behavioural motives at the individual level is an interesting endeavour[101].

Fourth, except for experiment 3, we have studied rule-following in single-decision-maker settings without feedback of what other people actually did. In the experiments in experiment 3, people made only one decision after they saw what their peers did (Fig. 3). Single-shot decisions are unrealistic in situations where people can repeatedly observe the rule-following of other people. Related research using public goods and donation games suggests observing others might lead to less rule-conformity[37,102,103]. Moreover, as our results also suggest, more rule violations may also make them more socially acceptable (Extended Data Fig. 3). Future research should therefore study the fragility of rule-conformity[32], its population-level dynamics[20,39,104–106] and the consequences for normative beliefs[107–110].

Fifth, in our experiments, the rule was stated by an experimenter, and rule-following in our experiments is therefore a kind of 'experimenter-demand effect'[111], which is often seen as problematic. As was also pointed out by previous research[55,68], in our context, the experimenter-demand effect is not problematic, but rather the object of our study, because it is the nature of rules that they demand a certain behaviour. In reality, some rules (laws, regulations, orders, guidelines) are set by an authority and others (social norms) emerge via social learning[13,112–115]. In this Article, we made a start in studying the behavioural principles of following a clearly stated rule by using the authority of the experimenter to set the rule. Because, in reality, some rules are set and some emerge[114,116], studying the role of rule-setting and emergence through the lens of CRISP is another interesting task for future research.

## Methods

Here we describe the most important details of the methodology of our experiments. Details are provided in the Supplementary Section 1. The experiments received ethical approval from the Research Ethics Committee of the Nottingham School of Economics (protocol no. 030_ERC_AP_MT) and complied with all relevant guidelines. Participants provided informed consent before entering the experiments. The experiments were not pre-registered. Participants received a participation fee and were paid according to their decisions (details below). No deception was used.

### Procedures common across all experiments

Participants for most experiments were recruited from Amazon Mechanical Turk[58,117] in 2014–2015 (experiments 1 to 3) and in 2018 (experiment 4). One experiment was run in 2016 with students in the CeDEx lab at the University of Nottingham. Participation was restricted to Amazon Mechanical Turk workers from the United States to ensure understanding of English instructions (Supplementary Table 1 provides the demographic details of all experiments reported in this Article). To minimize selection into our experiments, descriptions only contained general information on the task.

We used the software LIONESS Lab (https://lioness-lab.org/)[118] to develop and conduct our experiments. All experiment instructions are provided in Supplementary Section 4. The online experiments were run under pseudonymous conditions; that is, it was impossible for the experimenter and any other participant to identify individual participants. In the laboratory experiments, participant identities were known to the experimenter.

### Overview of all experiments

We conducted four sets of experiments with a total of 14,031 unique participants. No participant was excluded. Here, we briefly summarize the experiments. Details of how we implemented them are provided below and in the Supplementary Information. No statistical methods were used to predetermine sample sizes. We explain our sample size considerations separately for each experiment.

To describe our experiments compactly, we use the following notation borrowed from our framework CRISP. Extrinsic incentives $I$ can be negative ($I < 0$) or positive ($I > 0$); that is, people have an incentive to violate the rule (if $I < 0$) or to follow it if $I > 0$ is large enough. We also manipulated whether a rule violation created an externality for a third party, called $X$, which, in our experiments, is either $X = 0$ or $X < 0$. If $X = 0$, no third party is affected by whether the rule is followed or not. If $X < 0$, a third party is harmed by a rule violation (loss of a provisional donation to the Red Cross). If $X < 0$, there is room for pro-social motives to affect rule-following decisions, whereas if $X = 0$ pro-social motives are muted. Our experiments also manipulated whether the participant could observe the rule-following decisions of peers or not; if not, peers = 0, if yes, peers > 0.

In experiment 1, $n = 8,983$, $I < 0$, $X = 0$ and peers = 0. Experiment 1 was conducted in two versions. Experiment 1 with the traffic light task is the first of three iterations, in all of which $I < 0$, $X = 0$, peers = 0. The second and third iterations serve as the baseline condition for experiment 3. The purpose was to establish the degree of rule-conformity in an easy-to-understand stylized context and under conditions with no standard reasons to follow the rule. Some experiments had control questions, and some had not, which allows us to see how the salience of payoffs affects rule-following. Sample size was determined by the requirements for experiment 3, resulting in 7,732 participants as explained in the following. The results are illustrated in Fig. 1c,d. Experiment 1 with the abstract task (Fig. 1b) was conducted with $n = 1,251$ participants and had the purpose to test rule-following behaviour in an entirely decontextualized setting. The results are shown in Fig. 1e.

In experiment 2, $n = 511$, $I < 0$, $X = 0$ and peers = 0. The purpose was to measure the social expectations elements of CRISP, which are the normative beliefs ($b^n$), the descriptive beliefs ($b^d$), and the normative and descriptive rule-conformity functions $n(b^n)$ and $d(b^d)$ in four between-subjects experiments as described in the following. Sample size was planned to be similar to related experiments[37]. The results are presented in Fig. 2.

In experiment 3, $n = 7,732$, $I < 0$ and $X = 0$. There were three iterations. In iteration 1 (=experiment 1) and iteration 3, peers = 0, and in iteration 2, peers > 0. The purpose of iteration 2 was to measure the role of observing the rule-following behaviour of peers (a form of descriptive belief) for own rule-following behaviour. In iteration 2, we had 27 between-subjects treatments that manipulated the number of peers a participant saw in iteration 2 (either 1 or 2, or … 6) and whether these peers violated the rule or not. We planned to have at least 250 participants per treatment. The realized range was 258–286, with a total of 7,732 participants. In the benchmark treatment, peers = 0 in all three iterations ($n = 420$). We also ran experiments to see how peers' rule-conformity decisions influence normative beliefs $b^n$. These experiments ($n = 491$ participants) are described in Extended Data Fig. 3.

In experiment 4, total $n = 4,045$ and peers = 0 in all experiments. All experiments had three iterations. We ran four series of experiments with a planned sample size of roughly double the sample size of each of the treatments in experiment 3 (~250 each). Experiment 4 employed a nested design—where motives to conform with the rule according to CRISP are added cumulatively—to establish the roles of externalities and weak and strong incentives for rule-conformity:

Experiment BL ($n = 501$) was a replication of experiment 1 with $I < 0$ and $X = 0$.

In experiment EX, $n = 507$, $I < 0$ and $X < 0$. The purpose was to measure rule-following when social preferences are possible because a rule violation creates an externality. Comparison of EX with BL reveals the degree to which pro-social motives influence rule-following.

In experiment WP, $n = 515$, $I < 0$ and $X < 0$. This experiment introduced weak punishment for a rule violation that does not change the incentives to follow the rule. Comparison of WP with EX reveals how the presence of weak incentives influences rule-conformity.

In experiment SP, $n = 484$, $I > 0$ and $X < 0$. This experiment introduces strong punishment for a rule violation that creates an incentive to follow the rule. Comparison of SP with WP reveals the relative importance of weak and strong incentives for rule-conformity.

For experiment 4, we ran a further set of four experiments, like those in experiment 2, with the purpose to measure the social expectations elements of CRISP, that is, $b^n$, $b^d$, $n(b^n)$ and $d(b^d)$ in between-subjects experiments for each of BL, EX, WP and SP. The planned sample sizes for each of the four elements of social expectations were similar to the samples sizes of experiment 2, with a total of 2,038 participants. Sample details are provided in Supplementary Table 1, and the detailed results in Extended Data Fig. 4.

### Reporting proportions of rule-conformity

For reported proportions of rule-conformity, we indicate the precision of our estimates using 95% confidence intervals (CIs) in square brackets. These CIs provide the range within which we expect the true population proportion to fall with 95% certainty if the study were repeated multiple times; that is, they are indicators of the precision of the point estimates. CIs are calculated as $p \pm 1.96 \times$ sqrt($p \times (1 - p)/n$), where $p$ is the sample proportion and $n$ is the sample size.

### Details of the behavioural experiments

**Tasks.** In most experiments we used a traffic light task, which is a simplified and more abstract version of a related tool (which had five traffic lights, a stick figure and a 'walk' button)[55]. In our task, participants controlled a circle and had to 'move' it across the screen to a single red traffic light, and then over a finish line (Fig. 1a; for a video illustration, see TrafficLightTask_nopeers.mov in the Supplementary Information). The first click triggered a continual move of the circle to the traffic light (where the circle stopped); upon the second click, the circle continued moving automatically and at constant speed to the finishing line. Participants started with an endowment of 20 MU, which was worth US $1. Each second, this endowment dropped by 1 MU, indicated at the top of the screen. Across all treatments, the traffic light turned green after 12 s. The instructions made it clear that the participant would make the most money by moving the circle across the finish line as fast as possible; that is, they could click anytime they wanted before the traffic light turned green. However, following ref. 55, the instructions also explicitly stated that 'The rule is to wait until the traffic light turns green'. It was clear that there was no sanction for violating this rule (except in treatments with punishment).

The abstract version of the task (Fig. 1b) was developed by us and worked the same as the traffic light task. Instead of the traffic light, there was an X above a grey waiting area. The rule said 'The rule is to wait in the grey area until the cross disappears'.

**Experimental conditions.** In total, 7,732 participants completed the traffic light task, in two conditions—without and with control questions. In the condition with control questions ($n = 4,970$), after the instructions, but before the actual task, we included two compulsory control questions about possible earnings for rule violation and rule-following to maximize participants' understanding of how earnings were computed. In the condition without control questions ($n = 2,762$), no questions were asked. The abstract task ($n = 1,251$) was only run with control questions.

Each participant completed three iterations of the traffic light task, following an A–B–A design. In the first iteration, referred to as part I, the participant completed the task alone (these data are the basis for Fig. 1c). In iteration 2 (part II), we implemented 27 treatments, systematically varying the number of peers a participant observed (from 1 to 6), and in how many of these peers followed or violated the rule (Fig. 3b). In each treatment, the actual moving behaviour of randomly selected actual previous participants (subject to the treatment composition) was shown (for a video illustration, see TrafficLightTask_peers.mov in the Supplementary Information). In the 28th treatment (the baseline treatment), no peers were shown in B, that is, for these participants, the task in B was identical to A. The final iteration (part III) was the same as the first part A. Participants again completed

the task alone. Moving before the traffic light turned green counted as rule-breaking, and moving after the traffic light turned green counted as rule-following (Fig. 1).

**Eliciting social expectations—normative beliefs ($b^n$).** We measured normative beliefs in the traffic light task using an incentive-compatible methodology[51]. Participants acted as spectators and were asked to rate the 'social appropriateness' of behaviours of a hypothetical decision-maker participating in the traffic light task. After reading the same instructions as the participants, the spectators were asked to evaluate two possible behaviours: (i) following the rule and (ii) violating the rule. Ratings were given on the following scale (with numerical value in brackets): (1) 'very socially inappropriate', (2) 'somewhat socially inappropriate', (3) 'somewhat socially appropriate' and (4) 'very socially appropriate'. Participants whose ratings where the same as a randomly matched other participant received a bonus.

**Eliciting social expectations—descriptive beliefs ($b^d$).** Participants acted as spectators and had to estimate the rule-following rate among previous participants in the traffic light task. First, spectators were shown the instructions that these previous workers had received. Second, after confirming that spectators understood the previous workers' task, spectators entered their estimate of the percentage of workers who obeyed the rule. If their estimate was no more than five percentage points off the observed percentage of rule-following, spectators received a bonus payment of US $1. Otherwise, they received US $0. Spectators were informed of this incentive scheme before they participated in the task.

**Eliciting normative and descriptive conditional rule-conformity functions $n(b^n)$ and $d(b^d)$.** Participants decided to follow or to violate the rule for different, exogenously induced levels of normative and descriptive beliefs. To measure these conditional rule-conformity functions, we used an approach based on the strategy method[37,70–72]. We used a between-subject design, where participants completed the elicitation of either $n(b^n)$ or $d(b^d)$. After reading the instructions, we asked participants to choose whether to follow the rule or to break it across five different scenarios. In each case, participants were informed of the exact payoffs associated with rule-following and violation of the rule. These payoffs were based on the actual payoffs from rule-following and violation in the behavioural task.

In the case of descriptive conditional rule-conformity as a function of descriptive beliefs $d(b^d)$, we elicited participants' preference for rule-following for scenarios that differed in the percentage of other participants who would follow the rule. In scenario 1, we told participants to imagine that between 0 and 20% of other participants would follow the rule and asked them whether they themselves preferred to violate or follow the rule in this situation, knowing what the costs of rule-following are. In the other four scenarios we elicited participants' preference for following the rule in the cases where 21–40%, 41–60%, 61–80% or 81–100% of other participants would follow the rule. The elicitation thus measures participants' conditional preference for rule-following $d(b^d)$, in correspondence to five exogenous levels of descriptive beliefs ($b^d$) about others' rule-following.

The elicitation for normative conditional rule-conformity as a function of normative beliefs $n(b^n)$ was similar except that we described different scenarios in which 0–20%, 21–40%, 41–60%, 61–80% or 81–100% of other participants would disapprove of rule violations.

The elicitation of both $n(b^n)$ and $d(b^d)$ was incentive-compatible (Supplementary Section 1.6).

**Payments.** All participants in all experiments received a participation fee and, as is common in experimental economics[119], were paid according to their decisions in an incentive-compatible way. Details are provided in Supplementary Section 1.

## Reporting Summary

Further information on research design is available in the Nature Portfolio Reporting Summary linked to this Article.

## Data availability

The data are publicly available at https://doi.org/10.17605/OSF.IO/7WZ4F (ref. 120).

## Code availability

The analysis code is publicly available at https://doi.org/10.17605/OSF.IO/7WZ4F (ref. 120).

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

## Acknowledgements

We are grateful for detailed comments on our paper from C. Bicchieri, A. Falk, H. Kliemt and B. Lindström, as well as for comments from B. Bartling, S. V. Burks, R. Cubitt, E. Dimant, C. Engel, E. Fehr, P. Gately, S. Gavrilets, G. Grimalda, N. Kerr, M. Kosfeld, E. Krupka, S. Kube, T. Lane, S. Lindenberg, J. Schultz, D. Sure, M. Sutter, R. Weber, and various conference and seminar audiences. S.G. also acknowledges the hospitality of briq Bonn while working on this paper. This work was supported by the European Research Council (grant nos. ERC-AdG 295707 COOPERATION to S.G. and ERC-AdG 101020453 PRINCIPLES to S.G.), the Economic and Social Research Council (grant no. ES/K002201/1 to S.G.) and the Luxembourg Institute of Socio-Economic Research (LISER to D.N.). The funders had no role in study design, data collection and analysis, decision to publish or preparation of the manuscript.

## Author contributions

S.G., L.M. and D.N. designed the experiments. L.M. conducted the experiments and analysed the data. S.G., L.M. and D.N. wrote the paper.

## Competing interests

The authors declare no competing interests.

## Additional information

**Extended data** is available for this paper at https://doi.org/10.1038/s41562-025-02196-4.

**Correspondence and requests for materials** should be addressed to Simon Gächter.

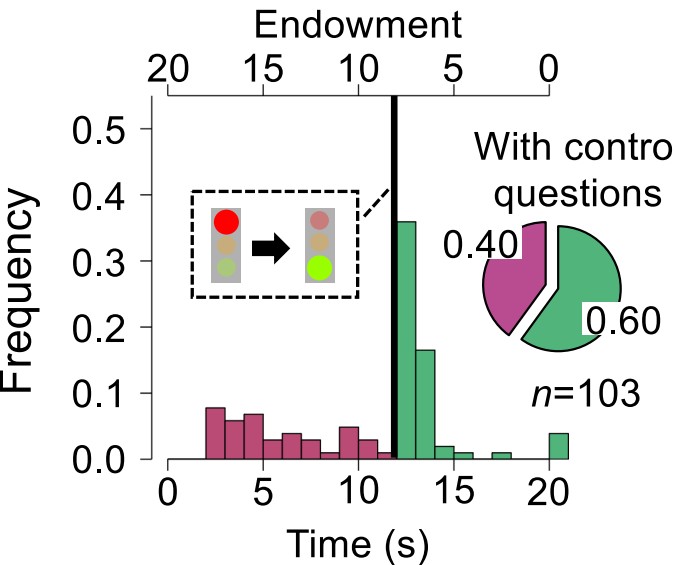

**Extended Data Fig. 1 | Rule-following in a student sample in a physical laboratory.** Experiments run in the CeDEx Lab at the University of Nottingham ($n = 103$ undergraduates). Purple: movements before the signal changed – violating the rule; Green: movements after the signal changed – conforming with the rule.

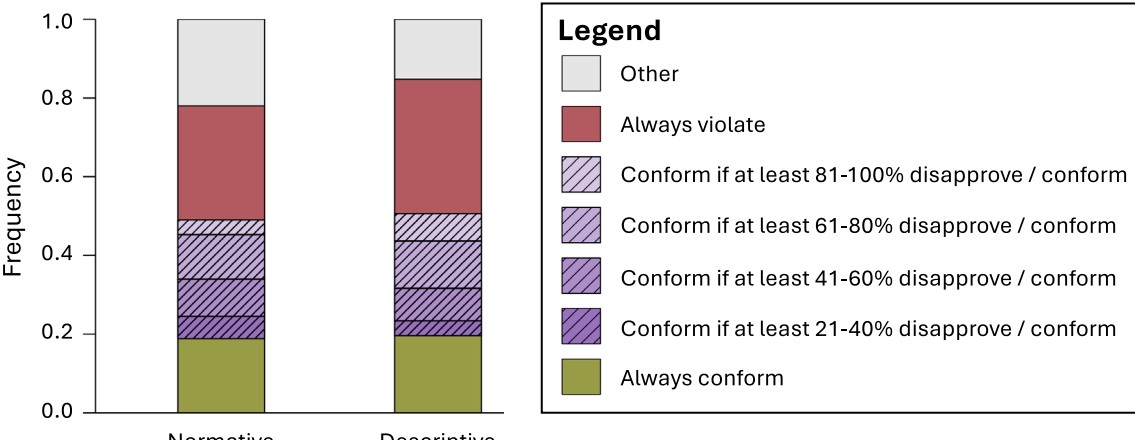

**Extended Data Fig. 2 | Individual profiles of rule-conformity conditional on social expectations.** This is the individual-level breakdown of Fig. 2c,d. The grey bars on top of the stacked bars reflect individuals whose rule-conformity did not monotonically increase with the percentage of others disapproving of violations (for normative beliefs $b^n$; left-hand-side bar) or conform with the rule (for descriptive beliefs $b^d$; right-hand-side bar).

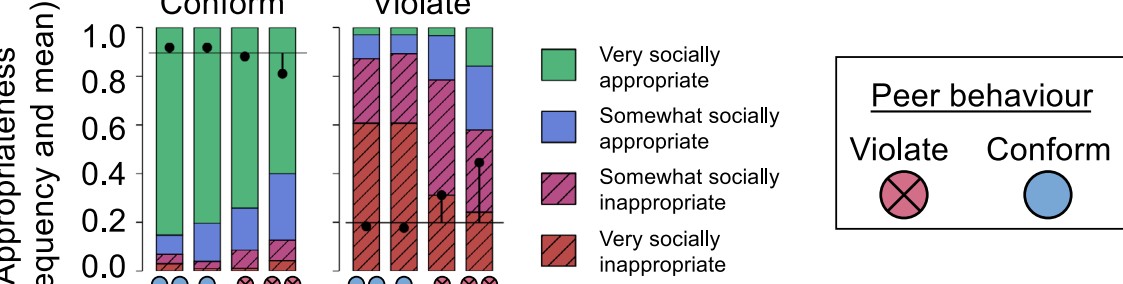

**Extended Data Fig. 3 | Observing rule-violations makes them socially more acceptable.** In these experiments ($n = 491$), we test the argument that descriptive beliefs $b^d$ may also influence normative beliefs $b^n$ if observing peers breaking the rule leads people to revise their normative beliefs. Shown is peer behaviour and perceived social appropriateness of actions: Perceptions of social appropriateness of conforming (left panel) and violating the rule (right panel) using an incentive-compatible mechanism of eliciting normative beliefs[16]. Bars show ratings for rule-following and rule violation, for four different between-subject conditions where either 1 or 6 peers follow the rule (blue circles) or violate it (red circles with ×). Black dots are normalised mean ratings with their deviations from the no-peers baseline rating (indicated with horizontal lines; these are the ratings in Fig. 2a in the main text). Extended Data Fig. 3 (and the

regression in Supplementary Table 8) support the hypothesis that descriptive beliefs causally influence normative beliefs. When peers (both six peers and one peer) conformed with the rule (left panel), normative beliefs about rule-conformity did not change relative to the no-peer baseline (Fig. 2a in main text). This was also the case when only one peer broke the rule. Even when six peers broke the rule, the mean rating of the social appropriateness of rule-conformity only changed little: at least 87% of participants still thought that conforming with the rule was very or somewhat socially appropriate. In contrast, the social appropriateness rating for rule violation (right panel) increased with the number of violators: when six peers violated the rule, 42% of participants found it at least somewhat socially appropriate to break the rule.

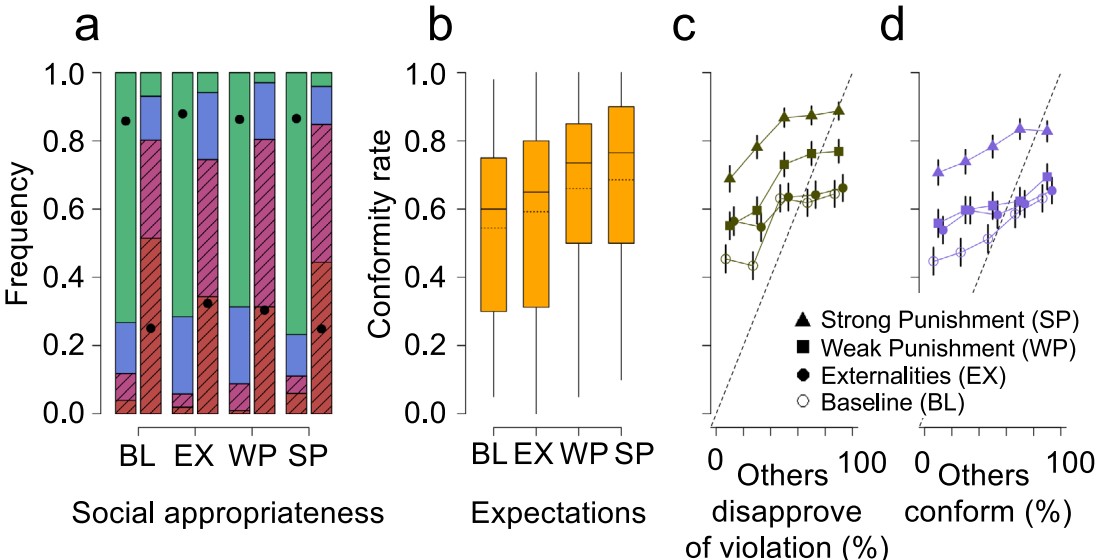

**Extended Data Fig. 4 | Normative beliefs, descriptive beliefs, and conditional rule-conformity in the presence of an externality, and weak and strong incentives.** The four nested treatments consist of a replication of the traffic light task without peers and no externality (Baseline, BL; replicating Fig. 1) and reported in Fig. 4, and three new treatments where a rule violation triggers an externality (a foregone donation of $1 to a charity, treatment Externality, EX) and where rule violations are punished with a probability of 10% (treatment Weak Punishment, WP) or 90% (treatment Strong Punishment, SP). Punishment implied loss of all earnings (except the flat participation fee). Panels show data across all four treatments **a** Normative beliefs $b^n$, that is, social appropriateness of rule-following and rule-breaking (total $n = 404$). Black dots are mean ratings. Pairs of bars show ratings for rule-following (lhs) and rule-breaking (rhs); legend as in Fig. 2a. **b** Descriptive beliefs $b^d$ of rule-following (total $n = 408$); legend as in Fig. 2b. (**c** and **d**) Average conditional rule-conformity, measured with the strategy method by eliciting people's rule-following or rule-breaking as a function of the percentage of people who **c** disapprove of a rule violation and **d** conform with the rule; bars show ±1 s.e.m (**c**: total $n = 607$; **d**: total $n = 619$).

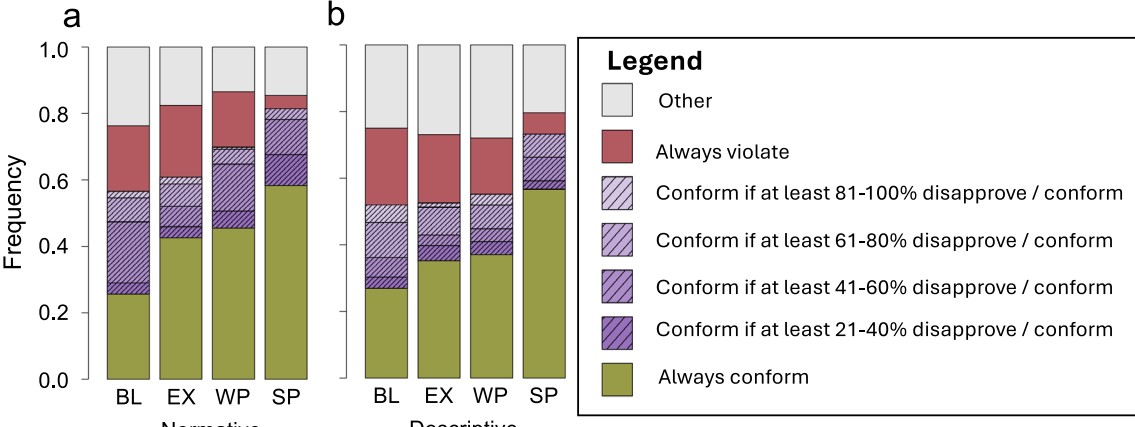

**Extended Data Fig. 5 | Individual profiles of rule-conformity $n_i(.)$ and $d_i(.)$ conditional on social expectations in the absence and presence of externalities and punishment.** (**a**: normative beliefs; **b**: descriptive beliefs). Stacked bars show the individual-level breakdown of Extended Data Fig. 4c,d. As in Extended Data Fig. 2, the grey bars on top of the stacked bars reflect individuals whose rule-following did not monotonically increase with the percentage of others disapproving of violations (for normative beliefs $b^n$; panel a) or following with the rule (descriptive beliefs $b^d$; panel b). The fraction of people who always conform (green bars) is also shown as dark-shaded part in Fig. 4 in the main text.

# Reporting Summary

## Statistics

For all statistical analyses, confirm that the following items are present in the figure legend, table legend, main text, or Methods section.

| n/a | Confirmed | |
|---|---|---|
| ☐ | ☒ | The exact sample size ($n$) for each experimental group/condition, given as a discrete number and unit of measurement |
| ☐ | ☒ | A statement on whether measurements were taken from distinct samples or whether the same sample was measured repeatedly |
| ☐ | ☒ | The statistical test(s) used AND whether they are one- or two-sided *Only common tests should be described solely by name; describe more complex techniques in the Methods section.* |
| ☐ | ☒ | A description of all covariates tested |
| ☐ | ☒ | A description of any assumptions or corrections, such as tests of normality and adjustment for multiple comparisons |
| ☐ | ☒ | A full description of the statistical parameters including central tendency (e.g. means) or other basic estimates (e.g. regression coefficient) AND variation (e.g. standard deviation) or associated estimates of uncertainty (e.g. confidence intervals) |
| ☐ | ☒ | For null hypothesis testing, the test statistic (e.g. $F$, $t$, $r$) with confidence intervals, effect sizes, degrees of freedom and $P$ value noted *Give P values as exact values whenever suitable.* |
| ☒ | ☐ | For Bayesian analysis, information on the choice of priors and Markov chain Monte Carlo settings |
| ☒ | ☐ | For hierarchical and complex designs, identification of the appropriate level for tests and full reporting of outcomes |
| ☒ | ☐ | Estimates of effect sizes (e.g. Cohen's $d$, Pearson's $r$), indicating how they were calculated |

*Our web collection on statistics for biologists contains articles on many of the points above.*

## Software and code

Policy information about availability of computer code

| | |
|---|---|
| Data collection | We used custom code of the open source software LIONESS Lab. All experiments are available in editable form via https://lioness-lab.org |
| Data analysis | Data and analysis code can be found on OSF (https://doi.org/10.17605/OSF.IO/7WZ4F) Data analysis software: R (version 4.2.0); R packages: lme4 (Bates et al 2015) and multcomp (Hothorn et al 2008). Citations are provided in the Supplementary Information. |

For manuscripts utilizing custom algorithms or software that are central to the research but not yet described in published literature, software must be made available to editors and reviewers. We strongly encourage code deposition in a community repository (e.g. GitHub). See the Nature Portfolio guidelines for submitting code & software for further information.

## Data

Policy information about availability of data

All manuscripts must include a data availability statement. This statement should provide the following information, where applicable:

- Accession codes, unique identifiers, or web links for publicly available datasets
- A description of any restrictions on data availability
- For clinical datasets or third party data, please ensure that the statement adheres to our policy

The data and analysis code are deposited at https://doi.org/10.17605/OSF.IO/7WZ4F

# Research involving human participants, their data, or biological material

Policy information about studies with human participants or human data. See also policy information about sex, gender (identity/presentation), and sexual orientation and race, ethnicity and racism.

| | |
|---|---|
| Reporting on sex and gender | We only asked for gender and 49% identified as male. Tables S1 and S2 in the Supplementary Information contain full details for each of our experiments. |
| Reporting on race, ethnicity, or other socially relevant groupings | We did not collect any race, ethnicity or other socially relevant grouping information; we only asked for age and gender. Details of distributions of participant age and gender are found in Tables S1 and Table S2 (Supplementary Information). |
| Population characteristics | A large majority of participants were volunteer Amazon Mechanical Turk US American workers; mean age 34.6 years; 49% male. Details for each experiment are reported in Tables S1 and S2 (Supplementary Information). In Experiment 1 we further show data from 103 UK student participants (mean age: 21.1; 46% male). |
| Recruitment | Participants were volunteers recruited on Amazon Mechanical Turk. Participation was restricted to US American residents of 18 years and older. Descriptions of the studies were kept vague to avoid any self-selection bias. |
| Ethics oversight | All studies reported in this paper were approved by the Research Ethics Committee, School of Economics, University of Nottingham (protocol ID 030_ERC_AP_MT). |

Note that full information on the approval of the study protocol must also be provided in the manuscript.

# Field-specific reporting

Please select the one below that is the best fit for your research. If you are not sure, read the appropriate sections before making your selection.

☐ Life sciences ☒ Behavioural & social sciences ☐ Ecological, evolutionary & environmental sciences

For a reference copy of the document with all sections, see [nature.com/documents/nr-reporting-summary-flat.pdf](http://nature.com/documents/nr-reporting-summary-flat.pdf)

# Behavioural & social sciences study design

All studies must disclose on these points even when the disclosure is negative.

| | |
|---|---|
| Study description | Data are quantitative data resulting from experiments we conducted using our custom-made software. |
| Research sample | A large majority of our study participants were US American volunteers recruited on the Amazon Mechanical Turk platform. They earned a flat fee for completing the study, and could earn an additional bonus payment dependent on their decisions (see Methods). We also report on data from 103 UK student participants who took part in the CeDEx lab at the University of Nottingham. They also earned a flat fee upon completion, plus a bonus that depended on their behaviour in the task (for details, see Section 1 of the Supplementary Information). |
| Sampling strategy | We used convenience sampling on Amazon Mechanical Turk to ensure we had sufficient participants from the same pool for our range of experiments. |
| Data collection | Data were collected online using a custom code of the software LIONESS Lab. Participants were recruited on the Amazon Mechanical Turk platform. Participants were anonymous and not known to the researcher (for details, see Section 1 of the Supplementary Information). The researcher was not blinded to experimental conditions and/or the study hypotheses. |
| Timing | Data for Experiments 1 - 3 were collected between 2014 and 2016; Experiments 4 were conducted in 2018. |
| Data exclusions | No data were excluded if participants completed the study. Incomplete submissions were discarded. |
| Non-participation | No participants declined participation. |
| Randomization | For Experiments 1 and 3 participants first completed the rule-following task on their own (with or without control questions; Experiment 1) and were then randomly assigned to one of the 28 treatments, which varied in the behaviour of "peers" (Experiment 3), For each of the tasks in Experiment 2, participants were randomly assigned to a treatment related to either normative or descriptive beliefs. In Experiments 4, participants were randomly assigned to one of the conditions BL, EX, WP, or SP described in the main text. |

# Reporting for specific materials, systems and methods

We require information from authors about some types of materials, experimental systems and methods used in many studies. Here, indicate whether each material, system or method listed is relevant to your study. If you are not sure if a list item applies to your research, read the appropriate section before selecting a response.

## Materials & experimental systems

| n/a | Involved in the study |
|---|---|
| ☒ | Antibodies |
| ☒ | Eukaryotic cell lines |
| ☒ | Palaeontology and archaeology |
| ☒ | Animals and other organisms |
| ☒ | Clinical data |
| ☒ | Dual use research of concern |
| ☒ | Plants |

## Methods

| n/a | Involved in the study |
|---|---|
| ☒ | ChIP-seq |
| ☒ | Flow cytometry |
| ☒ | MRI-based neuroimaging |

## Plants

| | |
|---|---|
| Seed stocks | *Report on the source of all seed stocks or other plant material used. If applicable, state the seed stock centre and catalogue number. If plant specimens were collected from the field, describe the collection location, date and sampling procedures.* |
| Novel plant genotypes | *Describe the methods by which all novel plant genotypes were produced. This includes those generated by transgenic approaches, gene editing, chemical/radiation-based mutagenesis and hybridization. For transgenic lines, describe the transformation method, the number of independent lines analyzed and the generation upon which experiments were performed. For gene-edited lines, describe the editor used, the endogenous sequence targeted for editing, the targeting guide RNA sequence (if applicable) and how the editor was applied.* |
| Authentication | *Describe any authentication procedures for each seed stock used or novel genotype generated. Describe any experiments used to assess the effect of a mutation and, where applicable, how potential secondary effects (e.g. second site T-DNA insertions, mosiacism, off-target gene editing) were examined.* |

