## [Peer Review File · Nature Human Behaviour]

Why people follow rules

Corresponding Author: Professor Simon Gaechter

Version 0:

Decision Letter:

12th January 2024

Dear Professor Gaechter,

Thank you for submitting your manuscript entitled "Why people follow rules". We have given the paper our careful consideration and find it of potential interest. However, we would like to invite you to revise your manuscript to address these concerns before we make a final determination on whether to send your manuscript for external review. In particular,

(1) Please report all replication studies in the same detail as the studies in the main text.

(2) Please inform whether any of the replication studies has been published. If so, please cite and discuss it/them in the main text.

We shall hope to receive your revised version as soon as you are able to complete the suggested revisions. If something similar is published in the interim we will have to consider the impact it has on the novelty of a revised manuscript.

If you anticipate a delay of more than four weeks, please let us know. We will be happy to consider your revision so long as nothing similar has been accepted for publication at Nature Human Behaviour or published elsewhere. Should your manuscript be substantially delayed without notifying us in advance and your article is eventually published, the received date may be that of the revised, not the original, version.

We are trying to improve the quality of methods and statistics reporting in our papers and to ensure compliance with editorial policies. To this end:

(A) The corresponding authors of all research manuscripts must complete an editorial policy checklist to ensure compliance with Nature Research editorial policies.

Editorial policy checklist:

<https://www.nature.com/documents/nr-editorial-policy-checklist.pdf>

(B) We want to ensure that the methods and statistics reporting in our papers is of the highest quality. To that end, we ask authors to fill out a reporting summary that collects information on research design and reporting.

Reporting summary:

(C) To help authors prepare their code for peer review, we ask them to follow Code Submission Guidelines (<https://www.nature.com/documents/GuidelinesCodePublication.pdf>) and to complete our Code and Software Submission Checklist.

Code and Software Submission Checklist:

<https://www.nature.com/documents/nr-software-policy.pdf>

Please complete the relevant forms when you submit the revised manuscript.

IMPORTANT: If you have opted for double-blind peer review, please do not include your name or any other identifying information within these forms.

Please note that these forms are dynamic 'smart pdfs' and must therefore be downloaded and completed in Adobe Reader.

We will then flatten them for ease of use by the reviewers. If you would like to reference the guidance text as you complete the template, please access these flattened versions at <http://www.nature.com/authors/policies/availability.html>.

Note that you are not required to revise your paper to include the information provided in the reporting summary. However, all points on the policy checklist must be addressed; please send us a new version of the manuscript with your completed checklist if needed.

Nature Human Behaviour is committed to improving transparency in authorship. As part of our efforts in this direction, we are now requesting that all authors identified as 'corresponding author' on published papers create and link their Open Researcher and Contributor Identifier (ORCID) with their account on the Manuscript Tracking System (MTS), prior to acceptance. ORCID helps the scientific community achieve unambiguous attribution of all scholarly contributions. You can create and link your ORCID from the home page of the MTS by clicking on 'Modify my Springer Nature account'. For more information please visit www.springernature.com/orcid.

If you are not interested in submitting a suitably revised manuscript in the future please let me know immediately so we can close your file. If you have any questions, please contact me.

Please use the link below when you are prepared to resubmit.
Link Redacted

Thank you for your interest in Nature Human Behaviour.

Sincerely,

Nature Human Behaviour

Version 1:

Decision Letter:

1st March 2024

Dear Professor Gaechter,

Thank you once again for your manuscript, entitled "Why people follow rules," and for your patience during the peer review process.

Your manuscript has now been evaluated by 4 reviewers, whose comments are included at the end of this letter. Although the reviewers find your work to be of interest, they also raise some important concerns. We are very interested in the possibility of publishing your study in Nature Human Behaviour, but would like to consider your response to these concerns in the form of a revised manuscript before we make a decision on publication.

To guide the scope of the revisions, the editors discuss the referee reports in detail within the team, with a view to (1) identifying key priorities that should be addressed in revision and (2) overruling referee requests that are deemed beyond the scope of the current study. We hope that you will find the prioritised set of referee points to be useful when revising your study. Please do not hesitate to get in touch if you would like to discuss these issues further.

In particular, our revision must address the following (as well as all other reviewer comments):

- (1) Reviewers 1, 2, and 3 suggest improving the literature review. Please situate your work in the extant literature and provide a concise yet clear overview on previous tasks and effects of rule-following research.
- (2) Reviewer 3 requests a broad comparison across all studies. Please carry out additional analysis and explain the relative importance of each factor on rule-following.
- (3) Reviewers 2, 3, and 4 raise questions for alternative explanations, including experimenter demand effect, self-control, and visibility of participants' own behaviour. Please carefully justify their influence on current results and discuss the implication.
- (4) Reviewers 3 and 4 have concerns on the statistical report and interpretation. In addition, it is journal policy that p-values larger than 0.05 must not be interpreted as evidence of an effect, unless you a priori specified an alternative threshold. Please improve the result reporting and interpretation accordingly.
- (5) Reviewers 3 and 4 have raised language issues. We do not require language quality from submissions as long as they are accurate and readable. However, please ensure that descriptions, including but not limited to concepts, research design, formulations, are accurately presented bearing the general audience in mind.

(6) Finally, in case that you need to remove some of the experiments for a more coherent story, we would encourage you to retain them in the Supplementary Information

In sum, we invite you to revise your manuscript taking into account all reviewer and editor comments. We are committed to providing a fair and constructive peer-review process. Do not hesitate to contact us if there are specific requests from the reviewers that you believe are technically impossible or unlikely to yield a meaningful outcome.

We hope to receive your revised manuscript within two months. I would be grateful if you could contact us as soon as possible if you foresee difficulties with meeting this target resubmission date.

- Include a "Response to the editors and reviewers" document detailing, point-by-point, how you addressed each editor and referee comment. If no action was taken to address a point, you must provide a compelling argument. When formatting this document, please respond to each reviewer comment individually, including the full text of the reviewer comment verbatim followed by your response to the individual point. This response will be used by the editors to evaluate your revision and sent back to the reviewers along with the revised manuscript.
- Highlight all changes made to your manuscript or provide us with a version that tracks changes.

Link Redacted

We look forward to seeing the revised manuscript and thank you for the opportunity to review your work. Please do not hesitate to contact me if you have any questions or would like to discuss these revisions further.

Sincerely,

Nature Human Behaviour

Reviewer expertise:

Reviewer #1: norms, pro-sociality, experimental economics

Reviewer #2: economics of corruption, behavioural economics

Reviewer #3: rule-following, norms, economics, psychology

Reviewer #4: norms, decision-making, psychology

REVIEWER COMMENTS:

Reviewer #1:

Remarks to the Author:

The authors build upon work by Kimbrough and Vostroknutov that introduced an incentivized measure of individual rule-following, and they perform a huge, systematic analysis (with over 14,000 respondents) of the factors influencing individual rule-following. They find that, although rule-following is sensitive to incentives, externalities, and information about both injunctive and descriptive norms, it is to a large extent reflective of an intrinsic motivation.

The amount of work undertaken and the authors' attention to detail are both commendable. The theoretical framework the

authors introduce is clean and neatly frames the research question; the experimental design is careful; the results are presented clearly with a series of elegant figures that tell the whole story in pictures; the main finding is striking and important, given the connections between rule-following and many aspects of social behavior.

My main suggestions are about other related work that ought to be cited:

- 1) Kimbrough, E.O. and Vostroknutov, A., 2018. A portable method of eliciting respect for social norms. *Economics Letters*, 168, pp.147-150. -- This paper introduces another abstract variant analogous to the original stop-light task and shows that similar patterns of rule-following are observed across university samples from 5 countries. In this task subjects are given a number of balls and told that the rule is to put them into the blue bucket; there is also a yellow bucket, and putting the balls in the yellow bucket earns them more money, so the tradeoff is again between self-interest and adherence to the rule.
- 2) Tate, C., Kumar, R., Murray, J.M., Sanchez-Franco, S., Sarmiento, O.L., Montgomery, S.C., Zhou, H., Ramalingam, A., Krupka, E., Kimbrough, E. and Kee, F., 2022. The personality and cognitive traits associated with adolescents' sensitivity to social norms. *Scientific Reports*, 12(1), p.15247. -- This paper uses a sample of ~1500 adolescents who completed the task in (1) and assesses the extent to which their behavior is related to a wide variety of personality and cognitive traits.
- 3) McBride, M. and Ridinger, G., 2021. Beliefs also make social-norm preferences social. *Journal of Economic Behavior & Organization*, 191, pp.765-784. -- This paper offers additional evidence that rule-following is, to some extent, conditional on beliefs about others' rule-following.
- 4) Ridinger, G., 2020. Shame and theory-of-mind predicts rule-following behavior. *Games*, 11(3), p.36. -- This paper shows that shame and rule-following are correlated in a university sample.
- 5) Abbink, K., Gangadharan, L., Handfield, T. and Thrasher, J., 2017. Peer punishment promotes enforcement of bad social norms. *Nature communications*, 8(1), p.609. -- This paper offers additional evidence for "bad norms", which you might cite on p. 14.
- 6) Gross, J., Emmerling, F., Vostroknutov, A. and Sack, A.T., 2018. Manipulation of pro-sociality and rule-following with non-invasive brain stimulation. *Scientific Reports*, 8(1), p.1827. - This paper offers neuroeconomic evidence that rule-following is partly governed by the right lateral prefrontal cortex, using transcranial magnetic stimulation to increase rule-following.
- 7) Kimbrough, E.O. and Wilson, B.J., 2022. Rule-following. In *The Routledge Handbook of Philosophy, Politics, and Economics* (pp. 108-120). Routledge. -- This chapter offers summaries of theories from economics, psychology, and philosophy regarding why people follow rules (and norms).
- 8) Gross, J. and Vostroknutov, A., 2022. Why do people follow social norms?. *Current Opinion in Psychology*, 44, pp.1-6. -- The authors consider theories of why people follow norms, which is relevant since many authors (perhaps sloppily) use the words rule and norm roughly interchangeably. They consider learning and internalization, social image and self-image, and conditional preferences, some of which overlap with the explanations tested herein.

In the interest of transparency, I'm signing my report. Thanks for an excellent paper. It's one I read wishing I had written it myself.

Sincerely,
Erik Kimbrough

Reviewer #2:

Remarks to the Author:

The authors conduct online experiments with a large sample of 14,000 participants to discover a surprising behaviour: Even without externalities and without punishment, participants follow rules that are costly to themselves. To do this, the authors invent a novel traffic light task. They elicit beliefs and conduct a wide variety of treatments to disentangle the contributions of prescriptive and descriptive norms, as well as the additional contribution of externalities and punishment, which may merely piggyback on intrinsic compliance. Overall, they provide compelling data to infer an intrinsic respect for arbitrary rules that survives even in dismal environments with many rule violators and poor prescriptive norms. Overall, I would like to see this published. At the same time, the current study needs substantial editing and a reworking of the punchline. In addition, some experiments can be omitted to make room for more intuitive explanations.

1. I would appreciate a bit more motivation at the start. In particular, the authors refer to "obedience or an intrinsic respect for a rule because a rule is a 'deontic constraint' that places an interior demand on how a person should behave... because [persons] respect the prescriptive or proscriptive demand a rule places on them regardless of the social expectations they hold". I am very sympathetic to this statement, but I think it deserves a little more elaboration, supported by examples or anecdotes. The reader should be involved in this deontic constraint and be drawn into the thought. Sometimes we observe the opposite, that rules are broken irrationally (children defying reasonable parental rules; subordinates denying the authority of superiors by doing the opposite of what they have been told to do). So where does the intrinsic tendency to follow rules fit in?

2. The intrinsic respect for rules could be seen as a kind of experimenter demand effect. I suspect that the two are not easily disentangled. I think the authors should address this and provide an argument for the relationship between the two, for example, that what is commonly referred to as the experimenter demand effect might just be a variant of intrinsic respect for rules.

3. Page 4: "rule-following participants lose at least 60% of their MU." At this stage, readers should be better informed about the instructions, i.e. that the green light appeared after 12 seconds in all runs and treatments. This also raises the question of whether participants may have formed biased beliefs about the cost of obedience. They might have expected the green light to appear earlier and would not have been willing to sacrifice 60% of their income. The authors should address this objection. Does the repetition of the game described later contradict such biased beliefs?

4. The description of the experimental design in "Measuring social expectations of rule-following" is difficult to read. Do these subjects play the same traffic light game as before (participants) or do they just observe others playing the game (spectators)? I have read the first paragraph several times and am still confused. I tried to understand this by reading S.1.4, which should not be necessary for the lay reader. Here we are informed that "after reading the instructions that this decision-maker would receive (these were the same as the instructions given to participants in the traffic light task) participants were asked to evaluate two possible behaviours". The confusion could be avoided by using "participants" only for those playing the traffic light game and "spectators" for the others. My confusion continues with the information about "four between-subjects diagnostic tasks". There are two tasks that elicit normative beliefs and two tasks that measure people's conditional willingness to follow rules, right? Do I understand correctly that the latter two tasks no longer refer to spectators, but to participants who are given information previously obtained from spectators? These participants (are they the same as in the first experiment?) are then shown five different scenarios which vary beliefs. I am not convinced by the use of the strategy method here, because social norms then seem rather hypothetical, rather "cold". The authors may wish to defend their approach or concede that 'hot' descriptions of social norms might have been more influential. That said, there is an excess of numbers in the description, and the authors should perhaps focus on the essential data.

5. Page 11: "Descriptive beliefs b " may also influence normative beliefs b ! if observing peers breaking the rule leads people to revise their normative beliefs." This appears to be a finding without context. Isn't that somewhat talking to a different literature? In my view this part may be considered for deletion.

6. I do not think that the conclusions summarise the results well and speak to the cited literature. For example, the authors state that "the conditionality on social expectations can also lead to contagious rule violations: the asymmetry we observe (Fig. 3a) suggests that fragility is more likely than stability". This is not in the spirit of the previous explanations, which show a surprisingly high degree of intrinsic rule compliance. It repeats what Bicchieri and many others have done before. Rather, the finding of this study should be that there is a substantial bias towards compliance that serves as a basis for conformity even in dismal environments. Another conclusion states: "Another hypothesis is that societies organised around kinship may have developed a weaker sense of obligations to follow rules in interactions with strangers. Our study offers a framework to address these important open questions". It would be interesting to investigate this, for example by observing whether compliance depends on whether the rules are issued by friend or foe, kin or stranger. But I do not see that the study provides a framework for understanding such questions.

My minor points:

1. Page 5 refers to figure S1. I suggest moving S1 to the main text. It took me quite long to understand the main text without this figure, which well shows the 12 seconds condition. Also, it separates between the two variants with and without the control questions. It then also explains how the two conditions add up to the 7732 participants. Otherwise, the calculations in the main text that refer to 0.66 (later referred to as 0.656) are hard to follow. Readers want to see the results being separated with and without control questions anyways.

2. Page 5: "These results suggest robustness of rule-following rates across platforms holding cultural background constant." This confusing. Is the cultural background across the UK, USA and Sweden really constant?

3. Page 9: "In the other twenty-seven treatments, participants in the second iteration could see on their screen, in addition to their own circle, other circles programmed to simultaneously display the movements." The task runs across 20 seconds. How did peers violate the rule? Dynamically such that one-by-one can be observed staying or crossing the red light or statically with only a single number being reported after 12 seconds? Please explain in the main text!

4. Fig 3 explains: "Dots show rule-following rates (with percentage point changes from the no-peers baseline rule-conformity rate of 77%; horizontal line) in iteration 2 in treatments with and without control questions." This sentence took me a while to understand. Consider deleting the numbers and show only the location of the dots. The figure will be easier to read without the numbers, which do not reveal a substantial value added.

5. Page 10: "Participants in all treatments completed a third and final iteration of the rule-following task without peers." The second iteration was described as taking place after observing the behavior of peers (not simultaneously). This suggests that the third iteration just repeats the second. If that is so, what is the value added of the third iteration?

6. Page 13: "The dark-shaded parts of Fig. 4 show the average fraction of people who conformed unconditionally." I am confused by "unconditionally". Did participants play the original traffic light game or the modified one with strategy method? I assume it was the latter but readers will want to know this upfront.

7. Page 13: "The presence of externalities (in EX) and weak and strong incentives (WP and SP) not only increased overall conformity but also the rate of participants who did not condition their rule-following on social expectations." Do we see this finding in the figure?

8. The study should more clearly be organized by reference to Experiment 1, Experiment 2, Experiment 3 ... to better guide readers.

Overall, the authors might consider deleting some of the experiments they designed and focusing more on others. For example, the experiments on "Measuring social expectations of rule-following" (Figure 2) with the four diagnostic tasks could be considered for deletion because the following experiments capture descriptive norms in a much better way. This would leave out prescriptive norms, which might be a price worth paying. Figure 3b also seems to speak to another literature. Whether descriptive beliefs influence normative beliefs is not the point of the study. The study addresses important issues that will add value to readers of *Nature Human Behaviour*, and once the issues are addressed, it can be a pleasurable read.

Reviewer #3:

Remarks to the Author:

In the present paper, the authors employ a simple rule-following task to investigate how various factors influence rule

abidance. The amount of data and the number of different studies are quite impressive, and the design of the various experiments and manipulations is creative and very clear in dissecting different elements behind rule abidance (intrinsic, normative expectations, observations of peer behavior, externalities, etc.). I genuinely enjoyed reading the paper. It happened to me multiple times that I thought, "But what about this?" only to then learn that the authors addressed this point just a couple of paragraphs further.

While none of the proposed and investigated reasons for rule abidance are novel, the work is outstanding in that it takes all of the prominent reasons into account and develops a clever general experimental framework that is employed to investigate each of the reasons in isolation and in comparison to each other. In my assessment, that is a significant advance in the field of norm- and rule-following, and the methodological innovations and empirical findings are insightful and, I am sure, will motivate a lot of follow-up work. I do not see any major flaws that should prohibit publication. However, I do have some questions and remarks that should be addressed, outlined in more detail below.

(1) There are numerous proposed reasons why people abide by or violate rules (or norms), as the authors elegantly outline in their introduction. Developing a general experimental framework to pit these reasons against each other is, therefore, very valuable. After reading the introduction, I, to some degree, expected that the authors would use their framework (the rule-following task and all its variations) to perform a 'horse-race', pitting the different explanations and their explanatory power against each other. While, to some degree, that is the case (e.g., Fig. 4), I was left wondering; What is now the most powerful predictor of rule-abidance? Playing the devil's advocate, one conclusion of the paper is that everything (expectations, intrinsic motivation, punishment, externalities, etc.) matters, which is (while very valuable to show that within one framework) not that groundbreaking. As such, I would like to push the authors a bit on whether a broad comparison across all studies would be possible. I am not sure if the structure of the experiments would allow that (since, from my understanding, the round structure of experiments also differed and what was tested). But if possible, it would be very informative to learn which factors promote (or reduce) rule-abidance the most. For example, just having comprehension questions seems to change rule-abidance quite a lot, but in the present manuscript, it is difficult to assess how that compares to, e.g., a 90% probability of sanctions. Having one figure that shows the effect sizes of all proposed mechanisms would be very helpful (maybe even including relevant self-reported personality characteristics like patience). I understand that there could also be interactions among mechanisms and that, again, the structure of experiments also varied, which could make that complicated. But it seems an essential advantage (that is not fully taken advantage of) of having one game and then manipulating all kinds of rules and information is the ability to quantitatively compare the different factors and their relative importance. Or even go beyond that and propose a unified utility model with parameters that can be estimated from the observations.

(2) The implemented task (traffic light or neutral frame) requires people to wait. As far as I understand, people also did not know how long they had to wait. This introduces an element of patience, such that other prominent psychological constructs could play a role here (i.e., patience, self-control, impulsivity). And indeed, the authors also find a correlation with self-reported patience (Table S3) and what appears to be dynamically inconsistent preferences: Many people decide to wait a bit before violating the rule rather than violating the rule from the start (Fig S1).

Since the game is kept constant across all treatments, this feature does not invalidate the results and comparisons across treatments. Still, this seems to be an important element (i.e., self-control capacity or whatever term one wants to use) that is not necessarily linked to norm/rule-following (and the investigated determinants in the paper), as many rules are arguably followed out of (internalized) habits and do not require self-control. Relatedly, another version of the original traffic light task – the ball-division task – in which people do not have to wait but have to divide balls between two buckets according to a rule gets rid of this confound with self-control/patience, etc. Again, I do not think that this challenges the validity of the results that rest on comparisons within the same task. But it remains a somewhat open question what role patience or self-control plays over and beyond the discussed reasons for rule abidance. This could be discussed more as a potential caveat/future direction.

As a side note, there are more studies than cited that used the ball-division task and investigated rule abidance (on the individual or group level). For readers, I think it would be nice to have a more complete reference list of previous work on this task to get an overview of previous work and how different factors change rule abidance (i.e., including studies using the ball-division task that is simply a variant of the traffic light task).

(3) I was wondering whether the rather high base rate of rule-abidance (e.g., Fig. 1) could be (partly) explained by, albeit, wrong expectations of punishment for violations. People may (wrongly) assume that not following the rule would lead to some negative consequences (e.g., a bad approval rating for AMT participants). In die-rolling or other cheating tasks, people often get the opportunity to violate rules in private such that neither the experimenter nor the computer can 'prove' that someone cheated (i.e., violated the rule of honesty). Arguably, such ambiguity may also play a role for rule-following outside of the lab, but I was wondering whether the authors have any data that could corroborate or debunk this reasoning (e.g., a post-experiment questionnaire). Maybe it could be discussed for follow-up work that could manipulate to which degree rule-violations are recorded/observed/can be detected only ambiguously (similar to cheating tasks).

(4) On page 10, the authors write: "Despite contagious rule violations, overall rule-conformity rates are still remarkably high in absolute terms: even when six peers violated the rule, 55% of participants still followed the rule. Focusing only on conditions with control questions, rule-following never dropped below 40%, even when six out of six peers violated the rule". Here, I asked myself: What would happen if the game is repeatedly played, and people repeatedly observe high (or low) rule abidance by others? Similar to the public goods game literature, I would expect that social dynamics actually need some time (or, in other words, repetition) to unfold. Also in reality, this may play an important role. When a person moves from a city with a low frequency of jaywalking to a city where this is very common (let's say New York), I would also expect that people first stick to their "internal" or learned inclination but would eventually adapt to others. The study considers many interesting

factors. The one thing I felt was missing, however, was contagiousness across multiple rounds. Maybe this could be at least discussed, especially when the danger is that readers take the above statement at face value and wrongly generalize (peer influence may be much, much stronger, but it may take more than one round).

Minor:

(5) I hope I did not miss it, but how strongly were normative beliefs (Fig 2a) and descriptive beliefs (Fig 2b) correlated on the individual level? Maybe that would be interesting to report.

(6) On page 8, the authors report that “19% of people conformed with the rule regardless of how many others disapproved of a violation; 29% never conformed; 30% followed after a threshold; and 22% had inconsistent responses (Fig. S2a).” and “The individual functions reveal that 20% always conform; 34% never conform; 31% conform conditionally; and 15% show no clear threshold effect (Fig. S2b).”

Looking at Figure S2, is that correctly coded? It looks like 29% and 34% always conform (i.e., the reverse).

(7) Page 8: The authors write “The results of Figs. 2c and 2d suggest a striking degree of conditional conformity in our asocial and anonymous setting.” A “striking degree” is arguably subjective, but I was not sure whether I would agree here with the strong word “striking”. Instead, it looks like many people are weakly influenced by social information (as mentioned in the following sentences).

(8) On page 9, the authors write: “To measure the causal effect of observing others’ rule-following and rule-breaking on one’s own rule-following, we designed a modified version of our traffic light task in which participants observed how other participants had previously behaved in the task.” Maybe I misunderstand something but why is that a causal test compared to the manipulation before in which information was given about the % of conformists (experiment underlying Fig. 2d)? The experiment in which “peer” behavior can be directly observed is arguably a more elegant setup, but does it really add something conceptually over and beyond the previous study (e.g., causal interpretation that was not possible in the previous experiment as the sentence seems to suggest)?

(9) “We did not observe a group size effect in the contagiousness of rule violations: rule compliance appears to be driven by the proportion rather than the absolute number of violators”. Here, I was wondering if this interpretation is actually following the regression results. If not mistaken, this refers to Table S6. In the first column, it looks like number of peers (absolute number) as well as fraction of rule-followers (proportion) significantly predict rule-abidance.

(10) From my understanding, violating the rule reduced expected payoff in the SP treatment (see Fig. 4). Given the high base-rate of rule abidance to begin with, I would then expect abidance rates close to 100% in this case. Is there any indication to which degree participants misunderstood the task or any reasons that the authors can think of why rule abidance is so low in this case?

(11) Based on information in the SI (page 6), it suggests that authors fitted (multilevel) linear regressions, even though their dependent variable is binary (?). The authors argue that results are easier to interpret. Yet, logistic regressions not only seem to be more appropriate but coefficients can also be quite nicely interpreted by calculating changes in odds (from baseline). They can also be directly interpreted as effect-sizes which seems a bit more difficult for linear models fitted to binary data. Hence, I was really wondering why that choice was made, as the provided reasoning was not completely convincing to me.

Reviewer #4:

Remarks to the Author:

Review of NATHUMBEHAV-23124337A: “Why people follow rules”

Thank you for giving me a chance to review this manuscript.

The authors present a series of studies examining people’s tendency to follow arbitrary rules, varying elements of the situation such as externalities, incentives, abstraction, and descriptive beliefs. Experiments show the majority of participants follow rules, even at a personal cost and with no generated benefit. The experiment also provide insight into the psychological dynamics of rule-following and the importance of norms and social learning related to norms. The authors discuss implications for systems built upon rules and rule conformity.

From the start, I would like to emphasize that I have a very positive opinion about the submitted manuscript. This paper addresses its central question with a good number of well-designed variations and well-proportioned sample sizes. Its results tell a consistent story and provide answers to relevant open questions. In fact, I have no major points (nor major concerns) to hold against this manuscript.

Below, I will still provide a list of points that if addressed could potentially improve the manuscript. I will split these into major minor points (where I personally see a good argument to address them) and minor minor points that consist of smaller self-evident issues. I conclude with some suggestions. Even though this list is not short, I would like to reinforce that this is all intended to support a manuscript that I consider a valuable contribution to the literature. I enjoyed learning about the sequence of experiments and agreed with their interpretation. Many of the points below are concerned with readability from my perspective with a clear understanding that the authors may have good arguments not to follow at least some of these ideas.

MAJOR MINOR POINTS

- 1) Introduction: The first paragraph might benefit from some more detail and full explanations of concept as opposed to merely citing central theoretical terms in scare quotes.
- 2) Use of variables: I am concerned about the use of formal variables to denote the elements of the model. On the positive side, this allows to formulate the proposed model as $C=f[I,E,X,B]$. On the negative side:
 - a. Little is learned from this formulation alone, it does not provide guidance on directionality, B is later split into b^n and b^d .
 - b. Whenever elements of this formula are used later on, they are accompanied by an explanation, making the letters redundant in the remainder of the text
 - c. When these variables are introduced, their domains and dimensionality is not specified. "Extrinsic motives" do not seem to be a unidimensional variable, for example. Thus, it is puzzling to find inequalities later on, such as $E<0$ (re-labeled as extrinsic incentives on p.4). In fact, this seems to mix up motives themselves and situational motivation as a sum of the activation of different motives [see my point about the causal model, as well]. Motives and incentives are clearly distinguished at least in psychology. On page 5, "I" is referenced as "intrinsic reasons", which again is a different concept.
 - d. The formalization might appeal to economists and some analytical philosophers (who would have separate concerns, though), but I question its value for a more general audience, when there is no clear added value provided. "Negative externalities" seems easier to comprehend than $X<0$.
 - e. The letters are not helpful to distinguish the concepts: extrinsic motives and externalities could be abbreviated as either "E" or "X", so could social expectations be (that are denoted as B for beliefs). Using the names of the concepts would thus be more helpful for the reader
 - f. The function names " $n(b^n)$ " and " $d(b^d)$ " are likewise not very evocative: they make sense in the context of the strategy method, but I am not sure they ease understanding.
 - g. " $E<0$ " is used to indicate that rule-following is costly for the person, but " $X<0$ " seems to describe not the consequences of rule-violations, rather than rule-following (this led to a state where I was confused what " $E<0$ " and " $E>0$ " referred to, and I would have expected " $X>0$ ", as the charity relatively benefits from following the rule). Explaining it in natural language would not sacrifice any precision and help the reader.
- 3) Pp.3/4: The task should be clearly described before discussing variations or relations to other tasks. Potentially, this could be done in a separate figure, showing starting position and intermediate steps. There are also some open questions: How is the circle moved by participants? Is the movement binary (one mouseclick covers the entire range) or continuous and incremental? I got enough information to evaluate the task (and the supplementary material addresses these questions), but it might still be helpful to have a clearer introduction in the main manuscript. Fig. 1 does not show the task but a screenshot of the task.
- 4) Conditions and variations are introduced consecutively and there is little structure given at the beginning. I would strongly suggest numbering the Studies and subconditions and to start the discussion with a table detailing the presented experiments and variations. Ideally this table could also offer details of variations (e.g., use one column labeled "extrinsic incentives" and another one labeled "externalities" with a brief description of their role in the given variation. I believe I was still able to get a clear idea about each study, but especially at the beginning I would have appreciated this guidance.
- 5) I appreciated all of the variations chosen in this design, but I believe that their merits could have been explained more explicitly in at least one case. Which potential weaknesses are (successfully) addressed by the variations? For example, I found the abstract variation of the task very important, as it addresses the potentially inherent moral coding of a traffic light. Knowing a very well-grounded rule about stopping at red lights might create social expectations about the traffic light outside of the traffic context. Results can be interpreted in this direction, but the change is relatively small. In general, the robust collection of findings across contexts and variants corroborates the chosen interpretation.
- 6) The purpose of control questions with crowdsourced participants goes beyond increasing the salience of incentives and constitutes a form of attention check. Data quality issues are appropriately discussed in the supplementary material. Again, robustness checks show that results are not based on inattention, but this concern could be explicitly named.
- 7) P.5: A correlation with being "patient" might not indicate a "general tendency" as much as a specific results for a feature of the paradigm: not moving requires some impulse control.
- 8) Possible alternative reasons for following rules in experiments is the potential that participants want to "help science", not forfeit future employment opportunities (very relevant for MTurk samples), accept the authority of the experimenter and want to demonstrate they can follow instructions. These explanations---while consistent with rule-following--- are more specific to experimental settings, so I wonder whether this should be at least be considered as a potential limitation (obviously difficult to address in experiments and not what I would expect to happen in this already extensive manuscript).
- 9) P.10: Again, I found the inclusion of the "third and final iteration" extremely reassuring. It directly addresses a concern I had about showing the movement of other participants: As this task seems to involve some response inhibition, seeing movement might therefore trigger responses. Seeing that the effect persists is thus important. This might be pointed out.
- 10) I stumbled across some phrasing issues that might be due to disciplinary preferences. To be clear, I believe that the studies provide highly interesting and useful evidence for learning about rule-following behavior. My question here was

rather to which extent the “psychology of rule-following” is fully captured in the theoretical model and whether this would be even desirable. From my observation, especially economists like to discuss behavior as “caused” by beliefs and preferences. To some degree, this entails the danger (if taken too literally and too far) of short-circuiting psychology and reducing decision making and action to a stimulus-response chain (and ignore agency, individual differences, and subjectivity). On balance, I had the impression that the theoretical level is well-chosen in this manuscript, I was concerned about a few phrases, though.

- a. “Observing peer behavior provides descriptive beliefs” p.8 --- I fully understand what is meant here, but it sounds a bit mechanistic.
- b. “causal role of descriptive beliefs” [p.9] --- In which sense are they causal? They can be a basis for justifying one’s action, for choosing a consideration set of actions, etc. I would argue that a full “psychology of rule-following (p.14)” would include the full picture of how this information is utilized, made sense of and integrated based on motives, motivation, personality, etc. This full picture, on the other hand, is unlikely to be as relevant for the theoretical aims pursued in this manuscript, so I would not call it a deficit at all if there is no full account of a “psychology of rule-following” here. Personally, I would tone down these terms and rather discuss regularities of rule-following or “psychological aspects/dynamics of rule-following” or some other similar term.
- c. “basic proximate psychological processes”: In a similar vein, the psychological perspective is reduced to normative and descriptive beliefs and preferences to follow rules, but the processes described are not necessarily “psychological” in the sense of full process model (that I would not expect to find fully fleshed out in an empirical paper).
Again, these points are not intended to criticize the studies, hypotheses, framework or main interpretations. I also fully understand the intention to connect findings to multiple literatures across disciplines.

MINOR MINOR POINTS

- 1) Abstract: The second sentence of the abstract is quite dense and could use some unpacking: What exactly is the relationship between respect for rules and the factors accounted for? Why would that accounting lead to a dispute?
- 2) Abstract: There is a recurring use of the term “their engendered social expectations”. I have trouble parsing this: What is meant (I believe) is that rules create social expectations in people, but the possessive pronoun referring to rules makes no sense. Active voice (and possibly a different verb) might address this.
- 3) Abstract: “piggyback” seems avoidable
- 4) P.1: “people with other-regarding concerns”: “prosocial” might distinguish this better from those who want to minimize others’ outcomes; “social preferences” include rivalrous motives
- 5) P.2: last sentence: check use of semi-cola
- 6) P.3: definition of intrinsic respect (end of first paragraph): The given definition would be fulfilled if someone is paid to follow a rule (by an algorithm). There would be no social expectations in this case, but this would not constitute an “intrinsic” respect in any meaningful sense (this makes sense in the context of the experimental variation later on, but not as a general definition).
- 7) P.4 suggested edit: unlike in PREVIOUS tasks that have been used...
- 8) Mturk participants are not anonymous, but pseudonymous (in some cases identities have been compromised through security leaks).
- 9) P.5: “an evolved capacity of mind-reading” needs some unpacking
- 10) P.7: caption b: switch in order between “mean: black dot” and “box: quartiles”; caption “c and d”[?] starts very dense (and in a nominalist style), I wonder whether the results at the end are ideally placed there [three digits for coefficients?]
- 11) P.7: “when almost everyone disapproves”: the sentence is correct, but there is a potential that “almost everyone” is connected to 56% (maybe: “...increases from 0% to 100%, normative conditional conformity increases substantially, from X% to 56%.)”
- 12) P.8: “19%” at the start of a sentence
- 13) P.8: “Descriptive conditional rule-conformity” could be unpacked and explained (it is not entirely clear from the sentence alone what “descriptive” refers to [there is nothing wrong in this sentence, it is just difficult to parse]).
- 14) Fig.3: I am a bit unsure about the use of “angels” and “demons” here; it brings in a moral evaluation that flies in the face of calling the rule “arbitrary”. In addition, there is no legend identifying the icons as “angels” and “demons”. I wonder about a possible x-axis label for the diagrams in subfigure a [I actually counted them to see that these were the 27 mentioned conditions]. The horizontal anchor lines represent the baseline condition, and it could have been easier to see if it had been labeled (maybe on the right side of all diagrams). Otherwise, I like this figure panel, as it is a great way of presenting the study results.

- 15) P.10: It might be useful to explain the "bad apple effect": What is the evidence for it and does this go beyond tautologies?
- 16) P.10: "small but statistically insignificant": Which type of test is the basis for this statement? It seems that there is a consistent pattern for all 6 conditions with fully conforming peers. Is this based on six individual tests? [This is far from crucial, as the pattern across conditions and within conditions is very robust].

SUGGESTIONS

1) P.5: When putting results in perspective, it might be mentioned that the rate is LOWER than the rate people give up when offered the opportunity to cheat (usually 75% of potential gains are forfeited for the benefit of being honest). Cheating is a form of not following rules. Similarly, rule-following increases with explicit promises given (even without sanctions). It might be interesting to compare and contrast the paradigm with those featuring cheating opportunities and/or promises.

2) The use of a reputation score of 90% on Mturk is probably no longer useful. In 2012, a threshold of 95% was suggested, but since it has been discovered in some studies that 99% of participants have scores higher than a 98%. After the "Mturk crisis" in 2018, when the platform encountered an influx of participants with low proficiency in English, the use of VPS/VPN-checks has been suggested as a better control method. An analysis of IP addresses is discussed in the supplement that shows corroborating evidence. Pointing at some of these discussions might be helpful in the supplement.

3) One interesting variant that would follow from the formalization would of course be an " $X < 0$ " condition, in which rule-following causes harm to others. I do not suggest to even consider this as an addition to the present manuscript, but it could be a discussion point.

Note that there is no attachment for this review.

Version 2:

Decision Letter:

Our ref: NATHUMBEHAV-23124337B

25th September 2024

Dear Dr. Gaechter,

Thank you for submitting your revised manuscript "Why people follow rules" (NATHUMBEHAV-23124337B). It has now been seen by the original referees and their comments are below. As you can see, the reviewers find that the paper has improved in revision. We will therefore be happy in principle to publish it in Nature Human Behaviour, pending minor revisions to satisfy the referees' final requests and to comply with our editorial and formatting guidelines.

We are now performing detailed checks on your paper and will send you a checklist detailing our editorial and formatting requirements within two weeks. Please do not upload the final materials and make any revisions until you receive this additional information from us.

Sincerely,

Nature Human Behaviour

Reviewer #1 (Remarks to the Author):

As I said in my original report, this is a major undertaking that deserves to be published. I am somewhat disappointed by the obliqueness of the references to prior work.

A great many of the findings reported in this paper have been put together in separate papers by others, using variants of the research design for measuring rule-following that Vostroknutov and I developed in our 2015, 2016 and 2018 papers. For example, conditionality of rule-following and its dependence on beliefs about others have been covered by Desmet and Engel (in a slider task rule-following task) and using our task directly by McBride and Ridinger. McBride and Ridinger also use the strategy method in one treatment to elicit conditional rule-following and employ a "descriptive norm" manipulation to shock beliefs in another. Similarly, associations with ToM were shown by Ridinger. In my view, the value of the paper comes from a) the conceptual framework, b) putting all these pieces together in a single omnibus study, and c) collecting large samples.

The paper cites most of the relevant work, but even our papers are largely referred to as "related work" rather than emphasizing that they are the very soil out of which this paper has grown. That this is the case is pretty transparent, if you know the literature, but to someone for whom this is new, it wouldn't be so obvious in the current writeup. The authors copy the traffic light setup from our paper (though theirs is simplified), and they directly use the language from our instructions "The rule is to wait at the light until it turns green." Moreover, the quote they use from Adam Smith is actually in the epigraph to our 2016 paper (and has been there since our original 2011 working paper). It's an apt quote - that's why we used it, but with the other dependencies here it is awkward. Perhaps given the timelines, there is some claim to priority with respect to these other papers (I don't know), but it is clear that our paper motivated this work. This should be transparent from the start.

In addition, a number of the concerns about the interpretation of the task raised by reviewers have already been addressed with theory and/or data in our papers.

1) We address concerns about experimenter demand explicitly in our 2016 paper, "Another way to view obedience in our RF task is as a pure "experimenter demand" effect. Under that interpretation, we are using demand effect sensitivity as a proxy for phi. This has the nice feature that a long-time bogeyman of experimenters turns out to be an ally. We are sympathetic to this view, but we would argue that any experimenter demand effect is actually a manifestation of the norm-dependence we seek to measure, else why should individuals be concerned about the demands of the experimenter?" (see fn 17) This is quite similar to the argument made by the authors in the conclusion. They might say "As pointed out by KV2016..."

2) We explicitly address concerns about the familiarity of the stop-light environment and the fact that there are pre-existing traffic rules which might be applied in this context with a "no rule" treatment (see Fig 2 of the 2016 paper). Subjects complete the same task with the same incentives, but we exclude "The rule is..." from the instructions. There is still some residual rule-following, though less than in the original task. This is a source of bias in the traffic task that caused us to design the ball-and-bucket version for subsequent papers. Since that source of bias is present in all treatments in this paper, it isn't a major problem for identifying treatment effects, but it may influence the reported levels.

3) We checked whether rule-following was associated with "respect for authority" as measured by the Moral Foundations Questionnaire (see Appendix E1 of the 2016 paper); it is not. We confirmed this also in table F2 of Kimbrough and Vostroknutov (2015), which relates rule-following to the management of common pool resources. The authors may wish to note this in response to concerns about the role of authority in the experiment.

Finally, since my original report, another paper using the data from the same experiment as Tate et al. (2022) has come out in *Experimental Economics* (Kimbrough et al. 2024). In that paper, we show that rule-following measured via the ball-and-bucket task does not change significantly when measured twice, 10 weeks apart, in a sample of Colombian and Northern Irish adolescents. This further corroborates the finding that RF behavior does not diminish with repetition.

Kimbrough, E.O., Krupka, E.L., Kumar, R. et al. On the stability of norms and norm-following propensity: a cross-cultural panel study with adolescents. *Exp Econ* 27, 351–378 (2024). <https://doi.org/10.1007/s10683-024-09821-5>

Kimbrough, E.O. and Vostroknutov, A., 2015. The social and ecological determinants of common pool resource sustainability. *Journal of environmental economics and management*, 72, pp.38-53.

Reviewer #2 (Remarks to the Author):

The revision has improved a lot. 1) The motivation at the beginning is now much clearer. 2) The authors refer to experimenter demand effects, strategically placed at the end. 3) The authors convincingly show that beliefs were not biased. 4) The description of "Measuring social expectations of rule-following" is now easy to read and has a clearer focus. 5) is resolved. 6) The discussion is now more focused. Essentially, I have only minor suggestions:

1. Page 5: "we therefore also elicit the functions $n(b^n)$ and $d(b^d)$ ". A short description of the metric of this function should be added here or on page 8. This should include that b^n and b^d denote the percentage of people who disapprove of rule breaking and that $n()$ and $d()$ are monotonically increasing functions bounded in the interval $[0, 1]$. Or, more explicitly, that $n(b^n)$ is the increase in rule compliance as a function of the expected percentage of people who disapprove of rule breaking.

2. I am a bit confused by the numbering and the alphabetical order. Is it correct that Experiment 1 consists of versions a and b? The authors might want to state this in the sentence "In Experiment 1 we observed a high degree of rule conformity".

3. On page 7, you write "The abstract task (Fig. 1b) also included control questions". What is the added value of the abstract task? The point is probably quite obvious, as participants are familiar with red lights and react in an immediate, unconscious way, which the abstract task avoids. The authors may want to explain their motivation.

4. Page 8 reads "Rule-followers are also more likely to be conditional cooperators in a social dilemma task." Consider moving this sentence further down as it better fits to experiment 2.

5. Page 8: "The results shown in Fig. 1 only inform us about a high prevalence of rule-following when it is costly and when it cannot affect anyone (no externalities). In terms of CRISP, rule conformity C can therefore only be due to intrinsic respect for rules R and social expectations S ." It would help readers get such a short hint already when experiment 1 is introduced.

6. Page 11: "Our results show that even arbitrary rules in an anonymous and asocial single decision maker setting generate social expectations that the rule should be followed and is followed, demonstrating a fundamental relationship between rules and social expectations." This is a nice point where the authors might want to reflect longer and deepen the insight to better carry home the argument. The results show that social expectations do not need a legitimate justification to restrict individual freedom, but operate even when rules do not serve reasonable ends. It is the authority of the rules that counts, not the underlying reasoning.

7. The motivation for Experiment 3 is a bit weak. The authors may want to add that the descriptive rule violation in

Experiment 2 is cold and may not fully influence the rule violation. The question is therefore whether rule compliance can survive even when actual rule violations are observed among others.

8. Page 12: "In a complementary analysis ...". There is no added value in this paragraph. I think it can be deleted, but leave the decision to the authors.

9. Page 14: "All experiments were conducted between-subjects and with control questions before the start of the experiment." For ease of understanding, the authors may wish to add here that information on normative and descriptive beliefs has been added.

10. Pages 17 and 18: "The 22% and 23% of people who follow the rule unconditionally". Add an explanation of where these figures come from. Are these the intercepts from Experiment 2? If so, they should be 33% and 22%. A more obvious choice would be the 35% and 28% from Experiment 2. This choice would also fit better with the following sentence "Moving from the lowest quintile of beliefs (0-20%) to the highest quintile of beliefs (81-100%) increased average rule-following rates by 20 and 23 percentage points".

11. The authors have now introduced a quantitative breakdown of the motivations for compliance. What strikes me is that the numbers do not add up to a total of 78% norm compliance, and the authors make no effort to somehow reach this threshold. A good guess at the numbers would be nice, e.g. maximum compliance of 78% = strong punishment 16 + social preferences 7 + social expectations 23 + unconditional 32 (or else). It would also be more intuitive if the second column in Table 1 gave such a calculation, in which case the entry in row 1 "-13" would not fit.

Overall, the study is now a nice contribution to Nature Human Behaviour and I think the authors can take care of the remaining points at their own discretion.

Reviewer #3 (Remarks to the Author):

In my view, the authors did an excellent job addressing my previous comments and concerns. The addition of Table 1 is particularly interesting, as it neatly summarizes the behavioral effects of different interventions or peer observations.

One aspect that remains somewhat puzzling to me is the substantial change in rule-following behavior when comprehension/control questions are introduced. This is especially intriguing also because it seems to introduce more inconsistencies in the rank-ordering of conditional rule-following (Fig. S3). While the white dots generally follow a logical rank-order, the black dots appear somewhat more inconsistent. I would have expected more noise or inconsistencies without control questions, but the opposite seems to be the case (although, I base this on eye-balling the data as plotted in Figure S3, of course). Hence, the two comprehension questions (a) change behavior (towards less rule-following) but also (b) the pattern of (conditional) choices.

This makes me wonder how comprehension questions in economic games/tasks influence observations and conclusions, more generally. It is more of a broader methodological puzzle for the field. Hence, I would leave it up to the authors, if they think it is worth highlighting with one or two sentences in the discussion, as studies usually do not implement manipulations on comprehension checks and this study speaks to the fact that we maybe should.

Lastly, I have one minor point. On page 13 the authors write: "This observation is consistent with the "bad apple effect", according to which just one defector can diminish cooperation in social dilemma problems". There is a previous study that used the rule-following task to show that 'rule-followers' can actually induce higher rule compliance (in rule-violators) in a collaborative cheating task, somewhat at odds with the bad apple effect. It may be worth citing this study, as it uses the rule-following task and examines (strategic) interactions between 'rule-followers' and 'rule-violators'; something not covered by CRISP; Gross & De Dreu (2021). "Rule following mitigates collaborative cheating and facilitates the spreading of honesty within groups." *Personality and Social Psychology Bulletin*, 47(3), 395-409.

Other than that, I would like to congratulate the authors on a fascinating study!

Reviewer #4 (Remarks to the Author):

Thank you very much for giving me a chance to revisit this manuscript. I have read the response to reviewers and found that the authors responded to every single point raised and took appropriate steps to address them. I find the CRISP formula a good compromise between demands of different disciplines and found the argument for changes convincing in all cases. I thus have no further comment save for the suggestion that the final graphics files might need to be of higher resolution (or in vector format as appropriate; they appear pixelated at the end of the document).

Reviewer #4 (Remarks on code availability):

I have checked the contents of the archive and found data and analysis code inside. The data is nicely organized and the code is fully commented and easy to relate to analyses in the paper.

18 January 2024

Re: Resubmission of manuscript NATHUMBEHAV-23124337

Dear [REDACTED]

This cover letter explains our response to your decision letter dated 12th January 2024 on our submission NATHUMBEHAV-23124337.

We are glad to hear that you find our paper of potential interest for NHB. You requested some revision before deciding whether to send the paper to reviewers.

We have revised our submission as per your request:

You wrote:

1. Please report all replication studies in the same detail as the studies in the main text.
2. Please inform whether any of the replication studies has been published. If so, please cite and discuss it/them in the main text.

Regarding (2), two replications of our baseline task (as part of separate research questions) are available either as a preprint (ref #35) and as a PhD thesis (ref #36). We now mention them both in the main text.

Regarding (1), all other replications, mentioned in the previous version of the SI, are co-authored by at least one of us, but are separate *unpublished* studies and are *not even written up*. Because they are not publicly available and therefore not citable (and will not be in the near future because we are still working on them and collecting further data) have therefore *removed* them from the Supplementary Information. We think this is the prudent decision, for the following reasons: First, these are all full-fledged separate studies with their own research questions, building on our paper and extending our research into rule-following to new questions beyond this paper, which we consider to be a foundational paper on the sources of rule-following behaviour. The other papers deal with questions unrelated to our current one, such as the role of self-control, and cooperation. Second, because the data of these replications will be part of separate studies and future publications, they are not included in the dataset of our paper which we have already made available (<https://doi.org/10.17605/OSF.IO/7WZ4F>).

We had included these replications in the SI because these studies all have a condition akin to our baseline setting and citing them was meant to assure the reader that we are talking about a robust phenomenon (hence we called it “replication”).

Reflecting upon this situation and realising that conventions regarding citing ongoing unpublished work differ in many disciplines from those in economics (where it is not uncommon), we decided to remove from the SI those studies which are not yet citable and, as you requested, to cite in the main text those that are publicly available.

We hope you agree with our response. We would of course be happy to reconsider our response if you think an alternative course of action would be more appropriate.

Many thanks for your kind consideration and best regards

Simon Gächter, Professor of the Psychology of Economic Decision Making (corresponding author)

Response to Reviewers

Reviewer #1:

Thank you for your comments and suggestions regarding the literature. This is very helpful and we have incorporated all suggested references.

Remarks to the Author:

The authors build upon work by Kimbrough and Vostroknutov that introduced an incentivized measure of individual rule-following, and they perform a huge, systematic analysis (with over 14,000 respondents) of the factors influencing individual rule-following. They find that, although rule-following is sensitive to incentives, externalities, and information about both injunctive and descriptive norms, it is to a large extent reflective of an intrinsic motivation.

The amount of work undertaken and the authors' attention to detail are both commendable. The theoretical framework the authors introduce is clean and neatly frames the research question; the experimental design is careful; the results are presented clearly with a series of elegant figures that tell the whole story in pictures; the main finding is striking and important, given the connections between rule-following and many aspects of social behavior.

We are glad you find our results “striking and important”. Thank you also for the detailed references, which we all cite at relevant places in the paper. We indicate underneath each suggested reference where we now cite it.

My main suggestions are about other related work that ought to be cited:

1) Kimbrough, E.O. and Vostroknutov, A., 2018. A portable method of eliciting respect for social norms. *Economics Letters*, 168, pp.147-150. -- This paper introduces another abstract variant analogous to the original stop-light task and shows that similar patterns of rule-following are observed across university samples from 5 countries. In this task subjects are given a number of balls and told that the rule is to put them into the blue bucket; there is also a yellow bucket, and putting the balls in the yellow bucket earns them more money, so the tradeoff is again between self-interest and adherence to the rule.

We now cite this paper prominently in the main text (we had cited it in the supplementary materials). We now write the following about it (on p.7): “High rates of rule-following were also observed in another abstract task, the “ball-division task”, where participants were told that the rule is to put balls into a blue bucket although a yellow bucket earned more money⁵⁸. Experiments revealed similar rule following rates across participants from five countries (n=1,138 participants; average rule-following rate of 56%)⁵⁸”

2) Tate, C., Kumar, R., Murray, J.M., Sanchez-Franco, S., Sarmiento, O.L., Montgomery, S.C., Zhou, H., Ramalingam, A., Krupka, E., Kimbrough, E. and Kee, F., 2022. The personality and cognitive traits associated with adolescents' sensitivity to social norms. *Scientific Reports*, 12(1), p.15247. -- This paper uses a sample of ~1500 adolescents who completed the task in (1) and assesses the extent to which their behavior is related to a wide variety of personality and cognitive traits.

We now cite this paper and ref. 4) below when we talk about the implications of our results on shame-proneness and rule-following. We write the following (on p. 7): "Rule-following varied little with socio-demographics and Big Five personality traits⁶⁰ (coefficients are very small and unsystematic, similar to findings in a repeated traffic like task⁶¹) but was positively related to being 'patient'⁶² and being 'shame-prone'⁶³ (Tables S2, S3).".

3) McBride, M. and Ridinger, G., 2021. Beliefs also make social-norm preferences social. *Journal of Economic Behavior & Organization*, 191, pp.765-784. -- This paper offers additional evidence that rule-following is, to some extent, conditional on beliefs about others' rule-following.

We now cite this paper (ref. #65) when we introduce our elicitation of conditional preferences (on p. 9).

4) Ridinger, G., 2020. Shame and theory-of-mind predicts rule-following behavior. *Games*, 11(3), p.36. -- This paper shows that shame and rule-following are correlated in a university sample.

We now cite this paper and have added a sentence describing it (on p. 8): "The latter result corroborates previous evidence that rule-following is positively correlated with shame-proneness⁶⁴." See also our response to paper 2).

5) Abbink, K., Gangadharan, L., Handfield, T. and Thrasher, J., 2017. Peer punishment promotes enforcement of bad social norms. *Nature communications*, 8(1), p.609. -- This paper offers additional evidence for "bad norms", which you might cite on p. 14.

As you suggested, we now cite this paper in the Discussion section on p. 20 ref. #74 indicating that rule following can also support "bad norms".

6) Gross, J., Emmerling, F., Vostroknutov, A. and Sack, A.T., 2018. Manipulation of pro-sociality and rule-following with non-invasive brain stimulation. *Scientific Reports*, 8(1), p.1827. - This paper offers neuroeconomic evidence that rule-following is partly governed by the right lateral prefrontal cortex, using transcranial magnetic stimulation to increase rule-following.

7) Kimbrough, E.O. and Wilson, B.J., 2022. Rule-following. In *The Routledge Handbook of Philosophy, Politics, and Economics* (pp. 108-120). Routledge. -- This chapter offers summaries of theories from economics, psychology, and philosophy regarding why people follow rules (and norms).

8) Gross, J. and Vostroknutov, A., 2022. Why do people follow social norms?. *Current Opinion in Psychology*, 44, pp.1-6. -- The authors consider theories of why people follow norms, which is relevant since many authors (perhaps sloppily) use the words rule and norm roughly interchangeably. They consider learning and internalization, social image and self-image, and conditional preferences, some of which overlap with the explanations tested herein.

We now cite papers 6) to 8) (refs. #34, #38, #37), when we introduce our conceptual framework on p. 2. We have added a sentence formally acknowledges some of the most important influences from across the human sciences on our framework: “CRISP incorporates arguments from across the behavioural sciences^{1,14,15,17,18,20,22-39} and is inspired by related efforts to integrate the behavioural sciences^{40,41}”.

In the interest of transparency, I'm signing my report. Thanks for an excellent paper. It's one I read wishing I had written it myself.

Sincerely,
Erik Kimbrough

We are glad you like our paper. Your 2016 paper provided crucial input into ours!

Reviewer #2:

Thank you for all your constructive comments on our paper. We have addressed them all and explain our respective revision to each point below.

Remarks to the Author:

The authors conduct online experiments with a large sample of 14,000 participants to discover a surprising behaviour: Even without externalities and without punishment, participants follow rules that are costly to themselves. To do this, the authors invent a novel traffic light task. They elicit beliefs and conduct a wide variety of treatments to disentangle the contributions of prescriptive and descriptive norms, as well as the additional contribution of externalities and punishment, which may merely piggyback on intrinsic compliance. Overall, they provide compelling data to infer an intrinsic respect for arbitrary rules that survives even in dismal environments with many rule

violators and poor prescriptive norms. Overall, I would like to see this published. At the same time, the current study needs substantial editing and a reworking of the punchline. In addition, some experiments can be omitted to make room for more intuitive explanations.

We are glad you think that we “provide compelling data to infer an intrinsic respect for arbitrary rules” and that you “would like to see this published”.

1. I would appreciate a bit more motivation at the start. In particular, the authors refer to “obedience or an intrinsic respect for a rule because a rule is a ‘deontic constraint’ that places an interior demand on how a person should behave... because [persons] respect the prescriptive or proscriptive demand a rule places on them regardless of the social expectations they hold”. I am very sympathetic to this statement, but I think it deserves a little more elaboration, supported by examples or anecdotes. The reader should be involved in this deontic constraint and be drawn into the thought. Sometimes we observe the opposite, that rules are broken irrationally (children defying reasonable parental rules; subordinates denying the authority of superiors by doing the opposite of what they have been told to do). So where does the intrinsic tendency to follow rules fit in?

Thanks for this comment. We have re-written the entire introduction and hopefully improved our motivation. We also made changes throughout the entire text to address your comments here. For instance, we now explicitly discuss irrational rule-breaking, when the incentives suggest following them, e.g., on p. 15 we now write: “There is also a fraction of 6% of people who, irrationally (in terms of foregone payoffs), and maybe defiantly, violate the rule unconditionally (Fig. S7).”

2. The intrinsic respect for rules could be seen as a kind of experimenter demand effect. I suspect that the two are not easily disentangled. I think the authors should address this and provide an argument for the relationship between the two, for example, that what is commonly referred to as the experimenter demand effect might just be a variant of intrinsic respect for rules.

Yes, rules and following them can be seen as an “experimenter demand effect”, not least because all rules are “demands” to perform a particular action. We take up this point in our discussion section, where we write under “caveats and tasks for future research (on p. 21): “Fifth, in our experiments, the rule was stated by an experimenter and rule-following in our experiments is therefore a kind of “experimenter-demand effect”⁹², which is often seen as problematic. In our context, the “experimenter demand effect” is not problematic but rather the object of our study, because it is the nature of rules that they demand a certain behaviour. In reality, some rules (laws, regulations, orders, guidelines) are set by an authority and others (social norms) emerge via social learning^{12,93-96}. In this paper, we made a start in studying the behavioural principles of following a clearly stated rule by

using the authority of the experimenter to set the rule. Because in reality some rules are set and some emerge, studying the role of rule-setting and emergence through the lens of CRISP is another interesting task for future research.”

3. Page 4: “rule-following participants lose at least 60% of their MU.” At this stage, readers should be better informed about the instructions, i.e. that the green light appeared after 12 seconds in all runs and treatments. This also raises the question of whether participants may have formed biased beliefs about the cost of obedience. They might have expected the green light to appear earlier and would not have been willing to sacrifice 60% of their income. The authors should address this objection. Does the repetition of the game described later contradict such biased beliefs?

We have three responses to these comments. First, we are now clearer in the main text (and in its new Methods section) that the choice of complying/violating is made in real-time, i.e., if people want to stop waiting for the green light after x seconds because they do not want to sacrifice more of their income, they can do so in the experiment. We now provide two short videos in the Supplementary Information that show what participants actually saw and how decisions unfolded (see also our response to your minor point 3) below).

Second, in the strategy method experiments of Experiments 2 people faced a binary choice Wait for Green Light/Move where we specified the earnings associated with each choice (7 points if Wait, 17 points if Move), so in those experiments they made choices knowing what the cost of obedience would be.

Third, regarding repetition: other reviewers made related comments. We now address stability more explicitly than previously and find that rule-following (without feedback) is stable across the three repetitions we ran. We now explain this on p. 13, where we write: “Participants in all treatments completed a third and final iteration of the rule-following task without peers. The participants of the baseline treatment (with no peers in either iteration) therefore had to choose three times alone whether or not to follow the rule. Their rule-following rates across the three iterations were 74%, 85%, and 83% (without control questions; $n=181$), and 50%, 63%, 59% (with control questions; $n=239$).” We conclude from this result that rule-following without feedback is stable and consistent with beliefs. Things might be different with feedback, and we address this point in our Discussion section, where we write (on p. 20/21): “Fourth, except for Experiments 3 we have studied rule-following in single-decision maker settings without feedback of what other people did. In Experiments 3 people made only one decision after they saw what their peers did (*cf.* Fig. 3). Single-shot decisions are unrealistic in situations where people can repeatedly

learn about the rule-following of other people. Related research using public goods and donation games suggests learning will likely lead to less rule-conformity^{31,83,84}. Moreover, as our results also suggest, more rule-violations may also make them more socially acceptable (Fig. S5). Future research should therefore study the fragility of rule-conformity²⁷, its population-level dynamics^{18,35,85-87}, and the consequences for normative beliefs⁸⁸⁻⁹¹.”

4. The description of the experimental design in "Measuring social expectations of rule-following" is difficult to read. Do these subjects play the same traffic light game as before (participants) or do they just observe others playing the game (spectators)? I have read the first paragraph several times and am still confused. I tried to understand this by reading S.1.4, which should not be necessary for the lay reader. Here we are informed that "after reading the instructions that this decision-maker would receive (these were the same as the instructions given to participants in the traffic light task) participants were asked to evaluate two possible behaviours". The confusion could be avoided by using "participants" only for those playing the traffic light game and "spectators" for the others. My confusion continues with the information about "four between-subjects diagnostic tasks". There are two tasks that elicit normative beliefs and two tasks that measure people's conditional willingness to follow rules, right? Do I understand correctly that the latter two tasks no longer refer to spectators, but to participants who are given information previously obtained from spectators? These participants (are they the same as in the first experiment?) are then shown five different scenarios which vary beliefs. I am not convinced by the use of the strategy method here, because social norms then seem rather hypothetical, rather "cold". The authors may wish to defend their approach or concede that 'hot' descriptions of social norms might have been more influential. That said, there is an excess of numbers in the description, and the authors should perhaps focus on the essential data.

Thank you for this comment and apologies for the lack of clarity. We are now following your advice and label the participants of the experiments that elicit normative and descriptive belief as “spectators”.

You have correctly understood the four tasks: in between-subject experiments, we elicited either the normative beliefs (shown in Fig. 2a); the descriptive beliefs (shown in Fig. 2b); the conditional rule-conformity as a function of normative beliefs (Fig. 2c) and the conditional rule-conformity as a function of descriptive beliefs (Fig. 2d). The first two were elicited from spectators, and the last two from participants who had to make a choice whether or not to follow the rule using an incentive-compatible strategy method (make a decision for five possible quintiles of disapproval rates of rule-breaking or five possible quintiles of actual rule-following rates).

The strategy method is crucial for our purposes: some people might follow the rule if sufficiently many people disapprove of not following or if sufficiently many do

follow. The strategy method allows measuring this threshold for each individual and it reveals that, on average, people are conditional rule followers. However, crucially, the strategy method also allows us to determine the unconditional rule violators and the unconditional rule followers. People may have correct beliefs about what others think is appropriate and what others do, but how this translates into own behaviour (if at all) is a different matter. The strategy method provides the necessary data. The crucial paragraph on the rationale of the strategy-method experiment within CRISP is this (on p. 8): “Eliciting the conditional rule-conformity functions $n(b^n)$ and $d(b^d)$ is crucial for the following reason: Some people may only conform with the rule if sufficiently many others disapprove of rule violations, or if sufficiently many others do follow it too, and thresholds may differ between people. Eliciting $n(b^n)$ and $d(b^d)$ also allows us to determine unconditional rule-followers as a proxy for those who have an intrinsic respect for rules, regardless of their social expectations, and unconditional rule-violators as a proxy for those who disrespect rules even if they believe most other people do follow them and approve of the rule.” With the help of the strategy method, we can determine that about 20-25% are unconditional rule followers (Fig. S2 and BL treatments in Fig. S6), which is evidence for an intrinsic respect for the rule because rule-following does not depend on what others do or think.

In response to your comments, we have rewritten the description of the experiments and their justification. See the first five paragraphs of section “Social expectations and conditional rule-conformity” on page 8/9. To comply with the NHB guidelines, we have also added a Methods section to the paper, where we provide more design details (see p. 24/25).

5. Page 11: “Descriptive beliefs b ” may also influence normative beliefs $b!$ if observing peers breaking the rule leads people to revise their normative beliefs.” This appears to be a finding without context. Isn't that somewhat talking to a different literature? In my view this part may be considered for deletion.

We have followed your advice and taken this mostly out of the paper. At the request of the editor, rather than deleting it, we have moved Fig. 3b to the Supplementary Information (now Fig. S5). In the main text, we now mention these experiments only briefly: on p. 13 we now write: “In a complementary analysis ($n=490$ new participants), reported in the SI (Fig. S5), we show that descriptive beliefs also influence normative beliefs. Observing rule-breaking does not change the social appropriateness of following the rule, but it makes rule-violations less socially inappropriate.”

6. I do not think that the conclusions summarise the results well and speak to the cited literature. For example, the authors state that “the conditionality on social expectations can also lead to contagious rule violations: the asymmetry we observe (Fig. 3a) suggests that fragility is more likely than stability”. This is not in the spirit of the previous explanations, which show a surprisingly high degree of intrinsic rule compliance. It

repeats what Bicchieri and many others have done before. Rather, the finding of this study should be that there is a substantial bias towards compliance that serves as a basis for conformity even in dismal environments. Another conclusion states: "Another hypothesis is that societies organised around kinship may have developed a weaker sense of obligations to follow rules in interactions with strangers. Our study offers a framework to address these important open questions". It would be interesting to investigate this, for example by observing whether compliance depends on whether the rules are issued by friend or foe, kin or stranger. But I do not see that the study provides a framework for understanding such questions.

Thank you for this fair comment. We agree with of your points here. In response to your comment, and some related comments by other reviewers, we have entirely re-written the Discussion section of the paper (see p. 19). We hope you like the new version of the Discussion section.

My minor points:

1. Page 5 refers to figure S1. I suggest moving S1 to the main text. It took me quite long to understand the main text without this figure, which well shows the 12 seconds condition. Also, it separates between the two variants with and without the control questions. It then also explains how the two conditions add up to the 7732 participants. Otherwise, the calculations in the main text that refer to 0.66 (later referred to as 0.656) are hard to follow. Readers want to see the results being separated with and without control questions anyways.

We have followed your advise and created a new Fig. 1 that merges the former SI figures and the rule following rates for the traffic light task and the abstract task which we now separate by whether control questions were present or not.

2. Page 5: "These results suggest robustness of rule-following rates across platforms holding cultural background constant." This confusing. Is the cultural background across the UK, USA and Sweden really constant?

We no longer make this argument.

3. Page 9: "In the other twenty-seven treatments, participants in the second iteration could see on their screen, in addition to their own circle, other circles programmed to simultaneously display the movements." The task runs across 20 seconds. How did peers violate the rule? Dynamically such that one-by-one can be observed staying or crossing the red light or statically with only a single number being reported after 12 seconds? Please explain in the main text!

Dynamically, but not one-by-one. The peers were programmed to move along the screen together with the participant's circle and then either jump the light or stop,

depending on the treatment. The peers were programmed based on previous actual participant choices. We now provide two short videos in the Supplementary Information that illustrate how the tasks looked to the participants. For ease of access, we provide two links here:

<https://cmstudies.net/media/TrafficLightTaskSolo.gif>

<https://cmstudies.net/media/TrafficLightTaskSocial.gif>

As requested, we also clarified this in the main text, where we now write (on p. 11): "... participants in the second iteration could see on their screen, in addition to their own circle, other circles programmed to display the dynamic movements of 'peers', sampled from real movements of previous participants (Fig. 3a shows an example with six peers who violate the rule)."

In the Methods section, now part of the main text, we write (on p.22): "In each treatment, the actual moving behaviour of randomly selected actual previous participants (subject to the treatment composition) was shown (for a video illustration see TrafficLightTask_peers.mov in the SI)."

4. Fig 3 explains: "Dots show rule-following rates (with percentage point changes from the no-peers baseline rule-conformity rate of 77%; horizontal line) in iteration 2 in treatments with and without control questions." This sentence took me a while to understand. Consider deleting the numbers and show only the location of the dots. The figure will be easier to read without the numbers, which do not reveal a substantial value added.

Reviewer 4 also commented on this. In response, we amended Fig. 3 and added an arrow with 'no peers baseline'. We also deleted the percentage numbers. See the new Fig. 3 on p. 12.

5. Page 10: "Participants in all treatments completed a third and final iteration of the rule-following task without peers." The second iteration was described as taking place after observing the behavior of peers (not simultaneously). This suggests that the third iteration just repeats the second. If that is so, what is the value added of the third iteration?

Two other reviewers made related comments. The purpose of the third iteration, where participants again act alone (as in the first iteration), is to see how the experience of peer behaviour in iteration 2 spills over to rule-following in iteration 3. We think the value added is two-fold. First, our baseline (no peers in either iteration) allows us to provide a benchmark of rule-following stability without any feedback and observation of peer behaviour. Second, as mentioned, it allows us to assess the spillover of prior experience of rule-breaking or rule-following (in iteration 2) on subsequent behaviour (in

iteration 3). We have therefore added the following information (on p. 13): “Participants in all treatments completed a third and final iteration of the rule-following task without peers. The participants of the baseline treatment (with no peers in either iteration) therefore had to choose three times alone whether or not to follow the rule. Their rule-following rates across the three iterations were 74%, 85%, and 83% (without control questions; $n=181$), and 50%, 63%, 59% (with control questions; $n=239$). Against this benchmark, those participants who had previously observed peers breaking the rule (in iteration 2) were more likely to break the rule later (in iteration 3, without peers). A 1% increase in observed peer violations in the second iteration increased participants' own rule-breaking in the third iteration by 0.12% (Fig. S4; Table S7). Across all treatments, the lowest rule-conformity rate in iteration 3 was 58%. These results indicate that observing rule-following behaviour by peers can have lasting effects, which is relevant for the dynamics of rule-following.”

6. Page 13: “The dark-shaded parts of Fig. 4 show the average fraction of people who conformed unconditionally.” I am confused by “unconditionally”. Did participants play the original traffic light game or the modified one with strategy method? I assume it was the latter but readers will want to know this upfront.

We are sorry that our explanations have been a bit too short here. “Unconditionally” indeed refers to the strategy method. This is the fraction of people who in the strategy method experiments (that are part of Experiments 4 and elicited the conditional descriptive preferences – see Fig. S6c,d in the SI) chose to comply regardless of how many other people violated the rule (that is, in all five quintiles for which we elicited a response). For instance, in the BL experiments, we observe a rule-following rate of 55%. The strategy method experiments (run with other participants) showed that 27% complied with the rule in all five quintiles and therefore we call these people “unconditional rule followers”.

We have relegated these data, which we have collected for all experiments reported in Fig. 4, to the Supplementary Information. Some participants played the baseline rule-following game, and other participants did the norm beliefs, the descriptive beliefs, and the conditional preferences experiments. These experiments are replications of Fig. 2 for each treatment of Fig. 4 and are reported as Fig. S6. We now also briefly describe these experiments in the Methods section (see Experiments 4 in Overview of all experiments, p. 22).

To make the experiments eliciting the social expectations for each of the new experiment salient upfront, we now write (on p. 14): “Experiments 4 also measure social expectations (b^n, b^d) and elicit conditional conformity with them ($n(b^n), d(b^d)$). We use a between-subjects design and the same techniques as in Experiments 2 to see, in the CRISP framework, how consequences for others and extrinsic incentives affect social expectations, and conditional conformity with them.”

We have also amended our description and now write the following (in the caption of Fig. 4): “The dark green shaded parts of the bars show average proportions of participants who conform unconditionally regardless of their normative and descriptive beliefs (averages calculated from the strategy method data eliciting $n(\cdot)$ and $d(\cdot)$, those who always conformed”).

In the text we now write (on p.16) : “The dark-shaded parts of Fig. 4 show the average fraction of people who conformed unconditionally. We estimated these fractions from the shares of those participants who followed the rule regardless of either their normative or their descriptive beliefs about others’ conformity as elicited in $n(\cdot)$ or $d(\cdot)$.”

7. Page 13: “The presence of externalities (in EX) and weak and strong incentives (WP and SP) not only increased overall conformity but also the rate of participants who did not condition their rule-following on social expectations.” Do we see this finding in the figure?

Yes, this finding is illustrated in the dark green shaded parts of Fig. 4. We hope that with the improved text this is now clearer (see our response to your point 6).

8. The study should more clearly be organized by reference to Experiment 1, Experiment 2, Experiment 3 ... to better guide readers.

Thank you for this suggestion, which we have adopted. We introduce Experiments 1 to 4 in the introduction and refer to them throughout the text. We also explain them in our new Methods section.

Overall, the authors might consider deleting some of the experiments they designed and focusing more on others. For example, the experiments on "Measuring social expectations of rule-following" (Figure 2) with the four diagnostic tasks could be considered for deletion because the following experiments capture descriptive norms in a much better way. This would leave out prescriptive norms, which might be a price worth paying. Figure 3b also seems to speak to another literature. Whether descriptive beliefs influence normative beliefs is not the point of the study. The study addresses important issues that will add value to readers of Nature Human Behaviour, and once the issues are addressed, it can be a pleasurable read.

We have followed your suggestion to delete Fig. 3b (we moved it to the Supplementary Information (now Fig. S5) as requested by the editor).

We have, however, kept Fig. 2. Fig. 2 reports the results of experiments that measures central elements of social expectations in our CRISP framework. Fig. 2 shows that even arbitrary rules acquire normative significance (Fig. 2a); people believe others will follow the arbitrary rule (Fig. 2b) and conformity with the rule is conditional on normative and descriptive beliefs. Eliciting conditional conformity with rules is conceptually important because only if we observe behavioral reactions to different

levels of beliefs do we know whether people are conditional rule-followers or not (and a significant fraction of the participants are either unconditional compliant or unconditional violators). We now make this point clear in the section “The CRISP framework for dissecting rule-following behaviour” where we write (on p. 4/5): “Central to CRISP are social expectations (beliefs) that humans can entertain in any situation. Social expectations exist in two empirically measurable forms that can be separate sources of rule conformity: “normative beliefs” and “descriptive beliefs”^{1,22,29,44}. Normative beliefs (b^n) represent the extent to which people think conforming with the rule is socially appropriate and violating the rule is deemed socially inappropriate by most other people. Descriptive beliefs (b^d) describe people’s perceived degree of rule-conformity of other people. Crucially, for social expectations to influence rule-following, it must be the case that people make their rule-conformity dependent on their social beliefs: we therefore also elicit the functions $n(b^n)$ and $d(b^d)$, that is, people’s rule-conformity as a function of the fraction of people who disapprove of a rule-violation ($n(b^n)$), and as a function of the fraction of people who comply with the rule ($d(b^d)$).

CRISP identifies intrinsic respect for rules as any rule-following that is independent of social expectations, social preferences and incentives: following a rule regardless of what others do or think about it, that is, for all normative or descriptive beliefs, and regardless of incentives and social preferences. Of course, unconditional disrespect for a rule is also possible, which manifests itself in breaking the rule regardless of social expectations (and possibly regardless of incentives and social preferences).”.

We make a complementary point when we introduce our design for measuring social expectations, in the section “Social expectations and conditional rule-conformity” on p. 8, where we now write: “Eliciting the conditional conformity functions $n(b^n)$ and $d(b^d)$ is crucial for the following reason: Some people may only conform with the rule if sufficiently many others disapprove of rule violations, or if sufficiently many others do follow it too, and thresholds may differ between people. Eliciting $n(b^n)$ and $d(b^d)$ also allows us to determine unconditional rule-followers as a proxy for those who have an intrinsic respect for rules, regardless of their social expectations, and unconditional rule-violators as a proxy for those who disrespect rules even if they believe most other people do follow them and approve of the rule.”

We now also explain explicitly how Experiment 2 and Experiment 3 are linked on p. 11 where we write: “The results reported in Fig. 2d suggest that, on average, people’s rule-following will depend positively on how many other people follow the rule. To test this prediction, we designed, in Experiments 3, a modified version of our traffic light task in which participants observed how other participants had previously behaved in the task.”.

We are sorry that our previous exposition was not clear enough about the central role of social expectations and conditional conformity for explaining rule-following. We hope you agree with us why we think Fig. 2 is central for our purposes.

Thanks again for your constructive comments. We are glad that you think our “study addresses important issues that will add value to readers of Nature Human Behaviour” and we hope that you like our new version.

Reviewer #3

Thank you for all your constructive comments on our paper. We have addressed them all and explain our respective revision to each point below.

Remarks to the Author:

In the present paper, the authors employ a simple rule-following task to investigate how various factors influence rule abidance. The amount of data and the number of different studies are quite impressive, and the design of the various experiments and manipulations is creative and very clear in dissecting different elements behind rule abidance (intrinsic, normative expectations, observations of peer behavior, externalities, etc.). I genuinely enjoyed reading the paper. It happened to me multiple times that I thought, “But what about this?” only to then learn that the authors addressed this point just a couple of paragraphs further.

While none of the proposed and investigated reasons for rule abidance are novel, the work is outstanding in that it takes all of the prominent reasons into account and develops a clever general experimental framework that is employed to investigate each of the reasons in isolation and in comparison to each other. In my assessment, that is a significant advance in the field of norm- and rule-following, and the methodological innovations and empirical findings are insightful and, I am sure, will motivate a lot of follow-up work. I do not see any major flaws that should prohibit publication. However, I do have some questions and remarks that should be addressed, outlined in more detail below.

Thank you for your positive assessment of our paper. We are glad you think our paper is “a significant advance in the field of norm- and rule-following, and the methodological innovations and empirical findings are insightful”. Below please find our detailed responses to your many helpful comments.

(1) There are numerous proposed reasons why people abide by or violate rules (or norms), as the authors elegantly outline in their introduction. Developing a general experimental framework to pit these reasons against each other is, therefore, very valuable. After reading the introduction, I, to some degree, expected that the authors would use their framework (the rule-following task and all its variations) to perform a

'horse-race', pitting the different explanations and their explanatory power against each other. While, to some degree, that is the case (e.g., Fig. 4), I was left wondering; What is now the most powerful predictor of rule-abidance? Playing the devil's advocate, one conclusion of the paper is that everything (expectations, intrinsic motivation, punishment, externalities, etc.) matters, which is (while very valuable to show that within one framework) not that groundbreaking. As such, I would like to push the authors a bit on whether a broad comparison across all studies would be possible. I am not sure if the structure of the experiments would allow that (since, from my understanding, the round structure of experiments also differed and what was tested). But if possible, it would be very informative to learn which factors promote (or reduce) rule-abidance the most. For example, just having comprehension questions seems to change rule-abidance quite a lot, but in the present manuscript, it is difficult to assess how that compares to, e.g., a 90% probability of sanctions. Having one figure that shows the effect sizes of all proposed mechanisms would be very helpful (maybe even including relevant self-reported personality characteristics like patience). I understand that there could also be interactions among mechanisms and that, again, the structure of experiments also varied, which could make that complicated. But it seems an essential advantage (that is not fully taken advantage of) of having one game and then manipulating all kinds of rules and information is the ability to quantitatively compare the different factors and their relative importance. Or even go beyond that and propose a unified utility model with parameters that can be estimated from the observations.

Thank you for this comment, which has pushed us to think harder about how the various results compare in importance. This is a rather difficult question because we have not set up the experiments to be able to run a horse race. There are three problems here.

First, a horse race is difficult because we have designed our experiments to be *between-subjects*. The reason why we did this is to measure social expectations and conditional compliance with them as cleanly as possible, that is, uninfluenced by other measurements. Ultimately, it would be useful to develop a within-subject design that measures all the elements of our framework in a way that allows for a structural estimation of parameters (of a utility function version of our framework). We think this is beyond the scope of this paper but an important task for future research and we mention this in our Discussion section, where we write the following in the "caveats and tasks for future research" part of the discussion: "Third, our experiments are "between-subjects", which does not allow us to study how the behavioural motives of CRISP are linked at the individual level. Because potentially all four motives behind CRISP can determine rule-following in many realistic settings, developing a within-subject framework that allows comparing the relative importance of the behavioural motives at the individual level is an important task for future research."

Second, conceptually, in many real-world situations demanding rule-following, there can be a confluence of motives, and which ones are relevant, and their relative strength, may vary across situations. Our framework (now labelled with the acronym CRISP to explain rule-conforming as a function of respect for rules, incentives, social expectations and social preferences) is meant to provide a conceptual tool to think about those motives in any situation of interest.

Third, specifically to our experiment, a horse race is difficult because motives are not systematically competing with each other but are present or not, and if present, add to each other. This is most easily seen in Fig. 4, where we present the results of a nested design: in baseline BL only intrinsic respect and social expectations are possible, whereas in EX we add externalities and hence a potential for prosocial motives; in WP we add the possibility of weak punishment to EX and in SP we add strong punishment to WP. Because of this nested design we can read off from Fig. 4 what the added motives do in terms of increased rule compliance when we move from BL to EX to WP to SP. A caveat to this is that the extent of how EX, WP, and SP influence rule-following in our setup is inevitably restricted to the specifics of our designs. Our experiments are one instance designed to provide a “proof-of-concept” that illustrates the point. How generalisable these effect sizes are, depends on the specific parameters and further design details, the exploration of which we deem beyond the scope of this paper but clearly interesting for future research. Our CRISP framework provides the conceptual toolbox to analyse any situation of interest. We discuss this at the end of the paper, where we now say (in our Discussion section on p. 20): Some caveats and tasks for future research are in order. First, our results are from one minimalist stylised setting, the traffic light task, where the rule asks people to wait, thereby demanding patience. While we kept our setting constant across our four experiments, the quantitative effects we found are valid within that setting and may be different in other tasks and rules that do not require patience. More generally, rules in reality come in many different forms and vary, for instance, in ambiguity and observability of their violation. However, our CRISP framework and its experimental methodology can be deployed to study how the motivational elements of CRISP interact to produce rule-conformity of any rule (lab rules or naturally occurring rules). For instance, intrinsic rule-following is likely more important in many naturally-occurring environments where rules have meaning^{1,5,8,9,14,17,70}, evoke feelings of moral obligation⁷¹, or appeal to internalised values³² and self-image concerns⁷². A related caveat is that rule-following in real-world settings can be socially harmful such as in bad social norms⁷³⁻⁷⁶ or if the rule demands nefarious acts^{19,77}. Again, the CRISP framework provides the toolbox to study the behavioural properties of such rules.”

Nevertheless, we take your point that it would be useful to compare the effect sizes (expressed in percentage point changes in rule conformity) of factors that affect rule-following behaviour. We now provide such data in terms of CRISP. Our approach to gauge effect sizes is descriptive and we have devised a table that collects the main results as informed by our framework. See Table 1 in the main text (p. 18) and the text

around it in the new section “Putting the behavioural elements of CRISP into perspective” (on pp. 16-18). We hope this table provides a helpful step in the direction you have in mind.

We have also taken up your point in the Discussion section of our paper (on p. 20), where we write: “We estimate that about 23% of people conform with the rule unconditionally, that is, independent of social expectations, and around 30% of people are conditional rule-followers. The 23% rate of unconditional rule-following is substantial when compared to the extent to which conformity with varying social expectations and extrinsic incentives shift rule-following (by 20-25% and 23%, respectively, cf. Figs. 3 and 4; Table 1).”

(2) The implemented task (traffic light or neutral frame) requires people to wait. As far as I understand, people also did not know how long they had to wait. This introduces an element of patience, such that other prominent psychological constructs could play a role here (i.e., patience, self-control, impulsivity). And indeed, the authors also find a correlation with self-reported patience (Table S3) and what appears to be dynamically inconsistent preferences: Many people decide to wait a bit before violating the rule rather than violating the rule from the start (Fig S1).

Since the game is kept constant across all treatments, this feature does not invalidate the results and comparisons across treatments. Still, this seems to be an important element (i.e., self-control capacity or whatever term one wants to use) that is not necessarily linked to norm/rule-following (and the investigated determinants in the paper), as many rules are arguably followed out of (internalized) habits and do not require self-control. Relatedly, another version of the original traffic light task – the ball-division task – in which people do not have to wait but have to divide balls between two buckets according to a rule gets rid of this confound with self-control/patience, etc. Again, I do not think that this challenges the validity of the results that rest on comparisons within the same task. But it remains a somewhat open question what role patience or self-control plays over and beyond the discussed reasons for rule abidance. This could be discussed more as a potential caveat/future direction.

We agree with your observation about patience, which one can indeed see as an *ex ante* disadvantage of our rule-following tasks. However, some rules in reality do have this feature and people’s impatience does make rule-following costly in these situations. The fact that we also observe that patience matters in our task therefore lends psychological credibility to it. We now write the following about his point in the main text (on p. 7): “Rule-following varied little with socio-demographics and Big Five personality domains⁶⁰ (coefficients are very small and unsystematic, similar to findings in a repeated traffic like task⁶¹) but was positively related to being ‘patient’⁶² and being ‘shame-prone’⁶³ (Tables S2, S3). The former result lends psychological credibility to our task because impulse control contributes to the psychological costs of rule-following in our setting. The latter result corroborates previous evidence that rule-following is positively correlated with shame-

proneness⁶⁴. Rule-followers are also more likely to be conditional cooperators in a social dilemma task³¹. These correlations suggest that our rule-following tasks capture a general tendency to follow rules, which activates psychological mechanisms that also play a role in other settings^{53,58,61}” Irrespective of this issue, we also agree with you that it should not affect between treatment comparisons because we have kept this feature constant in all experiments.

As a side note, there are more studies than cited that used the ball-division task and investigated rule abidance (on the individual or group level). For readers, I think it would be nice to have a more complete reference list of previous work on this task to get an overview of previous work and how different factors change rule abidance (i.e., including studies using the ball-division task that is simply a variant of the traffic light task).

Thank you. In response to your comment and of Reviewer 1 we now have added more references that also mention the ball-division task.

(3) I was wondering whether the rather high base rate of rule-abidance (e.g., Fig. 1) could be (partly) explained by, albeit, wrong expectations of punishment for violations. People may (wrongly) assume that not following the rule would lead to some negative consequences (e.g., a bad approval rating for AMT participants). In die-rolling or other cheating tasks, people often get the opportunity to violate rules in private such that neither the experimenter nor the computer can 'prove' that someone cheated (i.e., violated the rule of honesty). Arguably, such ambiguity may also play a role for rule-following outside of the lab, but I was wondering whether the authors have any data that could corroborate or debunk this reasoning (e.g., a post-experiment questionnaire). Maybe it could be discussed for follow-up work that could manipulate to which degree rule-violations are recorded/observed/can be detected only ambiguously (similar to cheating tasks).

Thank you. Although we do not have hard data on this issue, we think this is not a big problem. The main reason is that we get very similar results in our (and others') laboratory where participants are known to the experimenter and do not have to fear bad approval ratings. We now write the following about this issue in the main text (on p. 7): “These findings suggest that our online results are not due to diminished control in the online environment (and fear of not receiving approval after the experiment in the online experiment)⁵⁰.” We have taken up your points in our Discussion section (see also our response to your point (1)).

(4) On page 10, the authors write: “Despite contagious rule violations, overall rule-conformity rates are still remarkably high in absolute terms: even when six peers violated the rule, 55% of participants still followed the rule. Focusing only on conditions

with control questions, rule-following never dropped below 40%, even when six out of six peers violated the rule”. Here, I asked myself: What would happen if the game is repeatedly played, and people repeatedly observe high (or low) rule abidance by others? Similar to the public goods game literature, I would expect that social dynamics actually need some time (or, in other words, repetition) to unfold. Also in reality, this may play an important role. When a person moves from a city with a low frequency of jaywalking to a city where this is very common (let’s say New York), I would also expect that people first stick to their “internal” or learned inclination but would eventually adapt to others. The study considers many interesting factors. The one thing I felt was missing, however, was contagiousness across multiple rounds. Maybe this could be at least discussed, especially when the danger is that readers take the above statement at face value and wrongly generalize (peer influence may be much, much stronger, but it may take more than one round).

This is of course a valid point but in this paper we are not in a position to address this issue conclusively. We agree that it is very likely that in a repeated rule-following task with feedback rule-following likely will decline. We now briefly address this point in the Discussion section where we write: “Fourth, except for Experiments 3 we have studied rule-following in single-decision maker settings without feedback of what other people did. In Experiments 3 people made only one decision after they saw what their peers did (*cf.* Fig. 3). Single-shot decisions are unrealistic in situations where people can repeatedly learn about the rule-following of other people. Related research using public goods and donation games suggests learning will likely lead to less rule-conformity^{31,83,84}. Moreover, as our results also suggest, more rule-violations may also make them more socially acceptable (Fig. S5). Future research should therefore study the fragility of rule-conformity²⁷, its population-level dynamics^{18,35,85-87}, and the consequences for normative beliefs⁸⁸⁻⁹¹.”

However, in the three iterations of our setting (without peers and all without feedback), rule-following is stable. We now make this clear in the main text, where we write (on p. 13): “Participants in all treatments completed a third and final iteration of the rule-following task without peers. The participants of the baseline treatment (with no peers in either iteration) therefore had to choose three times alone whether or not to follow the rule. Their rule-following rates across the three iterations were 74%, 85%, and 83% (without control questions; n=181), and 50%, 63%, 59% (with control questions; n=239). ”

Minor:

(5) I hope I did not miss it, but how strongly were normative beliefs (Fig 2a) and descriptive beliefs (Fig 2b) correlated on the individual level? Maybe that would be interesting to report.

These experiments were run *between-subjects*, so we cannot report a correlation at the individual level. We hope that our improved description of the experimental design makes that clearer than the previous version.

(6) On page 8, the authors report that “19% of people conformed with the rule regardless of how many others disapproved of a violation; 29% never conformed; 30% followed after a threshold; and 22% had inconsistent responses (Fig. S2a).” and “The individual functions reveal that 20% always conform; 34% never conform; 31% conform conditionally; and 15% show no clear threshold effect (Fig. S2b).” Looking at Figure S2, is that correctly coded? It looks like 29% and 34% always conform (i.e., the reverse).

This is very well spotted, thank you. We made a coding error for the graph that reversed all the stacked bars for descriptive norms – the in-text numbers are correct, Fig. S2 was wrong; we have now corrected it.

(7) Page 8: The authors write “The results of Figs. 2c and 2d suggest a striking degree of conditional conformity in our asocial and anonymous setting.” A “striking degree” is arguably subjective, but I was not sure whether I would agree here with the strong word “striking”. Instead, it looks like many people are weakly influenced by social information (as mentioned in the following sentences).

We have removed the word “striking” and now write (on p. 11): “The results of Figs. 2c and 2d suggest that, on average, conformity is conditional in our asocial and anonymous setting. They also suggest that some rule-following is independent of social expectations: even if very few people disapprove of breaking the rule, or if almost everyone else breaks the rule, many participants still followed the rule in our non-social decision setting where nobody was affected by rule-breaking.”

(8) On page 9, the authors write: “To measure the causal effect of observing others’ rule-following and rule-breaking on one’s own rule-following, we designed a modified version of our traffic light task in which participants observed how other participants had previously behaved in the task.”. Maybe I misunderstand something but why is that a causal test compared to the manipulation before in which information was given about the % of conformists (experiment underlying Fig. 2d)? The experiment in which “peer” behavior can be directly observed is arguably a more elegant setup, but does it really add something conceptually over and beyond the previous study (e.g., causal interpretation that was not possible in the previous experiment as the sentence seems to suggest)?

Thank you for this comment. You are right, both approaches – the strategy method and the direct observation of what peers do – are ways to control for beliefs and therefore beliefs are “causal” for the rule-following we observe. We have now added some text at the beginning of the section “Testing the causal role of observing others for rule-following” on p. 11 to better motivate these experiments: “The results reported in Fig. 2d suggest that, on average, people’s rule-following will depend positively on how many other

people follow the rule. To test this prediction, we designed, in Experiments 3, a modified version of our traffic light task in which participants observed how other participants had previously behaved in the task.”

(9) “We did not observe a group size effect in the contagiousness of rule violations: rule compliance appears to be driven by the proportion rather than the absolute number of violators”. Here, I was wondering if this interpretation is actually following the regression results. If not mistaken, this refers to Table S6. In the first column, it looks like number of peers (absolute number) as well as fraction of rule-followers (proportion) significantly predict rule-abidance.

The group size effect in contagiousness refers to the interaction between fraction of rule-followers and number of peers (Table S6, “ $\pi \times$ Number of peers”). The small but significant main effect of “Number of peers” measures the effect of number of peers when the fraction of rule-followers is zero (and likely picks up the difference between 16% violation and 20/22% violation as we move from 1 peer to 6 peers, see left hand side dots in each of the panels in Fig. 3b). We added a sentence in the caption of Table S6 to clarify this point: “Moreover, there is no group size effect in the contagiousness of rule violations (the interaction “ $\pi \times$ Number of peers” is insignificant”.

(10) From my understanding, violating the rule reduced expected payoff in the SP treatment (see Fig. 4). Given the high base-rate of rule abidance to begin with, I would then expect abidance rates close to 100% in this case. Is there any indication to which degree participants misunderstood the task or any reasons that the authors can think of why rule abidance is so low in this case?

We don’t know to what extent misunderstanding is the reason for this lower-than-expected compliance but note that participants had to answer control questions that checked their understanding of payoffs. Note also that many people’s rule-conformity even under SP is conditional, see Fig. S6d in the Supplementary Information. People holding beliefs that not everyone is conforming might have induced some participants to not conform either. However, as Fig. S7 shows, in SP the majority of people do follow unconditionally as is predicted by standard economic theory in this case.

Here is what we now write (on p. 15): “Interestingly, although SP strongly increased rule-conformity, it is less than the predicted 100%. This might be due to the probabilistic nature of the incentives, which, despite a punishment likelihood of 90% might have led some people to gamble on getting away with breaking the rule. It can also be due to the fact that, in all treatments, including SP, many people’s rule-following preferences are conditional on others’ rule-conformity (see Fig. S6d). If some people believe that others do not obey the rule, conditional conformity with social expectations might have induced rule-breaking. There is also a fraction of 6% of people who, irrationally (in terms of foregone payoffs), and maybe defiantly, violate the rule unconditionally (Fig. S7).”

(11) Based on information in the SI (page 6), it suggests that authors fitted (multilevel) linear regressions, even though their dependent variable is binary (?). The authors argue that results are easier to interpret. Yet, logistic regressions not only seem to be more appropriate but coefficients can also be quite nicely interpreted by calculating changes in odds (from baseline). They can also be directly interpreted as effect-sizes which seems a bit more difficult for linear models fitted to binary data. Hence, I was really wondering why that choice was made, as the provided reasoning was not completely convincing to me.

Thank you for this comment. In the SI, section 1.7 we now write: “Tables S2-S6 and S9-S11 below report linear probability models fitted to participants’ binary decisions to follow (1) or violate (0) the rule. Linear models are typically robust to deviations from normality assumptions¹⁸, and their results are easier to interpret than models using a logit or probit link function to deal with the binary nature of the dependent variable. For all models we ran robustness checks by also fitting logistic models. For consistency with the other analyses, the models reported in Tables S2-S6 only include the demographic controls ‘age’ and ‘gender’. As additional robustness checks we also fitted linear models including all control variables. Unless stated otherwise, the results reported in the Tables below are robust to these variations in the model specification.”

We should add that the interpretation is easier for linear models, e.g., so that the estimates can be mapped onto the graphs and interpreted as percentage points increases. The R code we provide in the repository has the code for all models including the logistic ones (which are there clearly annotated as robustness checks).

Reviewer #4

Thank you for all your constructive comments on our paper. We have addressed them all and explain our respective revision to each point below.

Remarks to the Author:

Review of NATHUMBEHAV-23124337A: “Why people follow rules”

Thank you for giving me a chance to review this manuscript.

The authors present a series of studies examining people’s tendency to follow arbitrary rules, varying elements of the situation such as externalities, incentives, abstraction, and descriptive beliefs. Experiments show the majority of participants follow rules, even at a personal cost and with no generated benefit. The experiment also provide insight

into the psychological dynamics of rule-following and the importance of norms and social learning related to norms. The authors discuss implications for systems built upon rules and rule conformity.

From the start, I would like to emphasize that I have a very positive opinion about the submitted manuscript. This paper addresses its central question with a good number of well-designed variations and well-proportioned sample sizes. Its results tell a consistent story and provide answers to relevant open questions. In fact, I have no major points (nor major concerns) to hold against this manuscript.

Below, I will still provide a list of points that if addressed could potentially improve the manuscript. I will split these into major minor points (where I personally see a good argument to address them) and minor minor points that consist of smaller self-evident issues. I conclude with some suggestions. Even though this list is not short, I would like to reinforce that this is all intended to support a manuscript that I consider a valuable contribution to the literature. I enjoyed learning about the sequence of experiments and agreed with their interpretation. Many of the points below are concerned with readability from my perspective with a clear understanding that the authors may have good arguments not to follow at least some of these ideas.

We are glad you have “a very positive opinion” about our paper. We appreciate your many constructive comments, which we have all incorporated in our revised manuscript. We hope you find our paper more readable now.

MAJOR MINOR POINTS

1) Introduction: The first paragraph might benefit from some more detail and full explanations of concept as opposed to merely citing central theoretical terms in scare quotes.

We have rewritten and expanded this paragraph (and the entire introduction and the abstract) and hope we have made it clearer what our paper is about.

2) Use of variables: I am concerned about the use of formal variables to denote the elements of the model. On the positive side, this allows to formulate the proposed model as “ $C=f[I,E,X,B]$ ”. On the negative side:

- a. Little is learned from this formulation alone, it does not provide guidance on directionality, B is later split into b^n and b^d .
- b. Whenever elements of this formula are used later on, they are accompanied by an explanation, making the letters redundant in the remainder of the text
- c. When these variables are introduced, their domains and dimensionality is not specified. “Extrinsic motives” do not seem to be a unidimensional variable, for example. Thus, it is puzzling to find inequalities later on, such as “ $E < 0$ ” (re-labeled as extrinsic incentives on p.4). In fact, this seems to mix up motives themselves and situational

motivation as a sum of the activation of different motives [see my point about the causal model, as well]. Motives and incentives are clearly distinguished at least in psychology. On page 5, “I” is referenced as “intrinsic reasons”, which again is a different concept.

d. The formalization might appeal to economists and some analytical philosophers (who would have separate concerns, though), but I question its value for a more general audience, when there is no clear added value provided. “Negative externalities” seems easier to comprehend than “ $X < 0$ ”.

e. The letters are not helpful to distinguish the concepts: extrinsic motives and externalities could be abbreviated as either “E” or “X”, so could social expectations be (that are denoted as B for beliefs). Using the names of the concepts would thus be more helpful for the reader

f. The function names “ $n(b^n)$ ” and “ $d(b^d)$ ” are likewise not very evocative: they make sense in the context of the strategy method, but I am not sure they ease understanding.

g. “ $E < 0$ ” is used to indicate that rule-following is costly for the person, but “ $X < 0$ ” seems to describe not the consequences of rule-violations, rather than rule-following (this led to a state where I was confused what “ $E < 0$ ” and “ $E > 0$ ” referred to, and I would have expected “ $X > 0$ ”, as the charity relatively benefits from following the rule). Explaining it in natural language would not sacrifice any precision and help the reader.

Thank you for this valuable feedback, which has inspired us to completely rethink the writeup of our paper. We agree that this formal language might appeal to economists but might not be that useful/insightful for psychologists and other social scientists who we also want to reach with our paper. We have therefore pruned the text to make it more readable. This means that we mostly got rid of the formal language.

Because (conditional conformity with) social expectations are key in our framework (and we hope we are now doing a better job in explaining this), we have kept the formal notation of explaining conditional rule-following, to make clear they are a *function* of beliefs. We denote normative beliefs as b^n ; descriptive beliefs as b^d ; and conditional rule-following, which are functions of beliefs about others as $n(b^n)$ and $d(b^d)$, respectively.

In addition to pruning the text of unnecessary formality, we now label our framework CRISP. CRISP replaces the formula we had in the previous version of our paper (equation 1) but is conceptually exactly the same. We believe CRISP is a memorable acronym whereby rule-conformity C is a function of respect for rules R, extrinsic incentives I, social expectations S, and social preferences P. CRISP is inspired by a large literature across the human sciences and distils the most important motives for rule-conformity. CRISP also helps us to describe the logic of our experiments and we hope we explain that better than before. We introduce CRISP right from the start and have added a section that explains better how CRISP guides our experiments. We hope we also do a better job in explaining its various elements and how we manipulate and/or

measure them.

3) Pp.3/4: The task should be clearly described before discussing variations or relations to other tasks. Potentially, this could be done in a separate figure, showing starting position and intermediate steps. There are also some open questions: How is the circle moved by participants? Is the movement binary (one mouseclick covers the entire range) or continuous and incremental? I got enough information to evaluate the task (and the supplementary material addresses these questions), but it might still be helpful to have a clearer introduction in the main manuscript. Fig. 1 does not show the task but a screenshot of the task.

Our exposition has probably been too short. We have now added a Methods section (as per NHB guidelines to the paper that explains some of the requested details. We have also added an overview of all experiments (now labelled Experiments 1 to 4) to the Introduction and more details to the main text. We have also added a couple of videos for download to the Supplementary Information that show how the rule-following task works for participants. One video shows the single-decision maker task of Experiments 1 and 4 and one the task with peers of Experiments 3.

Here is the link to the single-decision maker video:

<https://cmstudies.net/media/TrafficLightTaskSolo.gif>

And here is the link to the example video of the rule-following task with six peers (all 26 other treatments work exactly the same):

<https://cmstudies.net/media/TrafficLightTaskSocial.gif>

4) Conditions and variations are introduced consecutively and there is little structure given at the beginning. I would strongly suggest numbering the Studies and subconditions and to start the discussion with a table detailing the presented experiments and variations. Ideally this table could also offer details of variations (e.g., use one column labeled “extrinsic incentives” and another one labeled “externalities” with a brief description of their role in the given variation. I believe I was still able to get a clear idea about each study, but especially at the beginning I would have appreciated this guidance.

We have responded in two ways to this comment. First, following a suggestion by another reviewer, we now speak of four experiments. Experiment 1 measures simple rule-conformity under our baseline conditions. Experiments 2 measure the social expectations element of CRISP. Experiments 3 measure the role of peers for rule-conformity. Experiments 4 add externalities and weak and strong incentives. Figures 1 to 4 illustrate the main finding of each of the four experiments. Towards the end of the introduction (the last four paragraphs on p. 3,4), we now refer to these experiments to give a clearer idea at the outset of the structure of our experiments and our paper but,

hopefully, without overburdening the reader with too many details at this stage of the paper.

Second, we provide an overview of all experiments, which clearly explains their structure and purpose, at the beginning of the Methods section, which is now part of the main text (as per the NHB guidelines).

5) I appreciated all of the variations chosen in this design, but I believe that their merits could have been explained more explicitly in at least one case. Which potential weaknesses are (successfully) addressed by the variations? For example, I found the abstract variation of the task very important, as it addresses the potentially inherent moral coding of a traffic light. Knowing a very well-grounded rule about stopping at red lights might create social expectations about the traffic light outside of the traffic context. Results can be interpreted in this direction, but the change is relatively small. In general, the robust collection of findings across contexts and variants corroborates the chosen interpretation.

We have three responses to this valuable comment: First, we hope that our framework CRISP helps the reader more easily to understand the purpose of each of our experiments. Second, in our new Methods section, we now explain all experiments (Experiments 1 to 4) and their purpose (see “Overview of all experiments”); we also summarise them briefly in the introduction. Third, another reviewer (R3) raised a related comment. In response, we have added Table 1, which compares the results in terms of our framework CRISP. See the new section “Putting the behavioural elements of CRISP into perspective” on p. 17.

6) The purpose of control questions with crowdsourced participants goes beyond increasing the salience of incentives and constitutes a form of attention check. Data quality issues are appropriately discussed in the supplementary material. Again, robustness checks show that results are not based on inattention, but this concern could be explicitly named.

Thank you for this comment. We have now added it. We write (on p. 5/6): “The condition with control questions arguably increases attentiveness and the salience of monetary incentives and thereby reduces the likelihood that rule-following is merely due to inattention, or mistakes.”

7) P.5: A correlation with being “patient” might not indicate a “general tendency” as much as a specific results for a feature of the paradigm: not moving requires some impulse control.

Another reviewer (R3) made a similar comment. We now write (on p. 8): “The former result [which relates to patience] lends psychological credibility to our task because

impulse control contributes to the psychological costs of rule-following in our setting. The latter result corroborates previous evidence that rule-following is positively correlated with shame-proneness⁶⁴. Rule-followers are also more likely to be conditional cooperators in a social dilemma task³¹. These correlations suggest that our rule-following tasks capture a general tendency to follow rules, which activates psychological mechanisms that also play a role in other settings^{53,58,61}.”

8) Possible alternative reasons for following rules in experiments is the potential that participants want to “help science”, not forfeit future employment opportunities (very relevant for MTurk samples), accept the authority of the experimenter and want to demonstrate they can follow instructions. These explanations---while consistent with rule-following--- are more specific to experimental settings, so I wonder whether this should be at least be considered as a potential limitation (obviously difficult to address in experiments and not what I would expect to happen in this already extensive manuscript).

Thank you for these comments. Here is our response:

- 1) Help science: We think that in our context it is unclear that this should translate into more obedience, since it depends on what participants believe is our research hypothesis (compliance? violation?) Also, our invitations to the experiment used very generic text (“decision-making experiment”) from which participants could not infer a purpose.
- 2) Employment opportunities: In general, this is certainly true for MTurkers, who rely on approval after finishing the job (all our participants had very high approval ratings) and therefore likely expected to be approved if they finished the job. Also note that in control questions we specify that breaking the rule and earning more money is a possible decision (hence we are implicitly telling them breaking the rule is “allowed” and won’t result in rejecting the HIT). Moreover, the concern about approval is absent in student samples, and yet the rule-following rate of students is 60% (58% on average online), and similar to other laboratory findings where participants do not have to fear bad approval ratings. We now write the following about this issue in the main text (on p. 7): “These findings suggest that our online results are not due to diminished control in the online environment (and fear of not receiving approval after the experiment in the online experiment)⁵⁰.”
- 3) Accept authority: yes, respecting a rule at least implicitly means accepting authority. This goes hand in hand with obedience and the fact that our task is a rule-following task (one needs some authority to establish rule in the first place).
- 4) Follow instructions: Participants follow the instructions if they move the circle by clicking the appropriate buttons, which they can show by either violating or complying with the rule. In other words, following the rule is following the instructions, but so is violating the rule.

We now discuss your points 3) and 4), which were also raised by another reviewer, in our Discussion section, where we write (on p.20/21): “Fifth, in our experiments, the rule was stated by an experimenter and rule-following in our experiments is therefore a kind of “experimenter-demand effect”⁹², which is often seen as problematic. In our context, the “experimenter demand effect” is not problematic but rather the object of our study, because it is the nature of rules that they demand a certain behaviour. In reality, some rules (laws, regulations, orders, guidelines) are set by an authority and others (social norms) emerge via social learning^{12,93-96}. In this paper, we made a start in studying the behavioural principles of following a clearly stated rule by using the authority of the experimenter to set the rule. Because in reality some rules are set and some emerge, studying the role of rule-setting and emergence through the lens of CRISP is another interesting task for future research.”

9) P.10: Again, I found the inclusion of the “third and final iteration” extremely reassuring. It directly addresses a concern I had about showing the movement of other participants: As this task seems to involve some response inhibition, seeing movement might therefore trigger responses. Seeing that the effect persists is thus important. This might be pointed out.

Another reviewer made a similar comment. We now write (on p. 13): “Participants in all treatments completed a third and final iteration of the rule-following task without peers. The participants of the baseline treatment (with no peers in either iteration) therefore had to choose three times alone whether or not to follow the rule. Their rule-following rates across the three iterations were 74%, 85%, and 83% (without control questions; n=181), and 50%, 63%, 59% (with control questions; n=239). Against this benchmark, those participants who had previously observed peers breaking the rule (in iteration 2) were more likely to break the rule later (in iteration 3, without peers). A 1% increase in observed peer violations in the second iteration increased participants' own rule-breaking in the third iteration by 0.12% (Fig. S4; Table S7). Across all treatments, the lowest rule-conformity rate in iteration 3 was 58%. These results indicate that observing rule-following behaviour by peers can have lasting effects, which is relevant for the dynamics of rule-following, but, in our setting, rule-following remained high.”

10) I stumbled across some phrasing issues that might be due to disciplinary preferences. To be clear, I believe that the studies provide highly interesting and useful evidence for learning about rule-following behavior. My question here was rather to which extent the “psychology of rule-following” is fully captured in the theoretical model and whether this would be even desirable. From my observation, especially economists like to discuss behavior as “caused” by beliefs and preferences. To some degree, this entails the danger (if taken too literally and too far) of short-circuiting psychology and reducing decision making and action to a stimulus-response chain (and ignore agency, individual differences, and subjectivity). On balance, I had the

impression that the theoretical level is well-chosen in this manuscript, I was concerned about a few phrases, though.

a. “Observing peer behavior provides descriptive beliefs” p.8 --- I fully understand what is meant here, but it sounds a bit mechanistic.

We now write (on p.11): “Observing peer behaviour provides information that feeds into descriptive beliefs, which, according to the result from Fig. 2d, should be positively related to own rule-following. Fig. 2d suggests we should see rule-following rates of at least 28% even if all observed peers violate the rule.”

b. “causal role of descriptive beliefs” [p.9] --- In which sense are they causal? They can be a basis for justifying one’s action, for choosing a consideration set of actions, etc. I would argue that a full “psychology of rule-following (p.14)” would include the full picture of how this information is utilized, made sense of and integrated based on motives, motivation, personality, etc. This full picture, on the other hand, is unlikely to be as relevant for the theoretical aims pursued in this manuscript, so I would not call it a deficit at all if there is no full account of a “psychology of rule-following” here. Personally, I would tone down these terms and rather discuss regularities of rule-following or “psychological aspects/dynamics of rule-following” or some other similar term.

We now call this section “Testing the causal role of observing peer rule-following”. This is causal, because we manipulate, via the experimental design, what people see their peers did in the rule-following task. Observing what peers did is a form of descriptive belief, in two possible ways: First, you see exactly how many out of the relevant group (of size 1 to 6 in our experiments) followed the rule and how many violated it. Second, in a Bayesian sense, observing a sample provides you with new data that might lead you to update your prior how many people out of the population you care about follow the rule. We accept your comment about the “psychology of rule-following” which is maybe too sweeping for what we actually do. We now write about “behavioural principles of rule-following”.

c. “basic proximate psychological processes”: In a similar vein, the psychological perspective is reduced to normative and descriptive beliefs and preferences to follow rules, but the processes described are not necessarily “psychological” in the sense of full process model (that I would not expect to find fully fleshed out in an empirical paper).

We have done that and now speak of “behavioural principles of rule-following”.

Again, these points are not intended to criticize the studies, hypotheses, framework or main interpretations. I also fully understand the intention to connect findings to

multiple literatures across disciplines.

Thank you for your comments. We appreciate it and because we do not want to overclaim we have toned down our claims as explained in response to previous comments.

MINOR MINOR POINTS

1) Abstract: The second sentence of the abstract is quite dense and could use some unpacking: What exactly is the relationship between respect for rules and the factors accounted for? Why would that accounting lead to a dispute?

We have rewritten the abstract not least to address this point. We now write: “Why people follow rules, especially laws and social norms, is debated across the human sciences. The importance of intrinsic respect for rules is particularly controversial. To reveal the behavioural principles of rule-following, we develop CRISP, an interdisciplinary experimental framework that explains rule-conformity as a function of intrinsic respect for rules, extrinsic incentives, social expectations, and social preferences.”

2) Abstract: There is a recurring use of the term “their engendered social expectations”. I have trouble parsing this: What is meant (I believe) is that rules create social expectations in people, but the possessive pronoun referring to rules makes no sense. Active voice (and possibly a different verb) might address this.

We have rewritten the abstract and shortened it as requested per NHB guidelines. This phrase no longer appears anywhere in the paper.

3) Abstract: “piggyback” seems avoidable

This phrase no longer appears in the paper.

4) P.1: “people with other-regarding concerns”: “prosocial” might distinguish this better from those we want to minimize others’ outcomes; “social preferences” include rivalrous motives

As suggested, we have replaced “people with other-regarding concerns” with “pro-social motives” or “social preferences”. However, in our CRISP framework, which we see as a general framework, we use the term “social preferences” which can include spiteful preferences. We now also cite Social Value Orientations (ref. #48), which allows for competitive and spiteful preferences.

5) P.2: last sentence: check use of semi-cola

We have rephrased this sentence, which now reads as follows (p. 3): “The approach in Experiment 1 is to eliminate all conventional reasons to follow the rule by creating a strong material incentive to violate it. Participants also act anonymously and alone, which mutes reputational concerns, and we remove any consequences of rule-following decisions for others to mute pro-social motives”.

6) P.3: definition of intrinsic respect (end of first paragraph): The given definition would be fulfilled if someone is paid to follow a rule (by an algorithm). There would be no social expectations in this case, but this would not constitute an “intrinsic” respect in any meaningful sense (this makes sense in the context of the experimental variation later on, but not as a general definition).

We hopefully have now clarified this by writing (in our section on the CRISP framework on p. 5): “CRISP identifies intrinsic respect for rules as any rule-following that is independent of social expectations, social preferences and incentives: following a rule regardless of what others do or think about it, that is, for all normative or descriptive beliefs, and regardless of incentives and social preferences. Of course, unconditional disrespect for a rule is also possible, which manifests itself in breaking the rule regardless of social expectations (and possibly regardless of incentives and social preferences).”

7) P.4 suggested edit: unlike in PREVIOUS tasks that have been used...

We have added the suggested edit. The sentence now reads (now p. 5): “The tasks are also non-strategic and asocial: unlike in previous tasks that have been used to study social norms²⁵, there is a single anonymous decision maker, and a violation affects nobody, which mutes pro-social motives”.

8) Mturk participants are not anonymous, but pseudonymous (in some cases identities have been compromised through security leaks).

Thank you. We now mention this in the new Methods section, where we write (on p. 20): “The online experiments were run under pseudonymous conditions, that is, it is impossible for the experimenter and any other participant to identify individuals. In the lab experiments, participant identities were known to the experimenter.”

9) P.5: “an evolved capacity of mind-reading” needs some unpacking

Thank you for this prompt. We now write the following (on p. 3): “Measuring social expectations is important because they can exist in any situation and people may have a desire to conform to these expectations^{1,22,31,44,49,50}. Humans can have social expectations even when anonymous and alone because they have a capacity of mind-reading and perspective-taking^{18,51,52}, which allows them to form beliefs what others expect of them (their normative beliefs) and what others are doing (their descriptive beliefs).”

10) P.7: caption b: switch in order between “mean: black dot” and “box: quartiles”; caption “c and d”[?] starts very dense (and in a nominalist style), I wonder whether the results at the end are ideally placed there [three digits for coefficients?]

Thank you. Caption b: We have fixed this. Caption (c and d): We have slightly reworded the text to make it easier to follow. We have kept the results at the end, but only report two digits for the coefficients to avoid pseudo-precision.

11) P.7: “when almost everyone disapproves”: the sentence is correct, but there is a potential that “almost everyone” is connected to 56% (maybe: “...increases from 0% to 100%, normative conditional conformity increases substantially, from X% to 56%.)”

We now write (on p. 9): “As the share of people who disapprove of rule-breaking increases from 0-20% to 80-100%, normative conditional conformity increases substantially, from 35% up to 56% (Fig. 2c).”

12) P.8: “19%” at the start of a sentence

Thank you. This now reads “Nineteen percent of people ...”

13) P.8: “Descriptive conditional rule-conformity” could be unpacked and explained (it is not entirely clear from the sentence alone what “descriptive” refers to [there is nothing wrong in this sentence, it is just difficult to parse]).

To enhance understanding, we have slightly amended the entire paragraph in which this phrase occurs. It now reads as follows (on p. 10): “Elicited rule-following is on average also conditional on whether other people are believed to conform with the rule. Average descriptive conditional rule-conformity $d(b^d)$ ranges from 28% when hardly anyone else is believed to conform with the rule (in the 0% to 20% quintile), to 56% when most other people (in the 80% to 100% quintile) are expected to conform with the rule (Fig. 2d). The individual response profiles of $d(b^d)$ reveal that 20% conform in all five quintiles (that is, their rule-following is unconditional on descriptive beliefs); 34% never conform (they are unconditional rule-violators); 31% conform conditionally, that is, their threshold is a quintile of descriptive beliefs larger than the lowest one; and 15% had non-monotonic responses (Fig. S2)”.

14) Fig.3: I am a bit unsure about the use of “angels” and “demons” here; it brings in a moral evaluation that flies in the face of calling the rule “arbitrary”. In addition, there is no legend identifying the icons as “angels” and “demons”. I wonder about a possible x-axis label for the diagrams in subfigure a [I actually counted them to see that these were the 27 mentioned conditions). The horizontal anchor lines represent the baseline

condition, and it could have been easier to see if it had been labeled (maybe on the right side of all diagrams). Otherwise, I like this figure panel, as it is a great way of presenting the study results.

We have implemented the suggested edits in the main text and the SI. See the new Fig. 3 in the main text, and Figs. S3, S4, and S5 in the SI.

15) P.10: It might be useful to explain the “bad apple effect”: What is the evidence for it and does this go beyond tautologies?

We have taken up your suggestion and reformulated the respective sentences, which now read (on p. 12/13): “In contrast, while all peers conforming with the rule increased rule-following slightly, the observation of just one rule-breaking peer led to a substantial decrease in rule-following. The decrease in rule-conformity for one rule-breaker is similar (around 8 percentage points) for all groups with at least two peers. This observation is consistent with the “bad apple effect”, according to which just one defector can diminish cooperation in social dilemma problems⁶⁷.”

16) P.10: “small but statistically insignificant”: Which type of test is the basis for this statement? It seems that there is a consistent pattern for all 6 conditions with fully conforming peers. Is this based on six individual tests? [This is far from crucial, as the pattern across conditions and within conditions is very robust].

Yes, these these are the 6 tests reported in the first two of Table S5. We now refer to this table in the main text.

SUGGESTIONS

1) P.5: When putting results in perspective, it might be mentioned that the rate is LOWER than the rate people give up when offered the opportunity to cheat (usually 75% of potential gains are forfeited for the benefit of being honest). Cheating is a form of not following rules. Similarly, rule-following increases with explicit promises given (even without sanctions). It might be interesting to compare and contrast the paradigm with those featuring cheating opportunities and/or promises.

Thank you for this comment. Our text was probably somewhat misleading. We now write the following (on p. 7): “To put the rule-following rates into perspective, participants in our experiments gave up 48 percent of their possible earnings by following the rule (implied by moving times, see Fig. 1). This rate is higher than the earnings people give up in dictator games⁴⁶ (where people on average forego 28% of their possible gains) but smaller than the fraction of possible gains (75%) people forfeit in honesty tasks where cheating is a form of rule-violation⁵⁹.”

2) The use of a reputation score of 90% on Mturk is probably no longer useful. In 2012, a threshold of 95% was suggested, but since it has been discovered in some studies that 99% of participants have scores higher than a 98%. After the “Mturk crisis” in 2018, when the platform encountered an influx of participants with low proficiency in English, the use of VPS/VPN-checks has been suggested as a better control method. An analysis of IP addresses is discussed in the supplement that shows corroborating evidence. Pointing at some of these discussions might be helpful in the supplement.

Thank you for this comment. We have followed your advice and have added some comments in the Supplementary Information, where we now write at the end of Section 1.3 (on p. 4 of the SI): “Based on some previous research on various topics that compared lab and AMT and found that lab and AMT samples behaved very similarly^{1,2,8,9}, we have no reason to believe that our data, which we collected between 2014 and 2018, are affected by data quality issues. However, as we learned through our problems with data quality mentioned above, things have changed in recent years^{10,11}. Relative to 2014-2018, sampling on AMT requires much sharper vetting than focussing on approval ratings of even 95% (e.g., by using expert vetting services¹²), adding attention checks¹³, and/or considering other platforms^{14,15}.”

3) One interesting variant that would follow from the formalization would of course be an “ $X < 0$ ” condition, in which rule-following causes harm to others. I do not suggest to even consider this as an addition to the present manuscript, but it could be a discussion point.

We have taken up this point in the Discussion section where we write (on p. 20): “A related caveat is that rule-following in real-world settings can be socially harmful such as in bad social norms⁷³⁻⁷⁶ or if the rule demands nefarious acts^{19,77}. Again, the CRISP framework provides the toolbox to study the behavioural properties of such rules.”

Response to reviewer comments on NATHUMBEHAV-23124337B, Why people follow rules

8 March 2025

Dear [REDACTED]

Thank you for the feedback and the guidance for the revision of our paper. We are grateful for the constructive comments we have received once again from you and the four reviewers of our paper. We are glad that they all think that the revision has improved our paper. The new set of comments has further helped to improve our paper. Below please find our response to each of their comments.

We hope you like our revision and look forward to hearing from you.

Kind regards

Simon Gaechter (corresponding author)

Lucas Molleman

Daniele Nosenzo

Reviewer #1:

Remarks to the Author:

As I said in my original report, this is a major undertaking that deserves to be published. I am somewhat disappointed by the obliqueness of the references to prior work.

Thank you for your comments. We were of course not intentionally oblique in referencing prior work. We cited related work in the more succinct science style (which tends not to mention author names) rather than the more verbose economics / social science style which is why our referencing to previous literature might have come across as “obliqueness”. We apologise if this impression has arisen.

A great many of the findings reported in this paper have been put together in separate papers by others, using variants of the research design for measuring rule-following that Vostroknutov and I developed in our 2015, 2016 and 2018 papers. For example, conditionality of rule-following and its dependence on beliefs about others have been covered by Desmet and Engel (in a slider task rule-following task) and using our task directly by McBride and Ridinger. McBride and Ridinger also use the strategy method in one treatment to elicit conditional rule-following and employ a "descriptive norm" manipulation to shock beliefs in another. Similarly, associations with ToM were shown by Ridinger. In my view, the value of the paper comes from a) the conceptual framework, b) putting all these pieces together in a single omnibus study, and c) collecting large samples.

We agree with the assessment that the value of our paper is due to its conceptual framework and putting the pieces together in a single study using large experiments.

We started our research in 2014 at which point Kimbrough and Vostroknutov's paper (published in 2016) existed only as a working paper. It did inspire our main task (which is a simplified version modelled on theirs) and we do now say this explicitly: "Our decision settings use a highly stylised 'traffic light task' (Fig. 1a) suitable for online participants. This task is a simplified version of the rule-following task originally introduced by Kimbrough and Vostroknutov⁵⁵." Further on in the text, when we compare our results to existing evidence in the literature (on p. 6), we now write "Kimbrough and Vostroknutov's seminal study, which was also conducted in a university laboratory, reports a rule-following rate of 62.5%⁵⁵." On p. 7 we now say: "High rates of rule-following were also observed in another abstract task, the "ball-division task", where participants were told that the rule is to put balls into a blue bucket although a yellow bucket earned more money⁵⁶. Their experiments revealed similar rule following rates across participants from five countries (n=1,138 participants; average rule-following rate of 56%)⁵⁶." Overall, we cite Kimbrough and Vostroknutov (2016) multiple times throughout the paper.

When we designed and ran the strategy method experiments as part of this paper, we built on our previous research on conditional cooperation. Using the strategy method in the rule-following context was therefore entirely natural for us because we had used it (indeed developed it: Fischbacher, Gächter, Fehr, *Econ Letters* 2001) in several papers (in the context of public goods games and gift-exchange games) long before we started this research (e.g., publications in 2010; 2012; 2013). We were not aware of the McBride and Ridinger papers. Both papers were published in 2021, long after we had run our experiments. We are grateful to the reviewer for having pointed out the Ridinger papers in the previous round and we are now citing them accordingly.

The paper cites most of the relevant work, but even our papers are largely referred to as "related work" rather than emphasizing that they are the very soil out of which this paper has grown. That this is the case is pretty transparent, if you know the literature, but to someone for whom this is new, it wouldn't be so obvious in the current writeup. The authors copy the traffic light setup from our paper (though theirs is simplified), and they directly use the language from our instructions "The rule is to wait at the light until it turns green." Moreover, the quote they use from Adam Smith is actually in the epigraph to our 2016 paper (and has been there since our original 2011 working paper). It's an apt quote - that's why we used it, but with the other dependencies here it is awkward. Perhaps given the timelines, there is some claim to priority with respect to these other papers (I don't know), but it is clear that our paper motivated this work. This should be transparent from the start.

As mentioned above, and as we now explicitly acknowledge, Kimbrough and Vostroknutov's (2016) rule-following task has directly influenced our task but the research programme that we then developed was our own development. Regarding the timeline, we ran all our experiments between 2014 and 2018. The starting point for our study was the task by Kimbrough and Vostroknutov but the direction our study took was inspired by a large literature on social norms, following on from Bicchieri (2005) (ref. #1)

and arguments from across the social sciences discussed by Kliemt (2020) (ref. #19). In terms of the methodology, we had worked with the strategy method for about 15 years before embarking on these experiments. In the Supplementary Information we are a bit clearer about this. We now write (on p. 5 of the SI): “To elicit these conditional rule-conformity functions, we used an approach based on the strategy method we had developed¹⁷ and used in previous research in public goods games¹⁸; subsequent research on rule-following has also used the strategy method to elicit conditional conformity functions¹⁹⁻²¹.”

We now also acknowledge that the wording of the rule is taken from Kimbrough and Vostroknutov (2016, ref #55).

We have deleted the Adam Smith quote.

In addition, a number of the concerns about the interpretation of the task raised by reviewers have already been addressed with theory and/or data in our papers.

1) We address concerns about experimenter demand explicitly in our 2016 paper, "Another way to view obedience in our RF task is as a pure “experimenter demand” effect. Under that interpretation, we are using demand effect sensitivity as a proxy for phi. This has the nice feature that a long-time bogeyman of experimenters turns out to be an ally. We are sympathetic to this view, but we would argue that any experimenter demand effect is actually a manifestation of the norm-dependence we seek to measure, else why should individuals be concerned about the demands of the experimenter?" (see fn 17) This is quite similar to the argument made by the authors in the conclusion. They might say "As pointed out by KV2016..."

Thanks for pointing this out. We now write (on p. 18) “As also pointed out by previous research^{55,68} in our context, the “experimenter demand effect” is not problematic but rather the object of our study, because it is the nature of rules that they demand a certain behaviour.”

2) We explicitly address concerns about the familiarity of the stop-light environment and the fact that there are pre-existing traffic rules which might be applied in this context with a "no rule" treatment (see Fig 2 of the 2016 paper). Subjects complete the same task with the same incentives, but we exclude "The rule is..." from the instructions. There is still some residual rule-following, though less than in the original task. This is a source of bias in the traffic task that caused us to design the ball-and-bucket version for subsequent papers. Since that source of bias is present in all treatments in this paper, it isn't a major problem for identifying treatment effects, but it may influence the reported levels.

This is a good point, and we have responded as follows (on p. 17): “People might also wait because the task is reminiscent of a real-world situation where the rule is inferred from the traffic light itself, without stating the rule explicitly⁵⁵. While we kept our setting constant across our four experiments, the quantitative effects we found are valid within

that setting and may be different in tasks that do not require patience, such as the ball-division task⁵⁶.”

3) We checked whether rule-following was associated with "respect for authority" as measured by the Moral Foundations Questionnaire (see Appendix E1 of the 2016 paper); it is not. We confirmed this also in table F2 of Kimbrough and Vostroknutov (2015), which relates rule-following to the management of common pool resources. The authors may wish to note this in response to concerns about the role of authority in the experiment.

We now cite this paper when we discuss the experimenter demand effect.

Finally, since my original report, another paper using the data from the same experiment as Tate et al. (2022) has come out in *Experimental Economics* (Kimbrough et al. 2024). In that paper, we show that rule-following measured via the ball-and-bucket task does not change significantly when measured twice, 10 weeks apart, in a sample of Colombian and Northern Irish adolescents. This further corroborates the finding that RF behavior does not diminish with repetition.

We describe this paper in our section on related evidence (p. 7): “In an experiment with the ball-division task conducted in two waves one month apart with the same individuals, individual propensities to follow the rule turned out to be stable across the two wave (rule-following rates of 60.8% and 62.2%)⁶².”

Kimbrough, E.O., Krupka, E.L., Kumar, R. et al. On the stability of norms and norm-following propensity: a cross-cultural panel study with adolescents. *Exp Econ* 27, 351–378 (2024). <https://doi.org/10.1007/s10683-024-09821-5>

Kimbrough, E.O. and Vostroknutov, A., 2015. The social and ecological determinants of common pool resource sustainability. *Journal of environmental economics and management*, 72, pp.38-53.

Reviewer #2:

Remarks to the Author:

The revision has improved a lot. 1) The motivation at the beginning is now much clearer. 2) The authors refer to experimenter demand effects, strategically placed at the end. 3) The authors convincingly show that beliefs were not biased. 4) The description of "Measuring social expectations of rule-following" is now easy to read and has a clearer focus. 5) is resolved. 6) The discussion is now more focused. Essentially, I have only minor suggestions:

We are glad that you like our revision. Many thanks also for the further suggestions you have provided.

1. Page 5: “we therefore also elicit the functions $n(b^n)$ and $d(b^d)$ “. A short description of the metric of this function should be added here or on page 8. This should include that b^n and b^d denote the percentage of people who disapprove of rule breaking and that $n()$ and $d()$ are monotonically increasing functions bounded in the interval $[0, 1]$. Or, more explicitly, that $n(b^n)$ is the increase in rule compliance as a function of the expected percentage of people who disapprove of rule breaking.

Thank you for this comment. We have responded as follows. On p. 4, where we introduce these functions for the first time, we now write: “Crucially, for social expectations to influence rule-following, it must be the case that people make their rule-conformity dependent on their social beliefs: we therefore also elicit the functions $n(b^n)$ and $d(b^d)$, that is, we elicit people’s rule-conformity for various increasing levels of the percentage of people b^n who disapprove of a rule-violation (resulting in the function $n(b^n)$), and, similarly, for increasing levels of the percentage of people b^d who conform with the rule (resulting in the function $d(b^d)$).”

On p. 8, where we describe the details, we now write: “To measure how social expectations influence rule-conformity, we elicited the conditional conformity functions $n(b^n)$ and $d(b^d)$. Participants had to decide whether to follow or break the rule in five different scenarios. The scenarios provide an exogenous variation in beliefs, presented as quintiles ($[0-20\%]$, ..., $[81-100\%]$), about what others said was socially appropriate to do (in the normative belief condition) or what others did in the task (in the descriptive belief condition).”

2. I am a bit confused by the numbering and the alphabetical order. Is it correct that Experiment 1 consists of versions a and b? The authors might want to state this in the sentence "In Experiment 1 we observed a high degree of rule conformity".

In response to this comment, we have re-written this section (see p. 6): “Experiment 1 was run in two versions: with the traffic light task (Experiment 1a) and with the abstract task (Experiment 1b). In Experiment 1a we observed a high degree of rule conformity (Fig. 1c,d). The average rate of rule-following across the two control questions conditions was 65.6%. In the condition without control questions ($n=4,970$), 70% of participants conformed with the rule. In the condition with control questions ($n=2,762$) rule-following was 58%.” [...] “Experiment 1b (Fig. 1b) was only run with control questions. The rule-following rate of 60% of Experiment 1b (Fig. 1e) confirmed the results from Experiment 1a with control questions (Fig. 1e). Thus, while salience of incentives reduces rule-following, there still exists substantial rule-following in conditions with control questions in both versions of Experiment 1.”

3. On page 7, you write “The abstract task (Fig. 1b) also included control questions”. What is the added value of the abstract task? The point is probably quite obvious, as participants are familiar with red lights and react in an immediate, unconscious way, which the abstract task avoids. The authors may want to explain their motivation.

Your intuition is right. We have added the following sentence (on p. 5): “This abstract task removes any naturalistic content that might guide people’s behaviour in the traffic light task^{55,56}”.

4. Page 8 reads “Rule-followers are also more likely to be conditional cooperators in a social dilemma task.” Consider moving this sentence further down as it better fits to experiment 2.

We have not followed this advice because we have measured conditional cooperation only in Experiment 1.

5. Page 8: “The results shown in Fig. 1 only inform us about a high prevalence of rule-following when it is costly and when it cannot affect anyone (no externalities). In terms of CRISP, rule conformity C can therefore only be due to intrinsic respect for rules R and social expectations S .” It would help readers get such a short hint already when experiment 1 is introduced.

We write the following on p. 6: “Thus, in terms of CRISP, other than an intrinsic respect for rules and a desire to conform to social expectations, there are no reasons to follow the rule.”

6. Page 11: “Our results show that even arbitrary rules in an anonymous and asocial single decision maker setting generate social expectations that the rule should be followed and is followed, demonstrating a fundamental relationship between rules and social expectations.” This is a nice point where the authors might want to reflect longer and deepen the insight to better carry home the argument. The results show that social expectations do not need a legitimate justification to restrict individual freedom, but operate even when rules do not serve reasonable ends. It is the authority of the rules that counts, not the underlying reasoning.

Thank you for this excellent comment. We have taken it up as follows (see p. 10): “Our results show that even arbitrary rules in an anonymous and asocial setting with only one decision maker generate social expectations that the rule should be followed and is followed, demonstrating a fundamental relationship between rules and social expectations: Social expectations about following the rules can arise even without a justification for a rule that does not serve an obvious purpose. The mere existence of a rule seems to be enough for triggering social expectations of rule-following.”

7. The motivation for Experiment 3 is a bit weak. The authors may want to add that the descriptive rule violation in Experiment 2 is cold and may not fully influence the rule violation. The question is therefore whether rule compliance can survive even when actual rule violations are observed among others.

Thank you for this comment. We have added the following sentences (on p. 10): “However, eliciting conditional rule-conformity with the strategy method might be considered psychologically “cold” and observing others actually following or breaking the rule might lead to more visceral rule-following or rule-breaking. Moreover, in many

naturally occurring situations, people often observe whether other people observe the rules or not.”

8. Page 12: “In a complementary analysis ...”. There is no added value in this paragraph. I think it can be deleted but leave the decision to the authors.

We have left it in because the editor in the last round suggested we should not remove data. We have therefore moved the results to Extended Data Fig. 3.

9. Page 14: “All experiments were conducted between-subjects and with control questions before the start of the experiment.” For ease of understanding, the authors may wish to add here that information on normative and descriptive beliefs has been added.

We write the following (on p. 12/13): “Our final set of experiments (Experiment 4) adds consequences for others (“an externality”) and weak and strong extrinsic incentives as motives for following the rule. Experiments 4 also measure social expectations (b^n, b^d) and elicit conditional conformity with them ($n(b^n)$ and $d(b^d)$). We use between-subjects designs and the same techniques as in Experiments 2 to see, in the CRISP framework, how consequences for others and extrinsic incentives affect social expectations, and conditional conformity with them.”

10. Pages 17 and 18: “The 22% and 23% of people who follow the rule unconditionally”. Add an explanation of where these figures come from. Are these the intercepts from Experiment 2? If so, they should be 33% and 22%. A more obvious choice would be the 35% and 28% from Experiment 2. This choice would also fit better with the following sentence “Moving from the lowest quintile of beliefs (0-20%) to the highest quintile of beliefs (81-100%) increased average rule-following rates by 20 and 23 percentage points”.

The 22% and the 23% refer to unconditional rule-following as elicited with the strategy method in Experiments 2 and 4 (treatment BL). See caption of Table 1. They are a conservative estimate of people who follow rules across all five quintiles of social expectations b^n and b^d . In response to this comment, we have made some slight changes to Table 1 and it’s caption which hopefully clarifies the issues raised here.

11. The authors have now introduced a quantitative breakdown of the motivations for compliance. What strikes me is that the numbers do not add up to a total of 78% norm compliance, and the authors make no effort to somehow reach this threshold. A good guess at the numbers would be nice, e.g. maximum compliance of 78% = strong punishment 16 + social preferences 7 + social expectations 23 + unconditional 32 (or else). It would also be more intuitive if the second column in Table 1 gave such a calculation, in which case the entry in row 1 “-13” would not fit.

The numbers add up in Table 4c, which the comprehensive set of nested experiments BL (baseline); EX (baseline + externality present); WP (EX + weak punishment) and SP (WP + strong punishment). We hope that the amended Table 4 also clarifies this

comment.

Overall, the study is now a nice contribution to Nature Human Behaviour and I think the authors can take care of the remaining points at their own discretion.

Thanks again for all the detailed comments. We are glad you like our paper.

Reviewer #3:

Remarks to the Author:

In my view, the authors did an excellent job addressing my previous comments and concerns. The addition of Table 1 is particularly interesting, as it neatly summarizes the behavioral effects of different interventions or peer observations.

Thank you for your comments, we glad you like our revision.

One aspect that remains somewhat puzzling to me is the substantial change in rule-following behavior when comprehension/control questions are introduced. This is especially intriguing also because it seems to introduce more inconsistencies in the rank-ordering of conditional rule-following (Fig. S3). While the white dots generally follow a logical rank-order, the black dots appear somewhat more inconsistent. I would have expected more noise or inconsistencies without control questions, but the opposite seems to be the case (although, I base this on eye-balling the data as plotted in Figure S3, of course). Hence, the two comprehension questions (a) change behavior (towards less rule-following) but also (b) the pattern of (conditional) choices. This makes me wonder how comprehension questions in economic games/tasks influence observations and conclusions, more generally. It is more of a broader methodological puzzle for the field. Hence, I would leave it up to the authors, if they think it is worth highlighting with one or two sentences in the discussion, as studies usually do not implement manipulations on comprehension checks and this study speaks to the fact that we maybe should.

While we agree that practice of adding or not comprehension questions is methodologically interesting, but we feel that it is beyond the scope of this paper. We have, however, added a sentence on p. 16 where we write: "Control questions can reduce misperceptions⁷⁸ but also induce more reflective thinking⁷⁹."

Lastly, I have one minor point. On page 13 the authors write: "This observation is consistent with the "bad apple effect", according to which just one defector can diminish cooperation in social dilemma problems". There is a previous study that used the rule-following task to show that 'rule-followers' can actually induce higher rule compliance (in rule-violators) in a collaborative cheating task, somewhat at odds with the bad apple effect. It may be worth citing this study, as it uses the rule-following task and examines (strategic) interactions between 'rule-followers' and 'rule-violators'; something not covered by CRISP; Gross & De Dreu (2021). "Rule following mitigates

collaborative cheating and facilitates the spreading of honesty within groups." Personality and Social Psychology Bulletin, 47(3), 395-409.

Thank you for alerting us to this study, which we now cite on p. 12 where we write: "This observation is consistent with the "bad apple effect", according to which just one defector can diminish cooperation in social dilemma problems⁷⁴ (it should be noted, however, that, depending on the environment, rule-following might have positive spillover effects on other behaviours such as honesty⁷⁵)."

Other than that, I would like to congratulate the authors on a fascinating study!

Thank you, we are glad you like our paper!

Reviewer #4:

Remarks to the Author:

Thank you very much for giving me a chance to revisit this manuscript. I have read the response to reviewers and found that the authors responded to every single point raised and took appropriate steps to address them. I find the CRISP formula a good compromise between demands of different disciplines and found the argument for changes convincing in all cases.

I thus have no further comment save for the suggestion that the final graphics files might need to be of higher resolution (or in vector format as appropriate; they appear pixelated at the end of the document).

Thank you very much. We are glad you like our revision. We will take care of the figure quality.